# White-Box Transformers via Sparse Rate Reduction

Yaodong Yu[1]  Sam Buchanan[2]  Druv Pai[1]  Tianzhe Chu[1]  Ziyang Wu[1]  Shengbang Tong[1]

Benjamin D. Haeffele[3]  Yi Ma[1]

[1]University of California, Berkeley    [2]TTIC    [3]Johns Hopkins University

## Abstract

In this paper, we contend that the objective of representation learning is to compress and transform the distribution of the data, say sets of tokens, towards a mixture of low-dimensional Gaussian distributions supported on incoherent subspaces. The quality of the final representation can be measured by a unified objective function called *sparse rate reduction*. From this perspective, popular deep networks such as transformers can be naturally viewed as realizing iterative schemes to optimize this objective incrementally. Particularly, we show that the standard transformer block can be derived from alternating optimization on complementary parts of this objective: the multi-head self-attention operator can be viewed as a gradient descent step to compress the token sets by minimizing their lossy coding rate, and the subsequent multi-layer perceptron can be viewed as attempting to sparsify the representation of the tokens. This leads to a family of *white-box* transformer-like deep network architectures which are mathematically fully interpretable. Despite their simplicity, experiments show that these networks indeed learn to optimize the designed objective: they compress and sparsify representations of large-scale real-world vision datasets such as ImageNet, and achieve performance very close to thoroughly engineered transformers such as ViT. Code is at https://github.com/Ma-Lab-Berkeley/CRATE.

## 1 Introduction

In recent years, deep learning has seen tremendous empirical success in processing massive amounts of high-dimensional and multi-modal data. Much of this success is owed to effective learning of the data distribution and then transforming the distribution to a parsimonious, i.e. *structured and compact*, representation [39, 50, 52, 62], which facilitates many downstream tasks (e.g., in vision, classification [23, 40], recognition and segmentation [25, 38, 77], and generation [31, 65, 66]). To this end, many models and methods have been proposed and practiced, each with its own strengths and limitations. Here, we give several popular methods a brief accounting as context for a complete understanding and unification that we seek in this work.

**Transformer models and self-attention.** Transformers [28] are one of the latest popular models for learning a representation for high-dimensional structured data, such as text [28, 30, 37], images [40, 75], and other types of signals [48, 57]. After the first block, which converts each data point (such as a text corpus or image) into a set or sequence of *tokens*, further processing is performed on the token sets, in a medium-agnostic manner [28, 40]. A cornerstone of the transformer model is the so-called *self-attention layer*, which exploits the statistical correlations among the sequence of tokens to refine the token representation. Transformers have been highly successful in learning compact representations that perform well on many downstream tasks. Yet the transformer network

---

[1]{yyu,yima}@eecs.berkeley.edu, {druvpai,chutzh,zywu,tsb}@berkeley.edu
[2]sam@ttic.edu
[3]bhaeffele@jhu.edu

37th Conference on Neural Information Processing Systems (NeurIPS 2023).

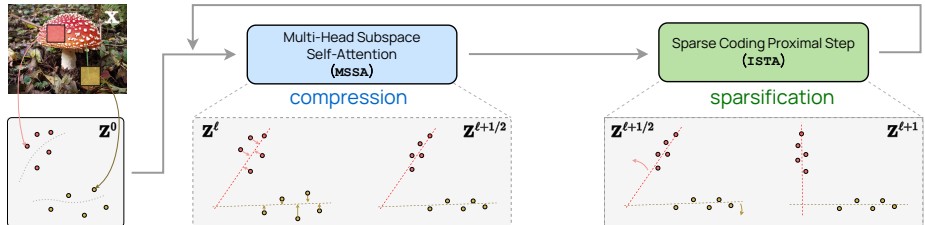

**Figure 1: The 'main loop' of the CRATE white-box deep network design.** After encoding input data $\boldsymbol{X}$ as a sequence of tokens $\boldsymbol{Z}^0$, CRATE constructs a deep network that transforms the data to a canonical configuration of low-dimensional subspaces by successive ***compression*** against a local model for the distribution, generating $\boldsymbol{Z}^{\ell+1/2}$, and ***sparsification*** against a global dictionary, generating $\boldsymbol{Z}^{\ell+1}$. Repeatedly stacking these blocks and training the model parameters via backpropagation yields a powerful and interpretable representation of the data.

architecture is empirically designed and lacks a rigorous mathematical interpretation. In fact, the output of the attention layer itself has several competing interpretations [68, 78]. As a result, the statistical and geometric relationship between the data distribution and the final representation learned by a transformer largely remains a mysterious black box.

**Diffusion models and denoising.** Diffusion models [22, 34, 41, 43, 44] have recently become a popular method for learning the data distribution, particularly for generative tasks and natural image data which are highly structured but notoriously difficult to effectively model [3, 5]. The core concept of diffusion models is to start with features sampled from a Gaussian noise distribution (or some other standard template) and *iteratively denoise* and deform the feature distribution until it converges to the original data distribution. This process is computationally intractable if modeled in just one step [61], so it is typically broken into multiple incremental steps. The key to each step is the so-called *score function*, or equivalently [13] an estimate for the "optimal denoising function"; in practice this function is modeled using a generic black-box deep network. Diffusion models have shown effectiveness at learning and sampling from the data distribution [56, 60, 65]. However, despite some recent efforts [81], they generally do not establish any clear correspondence between the initial features and data samples. Hence, diffusion models themselves do not offer a parsimonious or interpretable representation of the data distribution.

**Structure-seeking models and rate reduction.** In both of the previous two methods, the representations were constructed implicitly as a byproduct of solving a downstream task (e.g., classification or generation/sampling) using deep networks. However, one can also explicitly learn a representation of the data distribution as a task in and of itself; this is most commonly done by trying to identify and represent low-dimensional structures in the input data. Classical examples of this paradigm include model-based approaches such as sparse coding [2, 29] and dictionary learning [17, 21, 47], out of which grew early attempts at designing and interpreting deep network architectures [18, 32]. More recent approaches build instead from a model-free perspective, where one learns a representation through a sufficiently-informative pretext task (such as compressing similar and separating dissimilar data in contrastive learning [45, 69, 80], or maximizing the information gain in the class of maximal coding rate reduction methods [6, 46, 55]). Compared to black-box deep learning approaches, both model-based and model-free representation learning schemes have the advantage of being more interpretable: they allow users to explicitly design desired properties of the learned representation [46, 55, 63]. Furthermore, they allow users to construct new white-box forward-constructed deep network architectures [11, 55, 59] by *unrolling the optimization strategy for the representation learning objective*, such that each layer of the constructed network implements an iteration of the optimization algorithm [11, 53, 55]. Several recent works [71, 74, 76] consider the connections between transformer architectures [28] and unrolled optimization. Unfortunately, in this paradigm, if the desired properties are narrowly defined, it may be difficult to achieve good practical performance on large real-world datasets.

**Our contributions, and outline of this work.** In this work, we aim to remedy the limitations of these existing methods with a more unified framework for designing transformer-like network architectures that leads to both mathematical interpretability and good practical performance. To this end, we propose to learn a sequence of *incremental mappings* to obtain a most *compressed and sparse* representation for the input data (or their token sets) that optimizes *a unified objective function* known as the sparse rate reduction, specified later in (1). The goal of the mapping is illustrated in Figure 1. Within this framework, we unify the above three seemingly disparate approaches and show that *transformer-like deep network layers can be naturally derived from unrolling iterative*

*optimization schemes to incrementally optimize the sparse rate reduction objective.* In particular, our contributions and outline of the paper are as follows:

- In Section 2.2 we show, using an idealized model for the token distribution, that if one *iteratively denoises* the tokens towards a family of low-dimensional subspaces, the associated score function assumes an explicit form similar to a self-attention operator seen in transformers.

- In Section 2.3 we derive the multi-head self-attention layer as an unrolled gradient descent step to minimize the lossy coding rate part of the rate reduction, showing another interpretation of the self-attention layer as compressing the token representation.

- In Section 2.4 we show that the multi-layer perceptron which immediately follows the multi-head self-attention in transformer blocks can be interpreted as (and replaced by) a layer which incrementally optimizes the remaining part of the sparse rate reduction objective by constructing a sparse coding of the token representations.

- In Section 2.5 we use this understanding to create a new white-box (fully mathematically interpretable) transformer architecture called CRATE (i.e., Coding RAte reduction TransformEr), where each layer performs a *single step* of an alternating minimization algorithm to optimize the sparse rate reduction objective.

Hence, within our framework, the learning objective function, the deep learning architecture, and the final learned representation *all become white boxes* that are fully mathematically interpretable. As the experiments in Section 3 show, the CRATE networks, despite being simple, can already learn the desired compressed and sparse representations on large-scale real-world datasets and achieve performance on par with much more heavily engineered transformer networks (such as ViT) on a wide variety of tasks (e.g., classification and transfer learning).

## 2 Technical Approach and Justification

### 2.1 Objective and Approach

We consider a general learning setup associated with real-world signals. We have some random variable $\boldsymbol{X} = [\boldsymbol{x}_1, \ldots, \boldsymbol{x}_N] \in \mathbb{R}^{D \times N}$ which is our data source; each $\boldsymbol{x}_i \in \mathbb{R}^D$ is interpreted as a *token*[1], and the $\boldsymbol{x}_i$'s may have arbitrary correlation structures. We use $\boldsymbol{Z} = [\boldsymbol{z}_1, \ldots, \boldsymbol{z}_N] \in \mathbb{R}^{d \times N}$ to denote the random variable which defines our representations. Each $\boldsymbol{z}_i \in \mathbb{R}^d$ is the representation of the corresponding token $\boldsymbol{x}_i$. We are given $B \geq 1$ i.i.d. samples $\boldsymbol{X}_1, \ldots, \boldsymbol{X}_B \sim \boldsymbol{X}$, whose tokens are $\boldsymbol{x}_{i,b}$. The representations of our samples are denoted $\boldsymbol{Z}_1, \ldots, \boldsymbol{Z}_B \sim \boldsymbol{Z}$, and those of our tokens are $\boldsymbol{z}_{i,b}$. Finally, for a given network, we use $\boldsymbol{Z}^\ell$ to denote the output of the first $\ell$ layers when given $\boldsymbol{X}$ as input. Correspondingly, the sample outputs are $\boldsymbol{Z}_i^\ell$ and the token outputs are $\boldsymbol{z}_{i,b}^\ell$.

**Objective for learning a structured and compact representation.**    Following the framework of rate reduction [55], we contend that the goal of representation learning is to find a feature mapping $f\colon \boldsymbol{X} \in \mathbb{R}^{D \times N} \to \boldsymbol{Z} \in \mathbb{R}^{d \times N}$ which transforms input data $\boldsymbol{X} \in \mathbb{R}^{D \times N}$ with a potentially nonlinear and multi-modal distribution to a (piecewise) *linearized and compact* feature representation $\boldsymbol{Z} \in \mathbb{R}^{d \times N}$. While the joint distribution of tokens $(\boldsymbol{z}_i)_{i=1}^N$ in $\boldsymbol{Z}$ may be sophisticated (and task-specific), we further contend that it is reasonable and practical to require that the target marginal distribution of individual tokens $\boldsymbol{z}_i$ should be highly compressed and structured, amenable for compact coding. Particularly, we require the distribution to be *a mixture of low-dimensional (say $K$) Gaussian distributions*, such that the $k^{\text{th}}$ Gaussian has mean $\boldsymbol{0} \in \mathbb{R}^d$, covariance $\boldsymbol{\Sigma}_k \succeq \boldsymbol{0} \in \mathbb{R}^{d \times d}$, and support spanned by the orthonormal basis $\boldsymbol{U}_k \in \mathbb{R}^{d \times p}$. We denote $\boldsymbol{U}_{[K]} = (\boldsymbol{U}_k)_{k=1}^K$ to be the set of bases of all Gaussians. Hence to maximize the *information gain* [62] for the final token representation, we wish to maximize the rate reduction [6, 46] of the tokens, i.e., $\max_{\boldsymbol{Z}} \Delta R(\boldsymbol{Z}; \boldsymbol{U}_{[K]}) = R(\boldsymbol{Z}) - R^c(\boldsymbol{Z}; \boldsymbol{U}_{[K]})$, where $R$ and $R^c$ are estimates of lossy coding rates to be formally defined in (7) and (8). This also promotes token representations $\boldsymbol{z}_i$ from different Gaussians to be *incoherent* [46]. Since rate reduction is an intrinsic measure of goodness for the representation, it is invariant to arbitrary rotations of the representations. Therefore, to ensure the final representations are amenable to more compact coding, we would like to transform the representations (and their supporting subspaces) so that they become *sparse* with respect to the standard coordinates of the resulting

---

[1]For language transformers, tokens roughly correspond to words [28], while for vision transformers, tokens correspond to image patches [40].

representation space.[2] The combined rate reduction and sparsification process is illustrated in Figure 1. Computationally, we may combine the above two goals into a unified objective for optimization:

$$\max_{f \in \mathcal{F}} \mathbb{E}_{\boldsymbol{Z}}\big[\Delta R(\boldsymbol{Z}; \boldsymbol{U}_{[K]}) - \lambda \|\boldsymbol{Z}\|_0\big] = \max_{f \in \mathcal{F}} \mathbb{E}_{\boldsymbol{Z}}\big[R(\boldsymbol{Z}) - R^c(\boldsymbol{Z}; \boldsymbol{U}_{[K]}) - \lambda \|\boldsymbol{Z}\|_0\big] \text{ s.t. } \boldsymbol{Z} = f(\boldsymbol{X}), \quad (1)$$

where the $\ell^0$ norm $\|\boldsymbol{Z}\|_0$ promotes the sparsity of the final token representations $\boldsymbol{Z} = f(\boldsymbol{X})$.[3] We call this objective "*sparse rate reduction.*"

**White-box deep architecture as unrolled incremental optimization.** Although easy to state, each term of the above objective can be computationally very challenging to optimize [55, 70]. Hence it is natural to take an approximation approach that realizes the global transformation $f$ optimizing (1) through a concatenation of multiple, say $L$, simple *incremental and local* operations $f^\ell$ that push the representation distribution towards the desired parsimonious model distribution:

$$f: \boldsymbol{X} \xrightarrow{f^0} \boldsymbol{Z}^0 \to \cdots \to \boldsymbol{Z}^\ell \xrightarrow{f^\ell} \boldsymbol{Z}^{\ell+1} \to \cdots \to \boldsymbol{Z}^L = \boldsymbol{Z}, \quad (2)$$

where $f^0 : \mathbb{R}^D \to \mathbb{R}^d$ is the pre-processing mapping that transforms input tokens $\boldsymbol{x}_i \in \mathbb{R}^D$ to their token representations $\boldsymbol{z}_i^1 \in \mathbb{R}^d$.

Each incremental *forward mapping* $\boldsymbol{Z}^{\ell+1} = f^\ell(\boldsymbol{Z}^\ell)$, or a "layer", transforms the token distribution to *optimize* the above sparse rate reduction objective (1), conditioned on the distribution of its input tokens $\boldsymbol{Z}^\ell$. In contrast to other unrolled optimization approaches such as the ReduNet [55], we *explicitly model* the distribution of $\boldsymbol{Z}^\ell$ at each layer, say as a mixture of linear subspaces or sparsely generated from a dictionary. The model parameters are learned from data (say via *backward propagation* with end-to-end training). This separation of forward "optimization" and backward "learning" clarifies the mathematical role of each layer as an operator transforming the distribution of its input, whereas the input distribution is in turn modeled (and subsequently learned) by the parameters of the layer.

We show that we can derive these incremental, local operations through an unrolled optimization perspective to achieve (1) through Sections 2.3 to 2.5. Once we decide on using an incremental approach to optimizing (1), there are a variety of possible choices to achieve the optimization. Given a model for $\boldsymbol{Z}^\ell$, say a mixture of subspaces $\boldsymbol{U}_{[K]}$, we opt for a two-step *alternating minimization* process with a strong conceptual basis: first in Section 2.3, we *compress* the tokens $\boldsymbol{Z}^\ell$ via a gradient step to minimize the coding rate term $\min_{\boldsymbol{Z}} R^c(\boldsymbol{Z}; \boldsymbol{U}_{[K]})$; second, in Section 2.4, we *sparsify* the compressed tokens, with a suitably-relaxed proximal gradient step on the difference of the sparsity penalty and the expansion term, i.e., $\min_{\boldsymbol{Z}}[\lambda \|\boldsymbol{Z}\|_0 - R(\boldsymbol{Z})]$. Both actions are applied incrementally and repeatedly, as each $f^\ell$ in (2) is instantiated with these two steps.

## 2.2 Self-Attention via Denoising Tokens Towards Multiple Subspaces

There are many different ways to optimize the objective (1) incrementally. In this work, we propose arguably *the most basic* scheme. To help clarify the intuition behind our derivation and approximation, in this section (and Appendix A.1) we study a largely idealized model which nevertheless captures the essence of nearly the whole process and particularly reveals the reason why self-attention-like operators arise in many contexts. Assume that $N = 1$, and the single token $\boldsymbol{x}$ is drawn i.i.d. from an unknown mixture of Gaussians $(\mathcal{N}(\boldsymbol{0}, \boldsymbol{\Sigma}_k))_{k=1}^K$ supported on low-dimensional subspaces with orthonormal bases $\boldsymbol{U}_{[K]} = (\boldsymbol{U}_k)_{k=1}^K$ and corrupted with additive Gaussian noise $\boldsymbol{w} \sim \mathcal{N}(\boldsymbol{0}, \boldsymbol{I})$, i.e.,

$$\boldsymbol{x} = \boldsymbol{z} + \sigma \boldsymbol{w}, \quad (3)$$

where $\boldsymbol{z}$ is distributed according to the mixture. Our goal is simply to transform the distribution of the noisy token $\boldsymbol{x}$ to the mixture of low-dimensional Gaussians $\boldsymbol{z}$. Towards incremental construction of a representation $f$ for this model following (2), we reason inductively: if $\boldsymbol{z}^\ell$ is a noisy token (3) at noise level $\sigma^\ell$, it is natural to produce $\boldsymbol{z}^{\ell+1}$ by denoising at the level $\sigma^\ell$. In the mean-square sense, the optimal estimate is $\mathbb{E}[\boldsymbol{z} \mid \boldsymbol{z}^\ell]$, which has a variational characterization (e.g. [12]):

$$\mathbb{E}[\boldsymbol{z} \mid \cdot] = \arg\min_f \mathbb{E}_{\boldsymbol{z}, \boldsymbol{w}}\Big[\big\|f(\boldsymbol{z} + \sigma^\ell \boldsymbol{w}) - \boldsymbol{z}\big\|_2^2\Big]. \quad (4)$$

---

[2]That is, having the fewest nonzero entries.

[3]To simplify the notation, we will discuss the objective for one sample $\boldsymbol{X}$ at a time with the understanding that we always mean to optimize the expectation.

Setting $z^{\ell+1} = \mathbb{E}[z \mid z^\ell]$, (4) thus characterizes the next stage of (2) in terms of an optimization objective based on a *local signal model* for $z^\ell$. Moreover, letting $x \mapsto q^\ell(x)$ denote the density of $z^\ell$, Tweedie's formula [13] allows us to express the optimal representation solving (4) in closed-form:

$$z^{\ell+1} = z^\ell + (\sigma^\ell)^2 \nabla_x \log q^\ell(z^\ell). \tag{5}$$

Tweedie's formula expresses the optimal representation in terms of an additive correction (in general a nonlinear function of $z^\ell$) to the noisy observations by the gradient of the *log-likelihood* of the distribution of the noisy observations, giving the optimal representation a clear interpretation as an incremental perturbation to the current noisy distribution $q^\ell$. This connection is well-known in the areas of estimation theory and inverse problems [1, 13, 14, 19, 20, 27, 42], and more recently has found powerful applications in the training of generative models for natural images [4, 15, 22, 43, 44]. Here, we can calculate a closed-form expression for this *score function* $\nabla_x \log q^\ell$, which, when combined with (5) and some technical assumptions[4], gives the following approximation (shown in Appendix A.1). Let $\otimes$ denote the Kronecker product; then we have

$$z^{\ell+1} \approx [U_1, \ldots, U_K] \left[ \mathrm{diag} \left( \mathrm{softmax} \left( \frac{1}{2(\sigma^\ell)^2} \begin{bmatrix} \|U_1^* z^\ell\|_2^2 \\ \vdots \\ \|U_K^* z^\ell\|_2^2 \end{bmatrix} \right) \right) \otimes I_p \right] \begin{bmatrix} U_1^* z^\ell \\ \vdots \\ U_K^* z^\ell \end{bmatrix}, \tag{6}$$

This operation resembles a self-attention layer in a standard transformer architecture with $K$ heads, sequence length $N = 1$, the "query-key-value" constructs being replaced by a single linear projection $U_k^* z^\ell$ of the token $z^\ell$, and the aggregation of head outputs (conventionally modeled by an MLP) done with the two leftmost matrices in (6). We thus derive the following useful interpretation, which we will exploit in the sequel: *Gaussian denoising against a mixture of subspaces model leads to self-attention-type layers in the transformation $f$.* Given an initial sample $x$ following the model (3), we can repeatedly apply local transformations to the distribution with (6) in order to realize the incremental mapping $f: x \to z$ in (2).[5] These insights will guide us in the design of our white-box transformer architecture in the upcoming subsections.

## 2.3 Self-Attention via Compressing Token Sets through Optimizing Rate Reduction

In the last subsection, we have seen that the multi-head attention in a transformer resembles the score-matching operator that aims to transform a token $z^\ell$ towards a mixture of subspaces (or degenerate Gaussians). Nevertheless, to carry out such an operation on any data, one needs to first learn or estimate, typically from finite samples, the parameters of the mixture of (degenerate) Gaussians, which is known to be a challenging task [6, 24]. This challenge is made even harder because in a typical learning setting, the given set of tokens are *not* i.i.d. samples from the mixture of subspaces. The joint distribution among these tokens can encode rich information about the data—for example, co-occurrences between words or object parts in language and image data (resp.)—which we should also learn. Thus, we should compress / denoise / transform such a set of tokens together. To this end, we need a measure of quality, i.e., compactness, for the resulting representation of the set of tokens.

A natural measure of the compactness of such a set of tokens is the (lossy) coding rate to encode them up to a certain precision $\epsilon > 0$ [6, 46]. For a zero-mean Gaussian, this measure takes a closed form. If we view the tokens in $Z \in \mathbb{R}^{d \times N}$ as drawn from a single zero-mean Gaussian, an estimate of their (lossy) coding rate, subject to quantization precision $\epsilon > 0$, is given in [6] as:

$$R(Z) \doteq \frac{1}{2} \mathrm{logdet} \left( I + \frac{d}{N\epsilon^2} Z^* Z \right) = \frac{1}{2} \mathrm{logdet} \left( I + \frac{d}{N\epsilon^2} Z Z^* \right). \tag{7}$$

In practice, the data distribution is typically multi-modal, say an image set consisting of many classes or a collection of image patches as in Figure 1. It is more appropriate to require that the set of tokens map to a mixture of, say $K$, subspaces (degenerate Gaussians) [55]. As before we denote the (to be learned) bases of these subspaces as $U_{[K]} = (U_k)_{k=1}^K$, where $U_k \in \mathbb{R}^{d \times p}$. Although the joint distribution of the tokens $Z$ is unknown, the desired marginal distribution of each token $z_i$ is a

---

[4]Such as $\sigma$ being smaller than the nonzero eigenvalues of $\Sigma_k$ and the normalization assumption $\pi_i \det(\Sigma_i + \sigma^2 I)^{-1/2} = \pi_j \det(\Sigma_j + \sigma^2 I)^{-1/2}$ for all $i, j \in [K]$, where $\pi_k$ is the mixture proportion for the $k^{\mathrm{th}}$ Gaussian.

[5]This statement can be made mathematically rigorous by exploiting a deep connection between neural ODEs and diffusion models, following ideas in Song et al. [44] and Chen et al. [72].

mixture of subspaces. So we may obtain an upper bound of the coding rate for the token set $\boldsymbol{Z}$ by projecting its tokens onto these subspaces and summing up the respective coding rates:

$$R^c(\boldsymbol{Z};\boldsymbol{U}_{[K]}) = \sum_{k=1}^{K} R(\boldsymbol{U}_k^*\boldsymbol{Z}) = \frac{1}{2}\sum_{k=1}^{K}\operatorname{logdet}\left(\boldsymbol{I} + \frac{p}{N\epsilon^2}(\boldsymbol{U}_k^*\boldsymbol{Z})^*(\boldsymbol{U}_k^*\boldsymbol{Z})\right). \tag{8}$$

We would like to compress (or denoise) the set of tokens against these subspaces by minimizing the coding rate. The gradient of $R^c(\boldsymbol{Z};\boldsymbol{U}_{[K]})$ is

$$\nabla_{\boldsymbol{Z}} R^c(\boldsymbol{Z};\boldsymbol{U}_{[K]}) = \frac{p}{N\epsilon^2}\sum_{k=1}^{K}\boldsymbol{U}_k\boldsymbol{U}_k^*\boldsymbol{Z}\left(\boldsymbol{I} + \frac{p}{N\epsilon^2}(\boldsymbol{U}_k^*\boldsymbol{Z})^*(\boldsymbol{U}_k^*\boldsymbol{Z})\right)^{-1}. \tag{9}$$

The above expression approximates the residual of each projected token $\boldsymbol{U}_k^*\boldsymbol{z}_i$ regressed by other tokens $\boldsymbol{U}_k^*\boldsymbol{z}_j$ [55]. But, differently from [55], not all tokens in $\boldsymbol{Z}$ are from the same subspace. Hence, to denoise each token with tokens from its own group, we can compute their similarity through an auto-correlation among the projected tokens as $(\boldsymbol{U}_k^*\boldsymbol{Z})^*(\boldsymbol{U}_k^*\boldsymbol{Z})$ and convert it to a distribution of membership with a softmax, namely $\operatorname{softmax}((\boldsymbol{U}_k^*\boldsymbol{Z})^*(\boldsymbol{U}_k^*\boldsymbol{Z}))$. Then, as we show in Appendix A.2, if we only use similar tokens to regress and denoise each other, then a gradient step on the coding rate with learning rate $\kappa$ can be naturally approximated as follows:

$$\boldsymbol{Z}^{\ell+1/2} = \boldsymbol{Z}^\ell - \kappa\nabla_{\boldsymbol{Z}}R^c(\boldsymbol{Z}^\ell;\boldsymbol{U}_{[K]}) \approx \left(1 - \kappa\cdot\frac{p}{N\epsilon^2}\right)\boldsymbol{Z}^\ell + \kappa\cdot\frac{p}{N\epsilon^2}\cdot\texttt{MSSA}(\boldsymbol{Z}^\ell \mid \boldsymbol{U}_{[K]}), \tag{10}$$

where MSSA is defined through an SSA operator as:

$$\texttt{SSA}(\boldsymbol{Z}\mid\boldsymbol{U}_k) \doteq (\boldsymbol{U}_k^*\boldsymbol{Z})\operatorname{softmax}((\boldsymbol{U}_k^*\boldsymbol{Z})^*(\boldsymbol{U}_k^*\boldsymbol{Z})), \quad k\in[K], \tag{11}$$

$$\texttt{MSSA}(\boldsymbol{Z}\mid\boldsymbol{U}_{[K]}) \doteq \frac{p}{N\epsilon^2}\cdot[\boldsymbol{U}_1,\ldots,\boldsymbol{U}_K]\begin{bmatrix}\texttt{SSA}(\boldsymbol{Z}\mid\boldsymbol{U}_1)\\\vdots\\\texttt{SSA}(\boldsymbol{Z}\mid\boldsymbol{U}_K)\end{bmatrix}. \tag{12}$$

Here the SSA operator in (11) resembles the *attention operator* in a typical transformer [28], except that here the linear operators of value, key, and query are all set to be *the same* as the subspace basis, i.e., $\boldsymbol{V} = \boldsymbol{K} = \boldsymbol{Q} = \boldsymbol{U}_k^*$.[6] Hence, we name $\texttt{SSA}(\,\cdot\,|\boldsymbol{U}_k):\mathbb{R}^{d\times N}\to\mathbb{R}^{p\times N}$ the **S**ubspace **S**elf-**A**ttention (SSA) operator (more details and justification can be found in (72) in Appendix A.2). Then, the whole MSSA operator in (12), formally defined as $\texttt{MSSA}(\,\cdot\,|\boldsymbol{U}_{[K]}):\mathbb{R}^{d\times N}\to\mathbb{R}^{d\times N}$ and called the **M**ulti-Head **S**ubspace **S**elf-**A**ttention (MSSA) operator, aggregates the attention head outputs by averaging using model-dependent weights, similar in concept to the popular multi-head self-attention operator in existing transformer networks. The overall gradient step (10) resembles the multi-head self-attention implemented with a skip connection in transformers.

Notice that if we have $N = 1$ tokens as well as take an aggressive gradient step ($\kappa = 1$) and tune the quantization error ($\epsilon = \sqrt{p/N}$), the multi-head subspace self-attention operator in (12) becomes the ideal denoiser defined in (6), with the one minor difference that the aggregation of the heads is done by a linear function here, while in (6) it is done by a nonlinear mixture-of-experts type function.[7] This provides two very related interpretations of the multi-head self-attention operator, as denoising and compression against a mixture of low-dimensional subspaces.

### 2.4 MLP via Iterative Shrinkage-Thresholding Algorithms (ISTA) for Sparse Coding

In the previous subsection, we focused on how to compress a set of tokens against a set of (learned) low-dimensional subspaces. Optimizing the remaining terms in the sparse rate reduction objective (1), including the non-smooth term, serves to sparsify the compressed tokens, hence leading to a more compact and structured (i.e., *parsimonious*) representation. From (1) and (7), this term is

$$\max_{\boldsymbol{Z}}\left[R(\boldsymbol{Z}) - \lambda\|\boldsymbol{Z}\|_0\right] = \min_{\boldsymbol{Z}}\left[\lambda\|\boldsymbol{Z}\|_0 - \frac{1}{2}\operatorname{logdet}\left(\boldsymbol{I} + \frac{d}{N\epsilon^2}\boldsymbol{Z}^*\boldsymbol{Z}\right)\right], \tag{13}$$

---

[6]We note a recent suggestion of Hinton [51] that it is more sensible to set the "value, key, and query" projection matrices in a transformer to be equal. Our derivation in this section confirms this mathematically.

[7]This suggests that we could also consider such a mixture of expert type aggregation of the multiple attention heads. In this work, we use linear aggregation, and leave evaluation of more variants for future work.

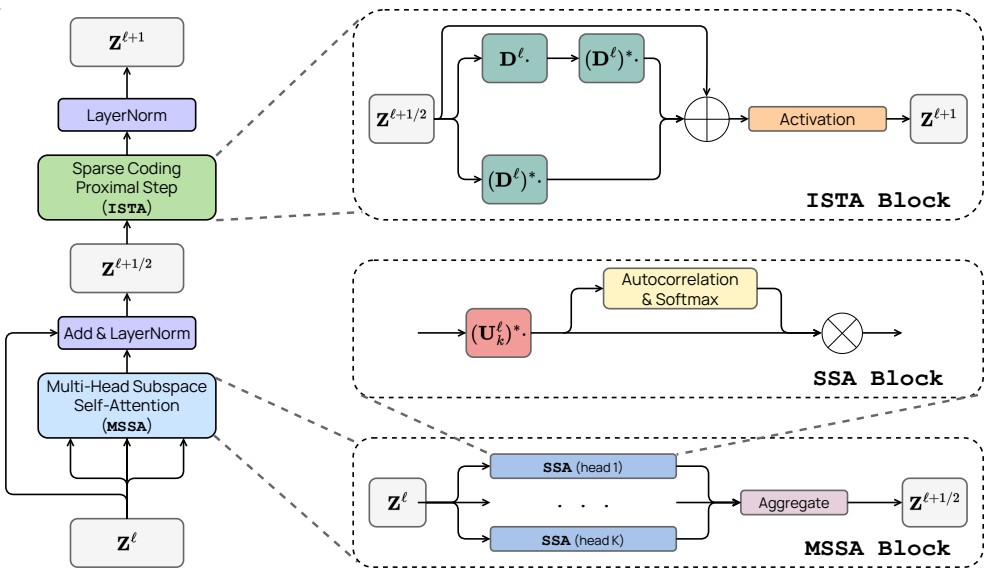

**Figure 2:** One layer of the CRATE architecture. The full architecture is simply a concatenation of such layers, with some initial tokenizer and final task-specific architecture (i.e., a classification head).

where $R(\boldsymbol{Z})$ denotes the coding rate of the whole token set, as defined in (7). In addition to sparsification via the $\|\boldsymbol{Z}\|_0$ term, the expansion term $R(\boldsymbol{Z})$ in (13) promotes diversity and non-collapse of the representation, a highly desirable property. However, prior work has struggled to realize this benefit on large-scale datasets due to poor scalability of the gradient $\nabla_{\boldsymbol{Z}} R(\boldsymbol{Z})$, which requires a matrix inverse [55].

To simplify things, we therefore take a different approach to trading off between representational diversity and sparsification: we posit a (complete) incoherent or orthogonal dictionary $\boldsymbol{D} \in \mathbb{R}^{d \times d}$, and ask to sparsify the intermediate iterates $\boldsymbol{Z}^{\ell+1/2}$ with respect to $\boldsymbol{D}$. That is, $\boldsymbol{Z}^{\ell+1/2} = \boldsymbol{D} \boldsymbol{Z}^{\ell+1}$ where $\boldsymbol{Z}^{\ell+1}$ is more sparse. The dictionary $\boldsymbol{D}$ is global, i.e., is used to sparsify all tokens simultaneously. By the incoherence assumption, we have $\boldsymbol{D}^* \boldsymbol{D} \approx \boldsymbol{I}_d$; thus from (7) we have $R(\boldsymbol{Z}^{\ell+1}) \approx R(\boldsymbol{D} \boldsymbol{Z}^{\ell+1}) = R(\boldsymbol{Z}^{\ell+1/2})$. Thus we approximately solve (13) with the following program:

$$\boldsymbol{Z}^{\ell+1} = \arg\min_{\boldsymbol{Z}} \|\boldsymbol{Z}\|_0 \quad \text{subject to} \quad \boldsymbol{Z}^{\ell+1/2} = \boldsymbol{D} \boldsymbol{Z}. \tag{14}$$

The above sparse representation program is usually solved by relaxing it to an unconstrained convex program, known as LASSO:

$$\boldsymbol{Z}^{\ell+1} = \arg\min_{\boldsymbol{Z}} \left[ \lambda \|\boldsymbol{Z}\|_1 + \|\boldsymbol{Z}^{\ell+1/2} - \boldsymbol{D} \boldsymbol{Z}\|_F^2 \right]. \tag{15}$$

In our implementation, motivated by Sun et al. [33] and Zarka et al. [35], we also add a non-negative constraint to $\boldsymbol{Z}^{\ell+1}$,

$$\boldsymbol{Z}^{\ell+1} = \arg\min_{\boldsymbol{Z} \geq \boldsymbol{0}} \left[ \lambda \|\boldsymbol{Z}\|_1 + \|\boldsymbol{Z}^{\ell+1/2} - \boldsymbol{D} \boldsymbol{Z}\|_F^2 \right], \tag{16}$$

which we then incrementally optimize by performing an unrolled proximal gradient descent step, known as an ISTA step [8], to give the update:

$$\boldsymbol{Z}^{\ell+1} = \mathrm{ReLU}(\boldsymbol{Z}^{\ell+1/2} + \eta \boldsymbol{D}^*(\boldsymbol{Z}^{\ell+1/2} - \boldsymbol{D}\boldsymbol{Z}^{\ell+1/2}) - \eta\lambda\mathbf{1}) \doteq \mathtt{ISTA}(\boldsymbol{Z}^{\ell+1/2} \mid \boldsymbol{D}). \tag{17}$$

In Appendix A.3, we will show one can arrive at a similar operator to the above ISTA-like update for optimizing (13) by properly linearizing and approximating the rate term $R(\boldsymbol{Z})$.

## 2.5 The Overall White-Box CRATE Architecture

By combining the above two steps:

1. (Sections 2.2 and 2.3) Local denoising and compression of tokens within a sample towards a mixture-of-subspace structure, leading to the multi-head subspace self-attention block – MSSA;

2. (Section 2.4) Global compression and sparsification of token sets across all samples through sparse coding, leading to the sparsification block – ISTA;

we can get the following rate-reduction-based transformer layer, illustrated in Figure 2,

$$\boldsymbol{Z}^{\ell+1/2} \doteq \boldsymbol{Z}^\ell + \texttt{MSSA}(\boldsymbol{Z}^\ell \mid \boldsymbol{U}_{[K]}^\ell), \qquad \boldsymbol{Z}^{\ell+1} \doteq \texttt{ISTA}(\boldsymbol{Z}^{\ell+1/2} \mid \boldsymbol{D}^\ell). \tag{18}$$

Composing multiple such layers following the incremental construction of our representation in (2), we obtain a white-box transformer architecture that transforms the data tokens towards a compact and sparse union of incoherent subspaces.

This model has the parameters $(\boldsymbol{U}_{[K]}^\ell)_{\ell=1}^L$ and $(\boldsymbol{D}^\ell)_{\ell=1}^L$, which are learned from data via *back-propagation*. Notably, in each layer $\ell$, the learned $\boldsymbol{U}_{[K]}^\ell$ retain their interpretation as incoherent bases for supporting subspaces for the mixture-of-Gaussians model at layer $\ell$, and the learned $\boldsymbol{D}^\ell$ retains its interpretation as a sparsifying dictionary at layer $\ell$. We emphasize that the parameters $\boldsymbol{U}_{[K]}^\ell$ and $\boldsymbol{D}^\ell$ are dependent on the layer $\ell$ — that is, we learn a different set of parameters at each layer. This is because at each layer we learn an approximate local parametric model for the input data distribution, then use that learned model to construct the layer operators that transform the distribution. Our procedure of parameterizing the data distribution at each layer distinguishes this work from previous works on unrolled optimization for neural networks such as the ReduNet [55]. Our interpretation clarifies the roles of the network forward pass (given local signal models at each layer, denoise/compress/sparsify the input) and the backward pass (learn the local signal models from data via supervision).

We note that in this work, at each stage of our construction, we have chosen arguably the *simplest possible* construction to use. We can substitute each part of this construction, so long as the new part maintains the same conceptual role, and obtain another white-box architecture. Nevertheless, our such-constructed architecture, called CRATE (i.e., Coding RAte TransformEr), connects to existing transformer models, obtains competitive results on real-world datasets, and is fully mathematically interpretable.

## 3 Experiments

In this section, we conduct experiments to study the performance of our proposed white-box transformer CRATE on real-world datasets and tasks. As the analysis in Section 2 suggests, either the compression or the sparsification step can be achieved through various alternative design choices or strategies. CRATE arguably adopts the most basic choices and so our goal with the experiments is *not* simply to compete with other heavily engineered transformers while using such a rudimentary design. Rather, our goals are twofold. First, unlike any empirically designed black-box networks that are usually evaluated only on end-to-end performance, the white-box design of our network allows us to *look inside* the deep architecture and verify if layers of the learned network indeed perform their design objective—say performing incremental optimization for the objective (1). Second, despite their simplicity, our experiments will actually reveal the vast practical potential of our so-derived CRATE architectures since, as we will show, they already achieve very strong performance on large-scale real-world datasets and tasks. In the remainder of this section we highlight a selection of results; additional experimental details and results can be found in Appendix B.

**Model architecture.** We implement the architecture that is described in Section 2.5, with minor modifications that are described in Appendix B.1. We consider different model sizes of CRATE by varying the token dimension $d$, number of heads $K$, and the number of layers $L$. We consider four model sizes in this work: CRATE-Tiny, CRATE-Small, CRATE-Base, and CRATE-Large. A PyTorch-style pseudocode can be found in Appendix B.1, which contains more implementation details. For training using supervised classification, we first take the CLS token $\overline{\boldsymbol{z}}_b = \boldsymbol{z}_{1,b}^{L+1}$ of for each sample, then apply a linear layer; the output of this linear layer $\boldsymbol{u}_b \doteq \boldsymbol{W}\overline{\boldsymbol{z}}_b$ is used as input to the standard cross-entropy loss. The overall loss averages over all samples $b \in [B]$.

**Datasets and optimization.** We mainly consider ImageNet-1K [9] as the testbed for our architecture. Specifically, we apply the Lion optimizer [73] to train CRATE models with different model sizes. Meanwhile, we also evaluate the transfer learning performance of CRATE: by considering the models trained on ImageNet-1K as pre-trained models, we fine-tune CRATE on several commonly used downstream datasets (CIFAR10/100, Oxford Flowers, Oxford-IIT-Pets). More details about the training and datasets can be found in Appendix B.1.

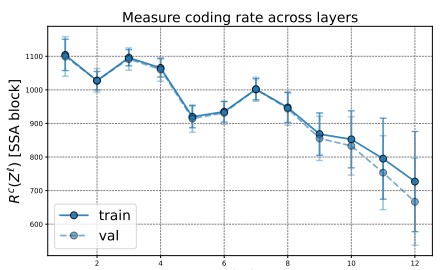
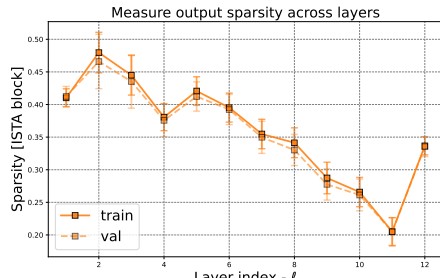

**Figure 3:** *Left*: The compression term $R^c(\boldsymbol{Z}^{\ell+1/2})$ of the MSSA outputs at different layers. *Right*: the sparsity of the ISTA output block, $\|\boldsymbol{Z}^{\ell+1}\|_0/(d \cdot N)$, at different layers. (Model: CRATE-Small).

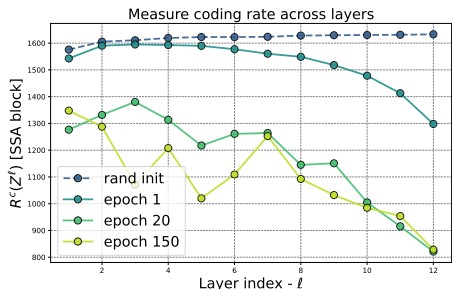
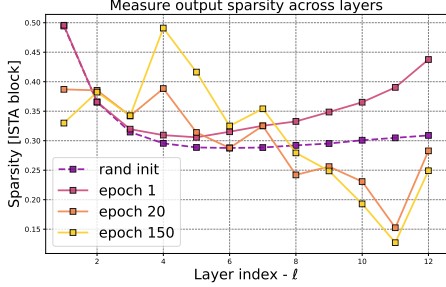

**Figure 4:** The compression term $R^c(\boldsymbol{Z})$ (*left*) and sparsification term $\|\boldsymbol{Z}\|_0/(d \cdot N)$ (*right*) across models trained with different numbers of epochs. (Model: CRATE-Base).

## 3.1 In-depth Layer-wise Analysis of CRATE

**Do layers of CRATE achieve their design goals?** As described in Section 2.3 and Section 2.4, the MSSA block is designed to optimize the compression term $R^c(\boldsymbol{Z})$ and the ISTA block to sparsify the token representations (corresponding to the sparsification term $\|\boldsymbol{Z}\|_0$). To understand whether CRATE indeed optimizes these terms, for each layer $\ell$, we measure (i) the compression term $R^c(\boldsymbol{Z}^{\ell+1/2})$ on the MSSA block outputs $\boldsymbol{Z}^{\ell+1/2}$; and (ii) sparsity $\|\boldsymbol{Z}^{\ell+1}\|_0$ on the ISTA block outputs $\boldsymbol{Z}^{\ell+1}$. Specifically, we evaluate these two terms by using training/validation samples from ImageNet-1K. Both terms are evaluated at the per-sample level and averaged over $B = 10^3$ samples.

Figure 3 shows the plots of these two key measures at all layers for the learned CRATE-small model. We find that as the layer index $\ell$ increases, both the compression and the sparsification terms improve in most cases. The increase in the sparsity measure of the last layer is caused by the extra linear layer for classification.[8] These results suggest that CRATE aligns well with the original design goals: once learned, it essentially learns to gradually compress and sparsity the representations through its layers. In addition, we also measure the compression and sparsification terms on CRATE models with different model sizes as well as intermediate model checkpoints and the results are shown by plots in Figure 5 of Appendix B.2. The observations are very consistent across all different model sizes—both the compression and sparsification terms improve in most scenarios. Models with more layers tend to optimize the objectives more effectively, confirming our understanding of each layer's roles.

To see the effect of learning, we present the evaluations on CRATE-Small trained with different number of epochs in Figure 4. When the model is not trained enough (e.g. untrained), the architecture does not optimize the objectives effectively. However, during training—learning better subspaces $\boldsymbol{U}^\ell_{[K]}$ and dictionaries $\boldsymbol{D}^\ell$—the designed blocks start to optimize the objectives much more effectively.

**Visualizing layer-wise token representations.** To gain a better understanding of the token representations of CRATE, we visualize the output of each ISTA block at layer $\ell$ in Figure 6 of Appendix B.2. Specifically, we visualize the $\boldsymbol{Z}^{\ell+1}$ via heatmap plots. We observe that the output $\boldsymbol{Z}^{\ell+1}$ becomes more sparse as the layer increases. Moreover, besides the sparsity, we also find that $\boldsymbol{Z}^{\ell+1}$ becomes

---

[8]Note that the learned sparse (tokens) features need to be mixed in the last layer for predicting the class. The phenomenon of increase in the sparsity measure at the last layer suggests that each class of objects may be associated with a number of features, and some of these features are likely to be shared across different classes.

**Table 1:** Top 1 accuracy of CRATE on various datasets with different model scales when pre-trained on ImageNet. For ImageNet/ImageNetReaL, we directly evaluate the top-1 accuracy. For other datasets, we use models that are pre-trained on ImageNet as initialization and the evaluate the transfer learning performance via fine-tuning.

| Datasets | CRATE-T | CRATE-S | CRATE-B | CRATE-L | ViT-T | ViT-S |
|---|---|---|---|---|---|---|
| # parameters | 6.09M | 13.12M | 22.80M | 77.64M | 5.72M | 22.05M |
| ImageNet | 66.7 | 69.2 | 70.8 | 71.3 | 71.5 | 72.4 |
| ImageNet ReaL | 74.0 | 76.0 | 76.5 | 77.4 | 78.3 | 78.4 |
| CIFAR10 | 95.5 | 96.0 | 96.8 | 97.2 | 96.6 | 97.2 |
| CIFAR100 | 78.9 | 81.0 | 82.7 | 83.6 | 81.8 | 83.2 |
| Oxford Flowers-102 | 84.6 | 87.1 | 88.7 | 88.3 | 85.1 | 88.5 |
| Oxford-IIIT-Pets | 81.4 | 84.9 | 85.3 | 87.4 | 88.5 | 88.6 |

more structured (i.e., low-rank), which indicates that the set of token representations become closer to linear subspaces, confirming our mental picture of the geometry of each layer (as in Figure 1).

**Visualizing layer-wise subspaces in multi-head self-attention.** We now visualize the $U_{[K]}^\ell$ matrices used in the MSSA block. In Section 2.3, we assumed that $U_{[K]}^\ell$ were incoherent to capture different "views" of the set of tokens. In Fig. 7 of Appendix B.2, we first normalize the columns in each $U_k^\ell$, then we visualize the $[U_1^\ell, \ldots, U_K^\ell]^*[U_1^\ell, \ldots, U_K^\ell] \in \mathbb{R}^{pK \times pK}$. The $(i,j)$-th block in each sub-figure corresponds to $(U_i^\ell)^* U_j^\ell$ for $i, j \in [K]$ at a particular layer $\ell$. We find that the learned $U_{[K]}^\ell$ are approximately incoherent, which aligns well with our assumptions. One interesting observation is that the $U_{[K]}^\ell$ becomes more incoherent when the layer index $\ell$ is larger, which suggests that the token representations are more separable. This mirrors the situation in other popular deep networks [58].

### 3.2 Evalutions of CRATE on Large Real-World Datasets and Tasks

We now study the empirical performance of the proposed networks by measuring their top-1 accuracy on ImageNet-1K as well as transfer learning performance on several widely used downstream datasets. We summarize the results in Table 1. As our designed architecture leverages parameter sharing in both the attention block (MSSA) and the MLP block (ISTA), our CRATE-Base model (22.08 million) has a similar number of parameters to the ViT-Small (22.05 million).

From Table 1, we find that with a similar number of model parameters, our proposed network achieves similar ImageNet-1K and transfer learning performance as ViT, despite the simplicity and interpretability of our design. Moreover, with the same set of training hyperparameters, we observe promising scaling behavior in CRATE—we consistently improve the performance by scaling up the model size. For comparison, directly scaling ViT on ImageNet-1K does not always lead to consistent performance improvement measured by top-1 accuracy [40]. To summarize, we achieve promising performance on real-world large-scale datasets by directly implementing our principled architecture.

## 4 Conclusion

In this paper, we propose a new theoretical framework that allows us to derive deep transformer-like network architectures as incremental optimization schemes to learn compressed and sparse representation of the input data (or token sets). The so derived and learned deep architectures are not only fully mathematically interpretable, but also consistent on a layer-by-layer level with their design objective. Despite being arguably the simplest among all possible designs, these networks already demonstrate performance on large-scale real-world datasets and tasks close to seasoned transformers. We believe this work truly helps bridge the gap between theory and practice of deep neural networks as well as help unify seemingly separate approaches to learning and representing data distributions. Probably more importantly for practitioners, our framework provides theoretical guidelines to design and justify new, potentially more powerful, deep architectures for representation learning.

## Acknowledgements

We thank the anonymous reviewers for their helpful comments. Yaodong Yu would like to thank Kwan Ho Ryan Chan for the valuable discussions we had regarding visualizing tokens in vision transformers. Yaodong Yu acknowledges support from the joint Simons Foundation-NSF DMS grant #2031899. Yi Ma acknowledges support from ONR grant N00014-22-1-2102 and the joint Simons Foundation-NSF DMS grant #2031899. This work was partially supported by NSF 1704458, the Northrop Grumman Mission Systems Research in Applications for Learning Machines (REALM) initiative, NIH NIA 1R01AG067396, and ARO MURI W911NF-17-1-0304.

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

# Appendix

## A    Technical Details from Section 2

### A.1    Companion to Section 2.2

We first wish to re-iterate the core contributions of our approach in Section 2.2 at a slightly more technical level. Connections between denoising and score matching are well-understood [60], and computing the optimal denoising function (i.e., the conditional expectation) against a mixture-of-Gaussians model is a rather simple computation giving existing tools such as Tweedie's formula [13]. These are not our main contributions. Instead, the main contributions of Section 2.2 are two-fold:

- First, we demonstrate a mechanism to learn representations via denoising within a idealized mixture of Gaussian data model for a single token (i.e., with sequence length $N = 1$).

- Second, we illustrate the similarities between a such-derived representation learning scheme and existing self-attention layers within the transformer (with sequence length 1), thus demonstrating an interpretation of the self-attention layer as a generalized mechanism to denoise against a mixture-of-Gaussian-marginal model for a set of tokens.

Now we produce the proofs alluded to in Section 2.2, which mostly form the technical aspects of the first listed contribution. To simplify the proofs, we use the following notation correspondences: $\boldsymbol{x} \mapsto \boldsymbol{z}^{\ell}$, $\boldsymbol{z} \mapsto \boldsymbol{z}^{\ell+1}$, and $\sigma \mapsto \sigma^{\ell}$.

**Proposition 1.** *Let $\boldsymbol{u}_1, \ldots, \boldsymbol{u}_K \in \mathbb{R}^d$ be independent and have distribution $\boldsymbol{u}_k \sim \mathcal{N}(\boldsymbol{0}, \boldsymbol{\Sigma}_k)$ for $\boldsymbol{\Sigma}_k \succeq \boldsymbol{0}$, and let $\boldsymbol{z}$ take value $\boldsymbol{u}_k$ with probability $\pi_k > 0$. Let $\boldsymbol{w} \sim \mathcal{N}(\boldsymbol{0}, \boldsymbol{I}_d)$ be independent of $\boldsymbol{z}$. Let $\boldsymbol{x} \doteq \boldsymbol{z} + \sigma \boldsymbol{w}$. Let $\boldsymbol{x} \mapsto q(\boldsymbol{x})$ be the density of $\boldsymbol{x}$. We define*

$$\boldsymbol{M}_k \doteq (\boldsymbol{\Sigma}_k + \sigma^2 \boldsymbol{I}_d)^{-1/2} \tag{19}$$

*and assume that $\pi_i \det(\boldsymbol{M}_i) = \pi_j \det(\boldsymbol{M}_j)$ for all $1 \leq i \leq j \leq K$. Then we have*

$$\nabla_{\boldsymbol{x}} \log q(\boldsymbol{x}) \tag{20}$$

$$= -[\boldsymbol{M}_1, \cdots, \boldsymbol{M}_K] \left[ \mathrm{diag}\left( \mathrm{softmax}\left( -\frac{1}{2} \begin{bmatrix} \|\boldsymbol{M}_1^* \boldsymbol{x}\|_2^2 \\ \vdots \\ \|\boldsymbol{M}_K^* \boldsymbol{x}\|_2^2 \end{bmatrix} \right) \right) \otimes \boldsymbol{I}_d \right] \begin{bmatrix} \boldsymbol{M}_1^* \boldsymbol{x} \\ \vdots \\ \boldsymbol{M}_K^* \boldsymbol{x} \end{bmatrix}, \tag{21}$$

*where $\otimes$ denotes the Kronecker product, i.e., the block matrix defined by*

$$\boldsymbol{A} \otimes \boldsymbol{B} = \begin{bmatrix} A_{11}\boldsymbol{B} & \cdots & A_{1n}\boldsymbol{B} \\ \vdots & \ddots & \vdots \\ A_{m1}\boldsymbol{B} & \cdots & A_{mn}\boldsymbol{B} \end{bmatrix} \tag{22}$$

*Proof.* Let $u$ be the multinomial random variable such that $\boldsymbol{z} = \boldsymbol{z}_u$, so that $u$ has probability mass function $\pi$. Then by the law of total probability, we have

$$\nabla_{\boldsymbol{x}} \log q(\boldsymbol{x}) = \nabla_{\boldsymbol{x}} \log \sum_{k=1}^{K} q(\boldsymbol{x} \mid k) \pi_k \tag{23}$$

$$= \frac{\sum_{k=1}^{K} \pi_k \nabla_{\boldsymbol{x}} q(\boldsymbol{x} \mid k)}{\sum_{k=1}^{K} q(\boldsymbol{x} \mid k) \pi_k} \tag{24}$$

where $q(\boldsymbol{x} \mid k)$ is the conditional density of $\boldsymbol{x}$ given the event $\{u = k\}$. To compute this quantity, note that *conditional on the value of $u$*, we have

$$\boldsymbol{x} = \boldsymbol{z}_u + \sigma \boldsymbol{w} \sim \mathcal{N}(\boldsymbol{0}, \boldsymbol{\Sigma}_u + \sigma^2 \boldsymbol{I}_d). \tag{25}$$

Thus we have

$$q(\boldsymbol{x} \mid k) = \frac{1}{\sqrt{(2\pi)^d \det(\boldsymbol{\Sigma}_k + \sigma^2 \boldsymbol{I}_d)}} \exp\left( -\frac{1}{2} \boldsymbol{x}^* (\boldsymbol{\Sigma}_k + \sigma^2 \boldsymbol{I}_d)^{-1} \boldsymbol{x} \right), \tag{26}$$

This gives

$$\nabla_{\boldsymbol{x}} q(\boldsymbol{x} \mid k) = -q(\boldsymbol{x} \mid k) \cdot (\boldsymbol{\Sigma}_k + \sigma^2 \boldsymbol{I}_d)^{-1} \boldsymbol{x}. \tag{27}$$

Putting this all together, we get

$$\nabla_{\boldsymbol{x}} \log q(\boldsymbol{x}) \tag{28}$$

$$= -\frac{\sum_{k=1}^{K} q(\boldsymbol{x} \mid k)\pi_k \cdot (\boldsymbol{\Sigma}_k + \sigma^2 \boldsymbol{I}_d)^{-1}\boldsymbol{x}}{\sum_{k=1}^{K} q(\boldsymbol{x} \mid k)\pi_k} \tag{29}$$

$$= -\frac{\sum_{k=1}^{K} \pi_k \det(\boldsymbol{\Sigma}_k + \sigma^2 \boldsymbol{I}_d)^{-1/2} \exp\left(-\frac{1}{2}\boldsymbol{x}^*(\boldsymbol{\Sigma}_k + \sigma^2 \boldsymbol{I}_d)^{-1}\boldsymbol{x}\right) \cdot (\boldsymbol{\Sigma}_k + \sigma^2 \boldsymbol{I}_d)^{-1}\boldsymbol{x}}{\sum_{k=1}^{K} \pi_k \det(\boldsymbol{\Sigma}_k + \sigma^2 \boldsymbol{I}_d)^{-1/2} \exp\left(-\frac{1}{2}\boldsymbol{x}^*(\boldsymbol{\Sigma}_k + \sigma^2 \boldsymbol{I}_d)^{-1}\boldsymbol{x}\right)}. \tag{30}$$

Now define $\boldsymbol{M}_k \doteq (\boldsymbol{\Sigma}_k + \sigma^2 \boldsymbol{I}_d)^{-1/2}$. With this notation, we have

$$\nabla_{\boldsymbol{x}} \log q(\boldsymbol{x}) = -\frac{\sum_{k=1}^{K} \pi_k \det(\boldsymbol{M}_k) \exp\left(-\frac{1}{2}\boldsymbol{x}^* \boldsymbol{M}_k \boldsymbol{M}_k^* \boldsymbol{x}\right) \cdot \boldsymbol{M}_k \boldsymbol{M}_k^* \boldsymbol{x}}{\sum_{k=1}^{K} \pi_k \det(\boldsymbol{M}_k) \exp\left(-\frac{1}{2}\boldsymbol{x}^* \boldsymbol{M}_k \boldsymbol{M}_k^* \boldsymbol{x}\right)} \tag{31}$$

$$= -\frac{\sum_{k=1}^{K} \pi_k \det(\boldsymbol{M}_k) \exp\left(-\frac{1}{2}\|\boldsymbol{M}_k^* \boldsymbol{x}\|_2^2\right) \cdot \boldsymbol{M}_k \boldsymbol{M}_k^* \boldsymbol{x}}{\sum_{k=1}^{K} \pi_k \det(\boldsymbol{M}_k) \exp\left(-\frac{1}{2}\boldsymbol{x}^* \boldsymbol{M}_k \boldsymbol{M}_k^* \boldsymbol{x}\right)}. \tag{32}$$

Given our assumption that each $\pi_k \det(\boldsymbol{M}_k)$ is the same, we have

$$\nabla_{\boldsymbol{x}} \log q(\boldsymbol{x}) \tag{33}$$

$$= -\frac{\sum_{k=1}^{K} \pi_k \det(\boldsymbol{M}_k) \exp\left(-\frac{1}{2}\|\boldsymbol{M}_k^* \boldsymbol{x}\|_2^2\right) \cdot \boldsymbol{M}_k \boldsymbol{M}_k^* \boldsymbol{x}}{\sum_{k=1}^{K} \pi_k \det(\boldsymbol{M}_k) \exp\left(-\frac{1}{2}\|\boldsymbol{M}_k^* \boldsymbol{x}\|_2^2\right)} \tag{34}$$

$$= -\frac{\sum_{k=1}^{K} \exp\left(-\frac{1}{2}\|\boldsymbol{M}_k^* \boldsymbol{x}\|_2^2\right) \cdot \boldsymbol{M}_k \boldsymbol{M}_k^* \boldsymbol{x}}{\sum_{k=1}^{K} \exp\left(-\frac{1}{2}\|\boldsymbol{M}_k^* \boldsymbol{x}\|_2^2\right)} \tag{35}$$

$$= -\sum_{k=1}^{K} \boldsymbol{e}_k^* \operatorname{softmax}\left(-\frac{1}{2}\begin{bmatrix} \|\boldsymbol{M}_1^* \boldsymbol{x}\|_2^2 \\ \vdots \\ \|\boldsymbol{M}_K^* \boldsymbol{x}\|_2^2 \end{bmatrix}\right) \boldsymbol{M}_k \boldsymbol{M}_k^* \boldsymbol{x} \tag{36}$$

$$= -[\boldsymbol{M}_1, \ldots, \boldsymbol{M}_K]\left[\operatorname{diag}\left(\operatorname{softmax}\left(-\frac{1}{2}\begin{bmatrix} \|\boldsymbol{M}_1^* \boldsymbol{x}\|_2^2 \\ \vdots \\ \|\boldsymbol{M}_K^* \boldsymbol{x}\|_2^2 \end{bmatrix}\right)\right) \otimes \boldsymbol{I}_d\right]\begin{bmatrix} \boldsymbol{M}_1^* \boldsymbol{x} \\ \vdots \\ \boldsymbol{M}_K^* \boldsymbol{x} \end{bmatrix}. \tag{37}$$

$\square$

Now we provide a final justification for the result cited in Section 2.2.

**Approximation 2.** *In the setting of Proposition 1, diagonalize $\boldsymbol{\Sigma}_k = \boldsymbol{U}_k \boldsymbol{\Lambda}_k \boldsymbol{U}_k^*$ where $\boldsymbol{U}_k \in \mathbb{R}^{d \times p}$ is orthogonal and $\boldsymbol{\Lambda}_k \succ \boldsymbol{0} \in \mathbb{R}^{p \times p}$ is diagonal.[9] Then we have the approximation*

$$\mathbb{E}[\boldsymbol{z} \mid \boldsymbol{x}] \approx [\boldsymbol{U}_1, \ldots, \boldsymbol{U}_K]\left[\operatorname{diag}\left(\operatorname{softmax}\left(\frac{1}{2\sigma^2}\begin{bmatrix} \|\boldsymbol{U}_1^* \boldsymbol{x}\|_2^2 \\ \vdots \\ \|\boldsymbol{U}_K^* \boldsymbol{x}\|_2^2 \end{bmatrix}\right)\right) \otimes \boldsymbol{I}_p\right]\begin{bmatrix} \boldsymbol{U}_1^* \boldsymbol{x} \\ \vdots \\ \boldsymbol{U}_K^* \boldsymbol{x} \end{bmatrix}. \tag{38}$$

*Proof.* We have

$$\nabla_{\boldsymbol{x}} \log q(\boldsymbol{x}) = -\sum_{k=1}^{K} \boldsymbol{e}_k^* \operatorname{softmax}\left(-\frac{1}{2}\begin{bmatrix} \|\boldsymbol{M}_1^* \boldsymbol{x}\|_2^2 \\ \vdots \\ \|\boldsymbol{M}_K^* \boldsymbol{x}\|_2^2 \end{bmatrix}\right) \boldsymbol{M}_k \boldsymbol{M}_k^* \boldsymbol{x} \tag{39}$$

$$= -\sum_{k=1}^{K} \boldsymbol{e}_k^* \operatorname{softmax}\left(-\frac{1}{2\sigma^2}\begin{bmatrix} \|\sigma \boldsymbol{M}_1^* \boldsymbol{x}\|_2^2 \\ \vdots \\ \|\sigma \boldsymbol{M}_K^* \boldsymbol{x}\|_2^2 \end{bmatrix}\right) \boldsymbol{M}_k \boldsymbol{M}_k^* \boldsymbol{x} \tag{40}$$

---

[9]This assumption can be easily relaxed to $\boldsymbol{\Lambda}_k \succeq \boldsymbol{0}$ for all $k$, but requires some more notation to handle, and the form of the solution does not change. Thus we handle the case where all matrices are full rank for simplicity.

$$= -\sum_{k=1}^{K} \boldsymbol{e}_k^* \operatorname{softmax}\left(\frac{1}{2\sigma^2}\begin{bmatrix} \|\boldsymbol{x}\|_2^2 - \|\sigma \boldsymbol{M}_1^* \boldsymbol{x}\|_2^2 \\ \vdots \\ \|\boldsymbol{x}\|_2^2 - \|\sigma \boldsymbol{M}_K^* \boldsymbol{x}\|_2^2 \end{bmatrix}\right) \boldsymbol{M}_k \boldsymbol{M}_k^* \boldsymbol{x}. \tag{41}$$

Now define $\boldsymbol{P}_k \doteq \boldsymbol{I}_d - \sigma \boldsymbol{M}_k$, and let $\boldsymbol{U}_k^\perp \in \mathbb{R}^{d \times (d-p)}$ be an orthogonal complement of $\boldsymbol{U}_k$. Then we have

$$\boldsymbol{P}_k = \boldsymbol{I}_d - \sigma \boldsymbol{M}_k \tag{42}$$

$$= \boldsymbol{I}_d - \sigma \left(\boldsymbol{\Sigma}_k + \sigma^2 \boldsymbol{I}_d\right)^{-1/2} \tag{43}$$

$$= \boldsymbol{I}_d - \sigma \left(\begin{bmatrix} \boldsymbol{U}_k & \boldsymbol{U}_k^\perp \end{bmatrix} \begin{bmatrix} \boldsymbol{\Lambda}_k & \boldsymbol{0} \\ \boldsymbol{0} & \boldsymbol{0} \end{bmatrix} \begin{bmatrix} \boldsymbol{U}_k^* \\ (\boldsymbol{U}_k^\perp)^* \end{bmatrix} + \sigma^2 \boldsymbol{I}_d\right)^{-1/2} \tag{44}$$

$$= \boldsymbol{I}_d - \sigma \left(\begin{bmatrix} \boldsymbol{U}_k & \boldsymbol{U}_k^\perp \end{bmatrix} \begin{bmatrix} \boldsymbol{\Lambda}_k + \sigma^2 \boldsymbol{I}_p & \boldsymbol{0} \\ \boldsymbol{0} & \sigma^2 \boldsymbol{I}_{d-p} \end{bmatrix} \begin{bmatrix} \boldsymbol{U}_k^* \\ (\boldsymbol{U}_k^\perp)^* \end{bmatrix}\right)^{-1/2} \tag{45}$$

$$= \boldsymbol{I}_d - \begin{bmatrix} \boldsymbol{U}_k & \boldsymbol{U}_k^\perp \end{bmatrix} \begin{bmatrix} \sigma(\boldsymbol{\Lambda}_k + \sigma^2 \boldsymbol{I}_p)^{-1/2} & \boldsymbol{0} \\ \boldsymbol{0} & \sigma \cdot (\sigma^2)^{-1/2} \boldsymbol{I}_{d-p} \end{bmatrix} \begin{bmatrix} \boldsymbol{U}_k^* \\ (\boldsymbol{U}_k^\perp)^* \end{bmatrix} \tag{46}$$

$$= \boldsymbol{I}_d - \begin{bmatrix} \boldsymbol{U}_k & \boldsymbol{U}_k^\perp \end{bmatrix} \begin{bmatrix} (\sigma^{-2}\boldsymbol{\Lambda}_k + \boldsymbol{I}_p)^{-1/2} & \boldsymbol{0} \\ \boldsymbol{0} & \boldsymbol{I}_{d-p} \end{bmatrix} \begin{bmatrix} \boldsymbol{U}_k^* \\ (\boldsymbol{U}_k^\perp)^* \end{bmatrix} \tag{47}$$

$$= \begin{bmatrix} \boldsymbol{U}_k & \boldsymbol{U}_k^\perp \end{bmatrix} \begin{bmatrix} \boldsymbol{I}_p - (\sigma^{-2}\boldsymbol{\Lambda}_k + \boldsymbol{I}_p)^{-1/2} & \boldsymbol{0} \\ \boldsymbol{0} & \boldsymbol{0} \end{bmatrix} \begin{bmatrix} \boldsymbol{U}_k^* \\ (\boldsymbol{U}_k^\perp)^* \end{bmatrix} \tag{48}$$

$$\approx \begin{bmatrix} \boldsymbol{U}_k & \boldsymbol{U}_k^\perp \end{bmatrix} \begin{bmatrix} \boldsymbol{I}_p & \boldsymbol{0} \\ \boldsymbol{0} & \boldsymbol{0} \end{bmatrix} \begin{bmatrix} \boldsymbol{U}_k^* \\ (\boldsymbol{U}_k^\perp)^* \end{bmatrix} \tag{49}$$

$$= \boldsymbol{U}_k \boldsymbol{U}_k^*. \tag{50}$$

Thus $\boldsymbol{P}_k$ is approximately a projection when $\sigma$ is small. Under this algebraic relation, we have

$$\nabla_{\boldsymbol{x}} \log q(\boldsymbol{x}) \tag{51}$$

$$= -\sum_{k=1}^{K} \boldsymbol{e}_k^* \operatorname{softmax}\left(\frac{1}{2\sigma^2}\begin{bmatrix} \|\boldsymbol{x}\|_2^2 - \|\sigma \boldsymbol{M}_1^* \boldsymbol{x}\|_2^2 \\ \vdots \\ \|\boldsymbol{x}\|_2^2 - \|\sigma \boldsymbol{M}_K^* \boldsymbol{x}\|_2^2 \end{bmatrix}\right) \boldsymbol{M}_k \boldsymbol{M}_k^* \boldsymbol{x} \tag{52}$$

$$= -\frac{1}{\sigma^2}\sum_{k=1}^{K} \boldsymbol{e}_k^* \operatorname{softmax}\left(\frac{1}{2\sigma^2}\begin{bmatrix} \|\boldsymbol{x}\|_2^2 - \|(\boldsymbol{I}_d - \boldsymbol{P}_1)^* \boldsymbol{x}\|_2^2 \\ \vdots \\ \|\boldsymbol{x}\|_2^2 - \|(\boldsymbol{I}_d - \boldsymbol{P}_K)^* \boldsymbol{x}\|_2^2 \end{bmatrix}\right) (\boldsymbol{I}_d - \boldsymbol{P}_k)(\boldsymbol{I}_d - \boldsymbol{P}_k)^* \boldsymbol{x} \tag{53}$$

$$\approx -\frac{1}{\sigma^2}\sum_{k=1}^{K} \boldsymbol{e}_k^* \operatorname{softmax}\left(\frac{1}{2\sigma^2}\begin{bmatrix} \|\boldsymbol{P}_1^* \boldsymbol{x}\|_2^2 \\ \vdots \\ \|\boldsymbol{P}_K^* \boldsymbol{x}\|_2^2 \end{bmatrix}\right) (\boldsymbol{I}_d - \boldsymbol{P}_k)(\boldsymbol{I}_d - \boldsymbol{P}_k)^* \boldsymbol{x} \tag{54}$$

$$\approx -\frac{1}{\sigma^2}\sum_{k=1}^{K} \boldsymbol{e}_k^* \operatorname{softmax}\left(\frac{1}{2\sigma^2}\begin{bmatrix} \|\boldsymbol{P}_1^* \boldsymbol{x}\|_2^2 \\ \vdots \\ \|\boldsymbol{P}_K^* \boldsymbol{x}\|_2^2 \end{bmatrix}\right) (\boldsymbol{I}_d - \boldsymbol{P}_k)^* \boldsymbol{x} \tag{55}$$

$$= -\frac{\boldsymbol{x}}{\sigma^2}\sum_{k=1}^{K} \boldsymbol{e}_k^* \operatorname{softmax}\left(\frac{1}{2\sigma^2}\begin{bmatrix} \|\boldsymbol{P}_1^* \boldsymbol{x}\|_2^2 \\ \vdots \\ \|\boldsymbol{P}_K^* \boldsymbol{x}\|_2^2 \end{bmatrix}\right) + \frac{1}{\sigma^2}\sum_{k=1}^{K} \boldsymbol{e}_k^* \operatorname{softmax}\left(\frac{1}{2\sigma^2}\begin{bmatrix} \|\boldsymbol{P}_1^* \boldsymbol{x}\|_2^2 \\ \vdots \\ \|\boldsymbol{P}_K^* \boldsymbol{x}\|_2^2 \end{bmatrix}\right) \boldsymbol{P}_k^* \boldsymbol{x} \tag{56}$$

$$= -\frac{1}{\sigma^2}\boldsymbol{x} + \frac{1}{\sigma^2}\sum_{k=1}^{K} \boldsymbol{e}_k^* \operatorname{softmax}\left(\frac{1}{2\sigma^2}\begin{bmatrix} \|\boldsymbol{P}_1^* \boldsymbol{x}\|_2^2 \\ \vdots \\ \|\boldsymbol{P}_K^* \boldsymbol{x}\|_2^2 \end{bmatrix}\right) \boldsymbol{P}_k^* \boldsymbol{x} \tag{57}$$

$$\approx -\frac{1}{\sigma^2}\boldsymbol{x} + \frac{1}{\sigma^2}\sum_{k=1}^{K} \boldsymbol{e}_k^* \operatorname{softmax}\left(\frac{1}{2\sigma^2}\begin{bmatrix} \|\boldsymbol{U}_1^* \boldsymbol{x}\|_2^2 \\ \vdots \\ \|\boldsymbol{U}_K^* \boldsymbol{x}\|_2^2 \end{bmatrix}\right) \boldsymbol{U}_k \boldsymbol{U}_k^* \boldsymbol{x} \tag{58}$$

$$= -\frac{1}{\sigma^2}\boldsymbol{x} + \frac{1}{\sigma^2}\left[\boldsymbol{U}_1,\cdots,\boldsymbol{U}_K\right]\left[\operatorname{diag}\left(\operatorname{softmax}\left(\frac{1}{2\sigma^2}\begin{bmatrix}\|\boldsymbol{U}_1^*\boldsymbol{x}\|_2^2\\\vdots\\\|\boldsymbol{U}_K^*\boldsymbol{x}\|_2^2\end{bmatrix}\right)\right)\otimes\boldsymbol{I}_p\right]\begin{bmatrix}\boldsymbol{U}_1^*\boldsymbol{x}\\\vdots\\\boldsymbol{U}_K^*\boldsymbol{x}\end{bmatrix}. \quad (59)$$

Plugging this into Tweedie's formula, we have

$$\mathbb{E}[\boldsymbol{z}\mid\boldsymbol{x}]\approx\left[\boldsymbol{U}_1,\cdots,\boldsymbol{U}_K\right]\left[\operatorname{diag}\left(\operatorname{softmax}\left(\frac{1}{2\sigma^2}\begin{bmatrix}\|\boldsymbol{U}_1^*\boldsymbol{x}\|_2^2\\\vdots\\\|\boldsymbol{U}_K^*\boldsymbol{x}\|_2^2\end{bmatrix}\right)\right)\otimes\boldsymbol{I}_p\right]\begin{bmatrix}\boldsymbol{U}_1^*\boldsymbol{x}\\\vdots\\\boldsymbol{U}_K^*\boldsymbol{x}\end{bmatrix}. \quad (60)$$

$\square$

*Remark* 3. Although Approximation 2 is stated as an approximation rather than as a proposition, we believe it should be possible without too much extra work to convert it into a statement of asymptotic equivalence as $\sigma\to 0$ (in particular, holding for $\sigma$ below the smallest (nonzero) eigenvalue of any $\boldsymbol{\Sigma}_k$. Most approximations taken in the derivation of Approximation 2 can immediately be turned into asymptotic claims; the only slightly delicate point is treating the softmax, which can be accomplished using standard "high temperature" convergence behavior of the softmax function (in particular, as $\sigma\to 0$ in our expressions, the softmax concentrates on the "best head").

### A.2   Companion to Section 2.3

We again wish to re-iterate the core contribution of our approach in Section 2.3. The application of a compression perspective to representation learning has been discussed before, for example in the line of maximal coding rate reduction works [46]. In Section 2.3, we provide the following contributions and developments to this perspective:

- We propose a generalized coding rate function $R^c(\cdot;\boldsymbol{U}_{[K]})$ which measures the coding rate with respect to a set of subspaces $\boldsymbol{U}_{[K]}$ as opposed to a set of classes (as in [46, 55]), making the underlying formulation unsupervised.

- We then show how if we adopt the framework of alternating minimization of the sparse rate reduction objective, then unrolling the first alternating step — gradient descent on this coding rate objective — nearly exactly recovers the common multi-head attention mechanism found in transformer networks (except that the query/key/value operators are all the same operation $\boldsymbol{U}_k^*$ now, which we interpret as projection onto a single subspace).

In the process of the second contribution, and in the following proofs, we make some simple approximations and technical assumptions. The validity of these assumptions may be explored, and the approximations refined, altogether providing a more complex (and possibly more performant) resulting self-attention like operator. For the sake of technical clarity and simplicity in this work, we make perhaps the *simplest possible choices*. As a result, we *do not* claim that our network is optimally designed, but rather that the principles we develop in this work (compression, denoising, sparsification, unrolled optimization) can provide the backbone for far superior and more interpretable network architectures in the future on sundry tasks. As it is, with our straightforward, simple, and interpretable design, we still obtain meaningful conceptual results and very solid empirical performance.

We now give the derivation of the approximation alluded to in Section 2.3.

**Approximation 4.** *Let $\boldsymbol{Z}\in\mathbb{R}^{d\times N}$ have unit-norm columns, and $\boldsymbol{U}_{[K]}=(\boldsymbol{U}_1,\ldots,\boldsymbol{U}_K)$ such that each $\boldsymbol{U}_k\in\mathbb{R}^{d\times p}$ is an orthogonal matrix, the $(\boldsymbol{U}_k)_{k=1}^K$ are incoherent, and the columns of $\boldsymbol{Z}$ approximately lie on $\bigcup_{k=1}^K\operatorname{Span}(\boldsymbol{U}_k)$. Let $\gamma=\frac{p}{N\epsilon^2}$. Let $\kappa>0$. Then*

$$\boldsymbol{Z}-\kappa\nabla_{\boldsymbol{Z}}R^c(\boldsymbol{Z}\mid\boldsymbol{U}_{[K]})\approx(1-\kappa\gamma)\boldsymbol{Z}+\kappa\gamma\,\texttt{MSSA}(\boldsymbol{Z}|\boldsymbol{U}_{[K]}), \quad (61)$$

*where as in Section 2.3 we have*

$$\texttt{SSA}(\boldsymbol{Z}|\boldsymbol{U}_k)=(\boldsymbol{U}_k^*\boldsymbol{Z})\operatorname{softmax}((\boldsymbol{U}_k^*\boldsymbol{Z})^*(\boldsymbol{U}_k^*\boldsymbol{Z})), \quad (62)$$

$$\texttt{MSSA}(\boldsymbol{Z}|\boldsymbol{U}_{[K]})=\gamma\left[\boldsymbol{U}_1,\ldots,\boldsymbol{U}_K\right]\begin{bmatrix}\texttt{SSA}(\boldsymbol{Z}|\boldsymbol{U}_1)\\\vdots\\\texttt{SSA}(\boldsymbol{Z}|\boldsymbol{U}_K)\end{bmatrix}, \quad (63)$$

*where* $\mathrm{softmax}(\cdot)$ *is the softmax operator (applied to each column of an input matrix), i.e.,*

$$\mathrm{softmax}(\boldsymbol{v}) = \frac{1}{\sum_{i=1}^{n} e^{v_i}} \begin{bmatrix} e^{v_1} \\ \vdots \\ e^{v_n} \end{bmatrix}, \tag{64}$$

$$\mathrm{softmax}([\boldsymbol{v}_1, \ldots, \boldsymbol{v}_K]) = [\mathrm{softmax}(\boldsymbol{v}_1), \ldots, \mathrm{softmax}(\boldsymbol{v}_K)]. \tag{65}$$

*Proof.* According to (9), the gradient $\nabla_{\boldsymbol{Z}} R^c(\boldsymbol{Z}; \boldsymbol{U}_{[K]})$ is

$$\nabla_{\boldsymbol{Z}} R^c(\boldsymbol{Z}; \boldsymbol{U}_{[K]}) = \gamma \sum_{k=1}^{K} \boldsymbol{U}_k \boldsymbol{U}_k^* \boldsymbol{Z} \left( \boldsymbol{I} + \gamma (\boldsymbol{U}_k^* \boldsymbol{Z})^* (\boldsymbol{U}_k^* \boldsymbol{Z}) \right)^{-1}. \tag{66}$$

Notice that according to [55], the gradient is precisely the residual of a ridge regression for each (projected) token $\boldsymbol{U}_k^* \boldsymbol{z}_i$ using other projected tokens $\boldsymbol{U}_k^* \boldsymbol{z}_j$ as the regressors, hence being the residual of an auto-regression.

However, as we have seen in the work of ReduNet [55], computing the inverse $(\boldsymbol{I} + \gamma (\boldsymbol{U}_k^* \boldsymbol{Z})^* (\boldsymbol{U}_k^* \boldsymbol{Z}))^{-1}$ can be expensive. Hence for computational efficiency, we may approximate it with the first order term of its von Neumann expansion:

$$\nabla_{\boldsymbol{Z}} R^c(\boldsymbol{Z}; \boldsymbol{U}_{[K]}) = \gamma \sum_{k=1}^{K} \boldsymbol{U}_k \boldsymbol{U}_k^* \boldsymbol{Z} \left( \boldsymbol{I} + \gamma (\boldsymbol{U}_k^* \boldsymbol{Z})^* (\boldsymbol{U}_k^* \boldsymbol{Z}) \right)^{-1} \tag{67}$$

$$\approx \gamma \sum_{k=1}^{K} \boldsymbol{U}_k \boldsymbol{U}_k^* \boldsymbol{Z} \left( \boldsymbol{I} - \gamma (\boldsymbol{U}_k^* \boldsymbol{Z})^* (\boldsymbol{U}_k^* \boldsymbol{Z}) \right) \tag{68}$$

$$= \gamma \sum_{k=1}^{K} \boldsymbol{U}_k \left( \boldsymbol{U}_k^* \boldsymbol{Z} - \gamma \boldsymbol{U}_k^* \boldsymbol{Z} [(\boldsymbol{U}_k^* \boldsymbol{Z})^* (\boldsymbol{U}_k^* \boldsymbol{Z})] \right) \tag{69}$$

Notice that the term $(\boldsymbol{U}_k^* \boldsymbol{Z})^* (\boldsymbol{U}_k^* \boldsymbol{Z})$ is the auto-correlation among the projected tokens. As the tokens $\boldsymbol{Z}$ may be from different subspaces, we would prefer to use only tokens that belong to the *same* subspace to regress and compress themselves. Hence we may convert the above correlation term into a subspace-membership indicator with a softmax operation, whence (69) becomes

$$\nabla_{\boldsymbol{Z}} R^c(\boldsymbol{Z}; \boldsymbol{U}_{[K]}) \quad \approx \quad \gamma \sum_{k=1}^{K} \boldsymbol{U}_k \left( \boldsymbol{U}_k^* \boldsymbol{Z} - \gamma \boldsymbol{U}_k^* \boldsymbol{Z} [(\boldsymbol{U}_k^* \boldsymbol{Z})^* (\boldsymbol{U}_k^* \boldsymbol{Z})] \right) \tag{70}$$

$$\approx \quad \gamma \sum_{k=1}^{K} \boldsymbol{U}_k \boldsymbol{U}_k^* \boldsymbol{Z} - \gamma^2 \sum_{k=1}^{K} \boldsymbol{U}_k \left( \boldsymbol{U}_k^* \boldsymbol{Z} \, \mathrm{softmax}((\boldsymbol{U}_k^* \boldsymbol{Z})^* (\boldsymbol{U}_k^* \boldsymbol{Z})) \right) \tag{71}$$

Then, we can rewrite the above approximation to the gradient of $R^c$ as:

$$\nabla_{\boldsymbol{Z}} R^c(\boldsymbol{Z}; \boldsymbol{U}_{[K]}) \approx \gamma \sum_{k=1}^{K} \boldsymbol{U}_k \boldsymbol{U}_k^* \boldsymbol{Z} - \gamma^2 \sum_{k=1}^{K} \boldsymbol{U}_k \left( \boldsymbol{U}_k^* \boldsymbol{Z} \, \mathrm{softmax}((\boldsymbol{U}_k^* \boldsymbol{Z})^* (\boldsymbol{U}_k^* \boldsymbol{Z})) \right) \tag{72}$$

$$= \gamma \sum_{k=1}^{K} \boldsymbol{U}_k \boldsymbol{U}_k^* \boldsymbol{Z} - \gamma^2 \sum_{k=1}^{K} \boldsymbol{U}_k \, \mathtt{SSA}(\boldsymbol{Z} \mid \boldsymbol{U}_k) \tag{73}$$

$$= \underbrace{\left( \gamma \sum_{k=1}^{K} \boldsymbol{U}_k \boldsymbol{U}_k^* \right)}_{\approx \gamma \boldsymbol{Z}} \boldsymbol{Z} - \gamma^2 \, [\boldsymbol{U}_1, \cdots, \boldsymbol{U}_K] \begin{bmatrix} \mathtt{SSA}(\boldsymbol{Z} \mid \boldsymbol{U}_1) \\ \vdots \\ \mathtt{SSA}(\boldsymbol{Z} \mid \boldsymbol{U}_K) \end{bmatrix} \tag{74}$$

$$\approx \gamma \boldsymbol{Z} - \gamma^2 \, [\boldsymbol{U}_1, \cdots, \boldsymbol{U}_K] \begin{bmatrix} \mathtt{SSA}(\boldsymbol{Z} \mid \boldsymbol{U}_1) \\ \vdots \\ \mathtt{SSA}(\boldsymbol{Z} \mid \boldsymbol{U}_K) \end{bmatrix}. \tag{75}$$

Thus the gradient descent step with learning rate $\kappa > 0$ gives

$$\boldsymbol{Z} - \kappa\nabla_{\boldsymbol{Z}} R^c(\boldsymbol{Z} \mid \boldsymbol{U}_{[K]}) \approx (1 - \kappa\gamma)\boldsymbol{Z} + \kappa\gamma^2 \left[\boldsymbol{U}_1, \ldots, \boldsymbol{U}_K\right] \begin{bmatrix} \mathrm{SSA}(\boldsymbol{Z}|\boldsymbol{U}_1) \\ \vdots \\ \mathrm{SSA}(\boldsymbol{Z}|\boldsymbol{U}_K) \end{bmatrix}. \qquad (76)$$

$\square$

## A.3 Companion to Section 2.4

We again wish to re-iterate the core contribution of our approach in Section 2.4.

- Within the framework of alternating minimization of the sparse rate reduction objective, we show that the second alternating step — gradient descent on the overall coding rate plus a sparse regularization term — has heuristic connections to a particular LASSO optimization.
- We show that the unrolling of the proximal gradient step to solve this LASSO optimization resembles the MLP which immediately follows the self-attention layer within transformer blocks.

In the main text, our connection between the second step of the alternating minimization and the LASSO optimization was high-level and heuristic. In some sense, the choice to pose the minimization step as a LASSO was a *simple, reliable, and interpretable choice* which works well in practice, but is nonetheless not backed up by rigorous theoretical justification. In the following subsection, we provide a mathematical justification for a reformulation of the minimization step using a majorization-minimization framework. We further show that the associated unrolled optimization step bears a strong resemblance to the ISTA step. This confirms our earlier discussion — we took the *simplest possible choice* in designing CRATE, but by more rigorous derivation we can uncover alternative operators which nonetheless have the same conceptual function and may perform better in practice.

**Assumptions.** In this section, we present a rigorous optimization analysis of an incremental minimization approach to the objective (13). We will show that under two simplifying assumptions, namely

1. The columns of $\boldsymbol{Z}^{\ell+1/2}$ are normalized, in the sense that $\mathrm{diag}((\boldsymbol{Z}^{\ell+1/2})^*\boldsymbol{Z}^{\ell+1/2}) = \boldsymbol{1}$;[10]

2. We have $d \geq N$,[11] and the columns of $\boldsymbol{Z}^{\ell+1/2}$ are orthogonal, so that $(\boldsymbol{Z}^{\ell+1/2})^*\boldsymbol{Z}^{\ell+1/2} = \boldsymbol{I}$.[12]

the approach leads to an update iteration that is equal to a slightly simplified version of the ISTA block (17). We see this as a justification for our derivation in Section 2.4, which obtained the ISTA block by introducing an additional simplifying assumption on the distribution of the data at layer $\ell$.

**Analysis.** Following (16), we will consider the natural relaxation of the $\ell_0$ "norm" to the $\ell^1$ norm, and incorporate a nonnegativity constraint. Consider the objective

$$\varphi(\boldsymbol{Z}) = \lambda\|\boldsymbol{Z}\|_1 + \chi_{\{\boldsymbol{Z}\geq\boldsymbol{0}\}}(\boldsymbol{Z}) - \underbrace{\frac{1}{2}\log\det\left(\boldsymbol{I} + \alpha\boldsymbol{Z}^*\boldsymbol{Z}\right)}_{R(\boldsymbol{Z})}, \qquad (77)$$

where $\boldsymbol{Z} \in \mathbb{R}^{d\times N}$ and $\alpha = d/N\varepsilon^2$, and $\chi_{\{\boldsymbol{Z}\geq\boldsymbol{0}\}}$ denotes the characteristic function for the set of elementwise-nonnegative matrices $\boldsymbol{Z}$. As in Appendix A.2, we calculate

$$\nabla_{\boldsymbol{Z}} R(\boldsymbol{Z}) = \alpha\boldsymbol{Z}\left(\boldsymbol{I} + \alpha\boldsymbol{Z}^*\boldsymbol{Z}\right)^{-1}. \qquad (78)$$

---

[10]This is a natural assumption in transformer-type architectures such as CRATE due to the use of LayerNorm blocks—although these blocks (indeed, as we use them in CRATE) include trainable mean and scale offsets as well as an additional mean subtraction operation [64], they are initialized to have zero mean and unit norm, hence this assumption corresponds to an analysis of the network at its initialization.

[11]This assumption is without loss of generality, as we will see in the analysis below. The reason is that $\boldsymbol{Z}^*\boldsymbol{Z}$ and $\boldsymbol{Z}^*\boldsymbol{Z}$ have the same nonzero eigenvalues regardless of the shape of $\boldsymbol{Z}$, which implies that $\log\det(\boldsymbol{I} + \alpha\boldsymbol{Z}^*\boldsymbol{Z}) = \log\det(\boldsymbol{I} + \alpha\boldsymbol{Z}\boldsymbol{Z}^*)$. In particular, interpreting the norms appropriately (with a slight abuse of notation), we have $\varphi(\boldsymbol{Z}) = \varphi(\boldsymbol{Z}^*)$, so for the purposes of analysis we can always proceed as though $\boldsymbol{Z}$ is a tall matrix (as long as we do not use any special properties of $\alpha$ in our derivation).

[12]This assumption is strictly stronger than the previous one, and strictly stronger than an assumption of incoherence on the columns. It corresponds to the representation $\boldsymbol{Z}^{\ell+1/2}$ being non-collapsed, which we expect to hold at initialization due to the projections $\boldsymbol{U}_{[K]}$ being random.

We consider an incremental optimization scheme for the highly nonlinear and nonconvex objective $\varphi$. Following Section 2.3, we optimize locally at a "post-compression" iterate $\boldsymbol{Z}^{\ell+1/2}$. We follow the standard proximal majorize-minimize framework [70] for incremental/local optimization: this begins with the second-order Taylor expansion for the smooth part of $\varphi$ in a neighborhood of the current iterate $\boldsymbol{Z}^{\ell+1/2}$:

$$
\begin{aligned}
R(\boldsymbol{Z}) = R(\boldsymbol{Z}^{\ell+1/2}) &+ \left\langle \nabla_{\boldsymbol{Z}} R(\boldsymbol{Z}^{\ell+1/2}), \boldsymbol{Z} - \boldsymbol{Z}^{\ell+1/2} \right\rangle \\
&+ \int_0^1 (1-t) \left\langle \boldsymbol{Z} - \boldsymbol{Z}^{\ell+1/2}, \nabla^2 R(\boldsymbol{Z}_t) \left( \boldsymbol{Z} - \boldsymbol{Z}^{\ell+1/2} \right) \right\rangle \mathrm{d}t,
\end{aligned}
\tag{79}
$$

where for any $\boldsymbol{Z} \in \mathbb{R}^{d \times N}$, $\boldsymbol{Z}_t = t\boldsymbol{Z}^{\ell+1/2} + (1-t)\boldsymbol{Z}$. The proximal majorization-minimization approach alternates two steps to minimize $\varphi$:

1. First, use assumptions on $\boldsymbol{Z}^{\ell+1/2}$ to derive an upper bound on the operator norm of the Hessian $\nabla^2 R(\boldsymbol{Z})$ over the effective domain of the optimization problem. We will write $L$ for this (uniform) upper bound. This yields a quadratic upper bound for the smooth part of the objective $\varphi$.

2. Then, alternately minimize the *smooth part* of the quadratic upper bound as a function of $\boldsymbol{Z}$, and take a *proximal step* on the nonsmooth part. It can be shown [70] that corresponds to the iteration

$$
\boldsymbol{Z}^+ = \operatorname{prox}_{\frac{\lambda}{L}(\|\cdot\|_1 + \chi_{\{\boldsymbol{Z} \geq \boldsymbol{0}\}})} \left( \boldsymbol{Z} + \frac{1}{L} \nabla_{\boldsymbol{Z}} R(\boldsymbol{Z}) \right)
\tag{80}
$$

In the alternating minimization setting of this paper for optimizing (1), we only take one such step, starting at $\boldsymbol{Z}^{\ell+1/2}$.

We will instantiate this program below, showing quantitative error bounds related to our assumptions above as necessary. Rather than directly applying the iteration (80), we will derive it below under our aforementioned assumptions.

Starting at (79), our first task is to upper bound the quadratic residual. This corresponds to estimating

$$
\left\langle \boldsymbol{Z} - \boldsymbol{Z}^{\ell+1/2}, \nabla^2 R(\boldsymbol{Z}_t) \left( \boldsymbol{Z} - \boldsymbol{Z}^{\ell+1/2} \right) \right\rangle
\tag{81}
$$

$$
\leq \sup_{t \in [0,1]} \left\| \nabla^2 R(\boldsymbol{Z}_t) \right\|_{\ell^2 \to \ell^2} \left\| \boldsymbol{Z} - \boldsymbol{Z}^{\ell+1/2} \right\|_{\mathrm{F}}^2
\tag{82}
$$

with Cauchy-Schwarz. Using Lemma 5, we can estimate the operator norm term in the previous bound in terms of properties of $\boldsymbol{Z}^{\ell+1/2}$. We need to bound

$$
\alpha \sup_{\|\boldsymbol{\Delta}\|_{\mathrm{F}} \leq 1} \left\| \left( \boldsymbol{\Delta} - \alpha \boldsymbol{Z}_t (\boldsymbol{I} + \alpha \boldsymbol{Z}_t^* \boldsymbol{Z}_t)^{-1} (\boldsymbol{Z}_t^* \boldsymbol{\Delta} + \boldsymbol{\Delta}^* \boldsymbol{Z}_t) \right) (\boldsymbol{I} + \alpha \boldsymbol{Z}_t^* \boldsymbol{Z}_t)^{-1} \right\|_{\mathrm{F}},
\tag{83}
$$

and Lemma 6 gives that this term is no larger than $9\alpha/4$ for any $\boldsymbol{Z}$ and any $t$. With this estimate and (79), we have a quadratic upper bound for $-R(\boldsymbol{Z})$:

$$
-R(\boldsymbol{Z}) \leq -R(\boldsymbol{Z}^{\ell+1/2}) + \left\langle -\nabla_{\boldsymbol{Z}} R(\boldsymbol{Z}^{\ell+1/2}), \boldsymbol{Z} - \boldsymbol{Z}^{\ell+1/2} \right\rangle + \frac{9\alpha}{8} \left\| \boldsymbol{Z} - \boldsymbol{Z}^{\ell+1/2} \right\|_{\mathrm{F}}^2.
\tag{84}
$$

Meanwhile, by our assumptions above, we have

$$
-\nabla_{\boldsymbol{Z}} R(\boldsymbol{Z}^{\ell+1/2}) = -\alpha \boldsymbol{Z}^{\ell+1/2} (\boldsymbol{I} + \alpha \boldsymbol{I})^{-1} = -\frac{\alpha}{1+\alpha} \boldsymbol{Z}^{\ell+1/2}.
\tag{85}
$$

We now minimize the preceding quadratic upper bound as a function of $\boldsymbol{Z}$. Differentiating, the minimizer $\boldsymbol{Z}_{\mathrm{opt}}$ is calculated as

$$
\boldsymbol{Z}_{\mathrm{opt}} = \left( 1 + \frac{4}{9(1+\alpha)} \right) \boldsymbol{Z}^{\ell+1/2},
\tag{86}
$$

and it is well-known that the proximal operator of the sum of $\chi_{\{\boldsymbol{Z} \geq \boldsymbol{0}\}}$ and $\lambda \| \cdot \|_1$ is simply the one-sided soft-thresholding operator [70]

$$
\operatorname{prox}_{\chi_{\{\boldsymbol{Z} \geq \boldsymbol{0}\}} + \lambda \| \cdot \|_1} (\boldsymbol{Z}) = \max\{\boldsymbol{Z} - \lambda \boldsymbol{1}, \boldsymbol{0}\},
\tag{87}
$$

where the maximum is applied elementwise. As in Section 2.4, we may write this elementwise maximum simply as ReLU. Thus, one step of proximal majorization-minimization under our simplifying assumptions takes the form

$$\boldsymbol{Z}^{\ell+1} = \mathrm{ReLU}\left(\left(1 + \frac{4}{9(1+\alpha)}\right)\boldsymbol{Z}^{\ell+1/2} - \frac{4\lambda}{9\alpha}\mathbf{1}\right). \tag{88}$$

Finally, we point out one additional elaboration which introduces the dictionary $\boldsymbol{D}$ that appears in the ISTA block in Section 2.4. Notice that for any orthogonal $\boldsymbol{D}$, one has $R(\boldsymbol{DZ}) = R(\boldsymbol{Z})$ for every $\boldsymbol{Z}$. This symmetry implies equivariance properties of $\nabla_{\boldsymbol{Z}}R(\boldsymbol{Z})$ and $\nabla^2_{\boldsymbol{Z}}R(\boldsymbol{Z})$: for every $\boldsymbol{Z}$ and every $\boldsymbol{\Delta}$ and every orthogonal $\boldsymbol{D}$,

$$\boldsymbol{D}\nabla_{\boldsymbol{Z}}R(\boldsymbol{Z}) = \nabla_{\boldsymbol{Z}}R(\boldsymbol{DZ}), \tag{89}$$

$$\langle \boldsymbol{D\Delta}, \nabla^2_{\boldsymbol{Z}}R(\boldsymbol{Z})\,(\boldsymbol{D\Delta})\rangle = \langle \boldsymbol{\Delta}, \nabla^2_{\boldsymbol{Z}}R(\boldsymbol{DZ})\,(\boldsymbol{\Delta})\rangle. \tag{90}$$

Hence the quadratic Taylor expansion (79) can be written equivalently as

$$\begin{aligned}
R(\boldsymbol{Z}) = R(\boldsymbol{D}^*\boldsymbol{Z}^{\ell+1/2}) + \left\langle \nabla_{\boldsymbol{Z}}R(\boldsymbol{D}^*\boldsymbol{Z}^{\ell+1/2}), \boldsymbol{Z} - \boldsymbol{Z}^{\ell+1/2}\right\rangle \\
+ \int_0^1 (1-t)\left\langle \boldsymbol{Z} - \boldsymbol{Z}^{\ell+1/2}, \nabla^2 R(\boldsymbol{D}^*\boldsymbol{Z}_t)\left(\boldsymbol{Z} - \boldsymbol{Z}^{\ell+1/2}\right)\right\rangle \mathrm{d}t,
\end{aligned} \tag{91}$$

for any orthogonal $\boldsymbol{D}$. The significance of this is that we have obtained an expression equivalent to (79), but with $\boldsymbol{Z}^{\ell+1/2}$ replaced by $\boldsymbol{D}^*\boldsymbol{Z}^{\ell+1/2}$; moreover, because our approximation arguments above are not affected by left-multiplication of $\boldsymbol{Z}^{\ell+1/2}$ by an orthogonal matrix (this operation does not change the norms of the columns of $\boldsymbol{Z}^{\ell+1/2}$, or their correlations, and hence the matrix's incoherence), we can apply exactly the same line of reasoning above to obtain that an equivalent proximal majorization-minimization iteration is given by

$$\boldsymbol{Z}^{\ell+1} = \mathrm{ReLU}\left(\left(1 + \frac{4}{9(1+\alpha)}\right)\boldsymbol{D}^*\boldsymbol{Z}^{\ell+1/2} - \frac{4\lambda}{9\alpha}\mathbf{1}\right), \tag{92}$$

for any orthogonal dictionary $\boldsymbol{D}$. This gives an update quite similar to the ISTA block (17) in the case where the dictionary used in Section 2.4 is orthogonal, but without a skip connection.

We thus obtain a natural white-box version of this part of the architecture, along with the natural interpretation *that its purpose is to sparsify the compressed tokens $\boldsymbol{Z}^{\ell+1/2}$ in a (learnable) dictionary*, which accords with recent empirical studies [79].

**Other architectures?** As we mentioned at the start of this section, the preceding derivation is performed in the most elementary possible setting in order to demonstrate the majorization-minimization approach for layer design. More precise approximations or assumptions may lead to superior layer designs that better optimize the target objective (1) (and in particular (13)). We mention two here:

1. **Beyond exactly-incoherent features**: our derivations above assumed that the incoming representations $\boldsymbol{Z}^{\ell+1/2}$ were already maximal for the expansion term $R$ in (13). It is desirable to obtain a 'perturbative' derivation, which applies in cases where $\boldsymbol{Z}^{\ell+1/2}$ is not fully orthogonal, but instead near-orthogonal, in particular *incoherent* [70]. The derivations above can be adapted to this setting; the perturbation bounds become slightly more delicate, and the ultimate layer (92) changes to involve additional normalization.

2. **Beyond orthogonal dictionaries**: The symmetries of the expansion term $R$ in (13) may be followed to lead to a pair of dictionaries $\boldsymbol{D}$ and $\boldsymbol{D}'$ and an objective that sparsifies $\boldsymbol{DZD}'$. This type of transformation is suggestive of popular architectures that mix over tokens [54, 67], however we consider the simpler form $\boldsymbol{DZ}$ in this work. In addition, we have focused for simplicity on orthogonal dictionaries $\boldsymbol{D}$; as in the previous bullet, one may consider in a similar way dictionaries $\boldsymbol{D}$ which are complete and near-orthogonal. Adapting the derivation to *overcomplete dictionaries* is an interesting future direction that we expect to improve the scalability of CRATE; one avenue to achieve this could be increasing the number of projections $\boldsymbol{U}_{[K]}$ and their embedding dimensions.

### A.3.1 Auxiliary Lemmas

**Lemma 5.** *Consider the function*

$$R(\boldsymbol{Z}) = \frac{1}{2} \log \det \left(\boldsymbol{I} + \alpha \boldsymbol{Z}^* \boldsymbol{Z}\right), \tag{93}$$

*where $\alpha > 0$ is a constant. Then we have*

$$\nabla_{\boldsymbol{Z}} R(\boldsymbol{Z}) = \alpha \boldsymbol{Z} \left(\boldsymbol{I} + \alpha \boldsymbol{Z}^* \boldsymbol{Z}\right)^{-1}, \tag{94}$$

*and the Hessian operator $\nabla_{\boldsymbol{Z}}^2 R(\boldsymbol{Z}) \colon \mathbb{R}^{d \times N} \to \mathbb{R}^{d \times N}$ satisfies that for any $\boldsymbol{\Delta} \in \mathbb{R}^{d \times N}$,*

$$\nabla_{\boldsymbol{Z}}^2 R(\boldsymbol{Z}) (\boldsymbol{\Delta}) \tag{95}$$

$$= \alpha \boldsymbol{\Delta} \left(\boldsymbol{I} + \alpha \boldsymbol{Z}^* \boldsymbol{Z}\right)^{-1} - \alpha^2 \boldsymbol{Z} \left(\boldsymbol{I} + \alpha \boldsymbol{Z}^* \boldsymbol{Z}\right)^{-1} \left(\boldsymbol{Z}^* \boldsymbol{\Delta} + \boldsymbol{\Delta}^* \boldsymbol{Z}\right) \left(\boldsymbol{I} + \alpha \boldsymbol{Z}^* \boldsymbol{Z}\right)^{-1}. \tag{96}$$

*Proof.* The gradient calculation follows from [46], for example. For the Hessian, we use the usual approach to calculating derivatives: if $\boldsymbol{\Delta}$ is any matrix with the same shape as $\boldsymbol{Z}$ and $t > 0$,

$$\nabla_{\boldsymbol{Z}}^2 R(\boldsymbol{Z}) (\boldsymbol{\Delta}) = \left.\frac{\partial}{\partial t}\right|_{t=0} \left[t \mapsto \nabla_{\boldsymbol{Z}} R(\boldsymbol{Z} + t\boldsymbol{\Delta})\right], \tag{97}$$

valid since $R$ is smooth. We have

$$\nabla_{\boldsymbol{Z}} R(\boldsymbol{Z} + t\boldsymbol{\Delta})$$

$$= \alpha(\boldsymbol{Z} + t\boldsymbol{\Delta}) \left(\boldsymbol{I} + \alpha(\boldsymbol{Z} + t\boldsymbol{\Delta})^*(\boldsymbol{Z} + t\boldsymbol{\Delta})\right)^{-1}$$

$$= \alpha(\boldsymbol{Z} + t\boldsymbol{\Delta}) \left(\boldsymbol{I} + \alpha \boldsymbol{Z}^* \boldsymbol{Z} + \alpha t \left[\boldsymbol{Z}^* \boldsymbol{\Delta} + \boldsymbol{\Delta}^* \boldsymbol{Z} + t\boldsymbol{\Delta}^* \boldsymbol{\Delta}\right]\right)^{-1}$$

$$= \alpha(\boldsymbol{Z} + t\boldsymbol{\Delta}) \left(\boldsymbol{I} + \alpha t \left(\boldsymbol{I} + \alpha \boldsymbol{Z}^* \boldsymbol{Z}\right)^{-1} \left[\boldsymbol{Z}^* \boldsymbol{\Delta} + \boldsymbol{\Delta}^* \boldsymbol{Z} + t\boldsymbol{\Delta}^* \boldsymbol{\Delta}\right]\right)^{-1} \left(\boldsymbol{I} + \alpha \boldsymbol{Z}^* \boldsymbol{Z}\right)^{-1}$$

$$= \alpha(\boldsymbol{Z} + t\boldsymbol{\Delta}) \left(\sum_{k=0}^{\infty} (-\alpha t)^k \left(\left(\boldsymbol{I} + \alpha \boldsymbol{Z}^* \boldsymbol{Z}\right)^{-1} \left[\boldsymbol{Z}^* \boldsymbol{\Delta} + \boldsymbol{\Delta}^* \boldsymbol{Z} + t\boldsymbol{\Delta}^* \boldsymbol{\Delta}\right]\right)^k\right) \left(\boldsymbol{I} + \alpha \boldsymbol{Z}^* \boldsymbol{Z}\right)^{-1},$$

where in the fourth line we require that $t$ is sufficiently close to 0 in order to invoke the Neumann series. First, notice that the term involving $\boldsymbol{\Delta}^* \boldsymbol{\Delta}$ does not play a role in the final expression: after we differentiate with respect to $t$ and take a limit $t \to 0$, terms arising due to differentiation of $t \mapsto t\boldsymbol{\Delta}^* \boldsymbol{\Delta}$ go to zero, because whenever the summation index $k > 0$ we have a term $(-\alpha t)^k$ that goes to zero as $t \to 0$. We thus obtain with the product rule

$$\left.\frac{\partial}{\partial t}\right|_{t=0} \left[t \mapsto \nabla_{\boldsymbol{Z}} R(\boldsymbol{Z} + t\boldsymbol{\Delta})\right] \tag{98}$$

$$= \alpha \boldsymbol{\Delta} \left(\boldsymbol{I} + \alpha \boldsymbol{Z}^* \boldsymbol{Z}\right)^{-1} - \alpha^2 \boldsymbol{Z} \left(\boldsymbol{I} + \alpha \boldsymbol{Z}^* \boldsymbol{Z}\right)^{-1} \left(\boldsymbol{Z}^* \boldsymbol{\Delta} + \boldsymbol{\Delta}^* \boldsymbol{Z}\right) \left(\boldsymbol{I} + \alpha \boldsymbol{Z}^* \boldsymbol{Z}\right)^{-1}. \tag{99}$$

$\square$

**Lemma 6.** *One has*

$$\sup_{\|\boldsymbol{\Delta}\|_{\mathrm{F}} \leq 1} \left\|\left(\boldsymbol{\Delta} - \alpha \boldsymbol{Z}_t (\boldsymbol{I} + \alpha \boldsymbol{Z}_t^* \boldsymbol{Z}_t)^{-1} (\boldsymbol{Z}_t^* \boldsymbol{\Delta} + \boldsymbol{\Delta}^* \boldsymbol{Z}_t)\right) (\boldsymbol{I} + \alpha \boldsymbol{Z}_t^* \boldsymbol{Z}_t)^{-1}\right\|_{\mathrm{F}} \leq \frac{9}{4}. \tag{100}$$

*Proof.* Fix $\boldsymbol{\Delta}$ satisfying $\|\boldsymbol{\Delta}\|_{\mathrm{F}} \leq 1$. By the triangle inequality,

$$\left\|\left(\boldsymbol{\Delta} - \alpha \boldsymbol{Z}_t (\boldsymbol{I} + \alpha \boldsymbol{Z}_t^* \boldsymbol{Z}_t)^{-1} (\boldsymbol{Z}_t^* \boldsymbol{\Delta} + \boldsymbol{\Delta}^* \boldsymbol{Z}_t)\right) (\boldsymbol{I} + \alpha \boldsymbol{Z}_t^* \boldsymbol{Z}_t)^{-1}\right\|_{\mathrm{F}} \tag{101}$$

$$\leq \left\|\boldsymbol{\Delta} (\boldsymbol{I} + \alpha \boldsymbol{Z}_t^* \boldsymbol{Z}_t)^{-1}\right\|_{\mathrm{F}} + \alpha \left\|\boldsymbol{Z}_t (\boldsymbol{I} + \alpha \boldsymbol{Z}_t^* \boldsymbol{Z}_t)^{-1} (\boldsymbol{Z}_t^* \boldsymbol{\Delta} + \boldsymbol{\Delta}^* \boldsymbol{Z}_t) (\boldsymbol{I} + \alpha \boldsymbol{Z}_t^* \boldsymbol{Z}_t)^{-1}\right\|_{\mathrm{F}}. \tag{102}$$

For the first term, we note that

$$\left\|\boldsymbol{\Delta} (\boldsymbol{I} + \alpha \boldsymbol{Z}_t^* \boldsymbol{Z}_t)^{-1}\right\|_{\mathrm{F}} = \left\|\left((\boldsymbol{I} + \alpha \boldsymbol{Z}_t^* \boldsymbol{Z}_t)^{-1} \otimes \boldsymbol{I}\right) \mathrm{vec}(\boldsymbol{\Delta})\right\|_{\mathrm{F}}, \tag{103}$$

and since $(\boldsymbol{I} + \alpha \boldsymbol{Z}_t^* \boldsymbol{Z}_t)^{-1} \preceq \boldsymbol{I}$, we obtain from Cauchy-Schwarz[13]

$$\left\|\boldsymbol{\Delta} (\boldsymbol{I} + \alpha \boldsymbol{Z}_t^* \boldsymbol{Z}_t)^{-1}\right\|_{\mathrm{F}} \leq \|\boldsymbol{\Delta}\|_{\mathrm{F}}. \tag{104}$$

---

[13]Recall that the eigenvalues of a Kronecker product of symmetric matrices are the tensor product of the eigenvalues (with multiplicity).

We can use a similar idea to control the second term. We have from the triangle inequality

$$\left\| \boldsymbol{Z}_t(\boldsymbol{I} + \alpha \boldsymbol{Z}_t^* \boldsymbol{Z}_t)^{-1}(\boldsymbol{Z}_t^* \boldsymbol{\Delta} + \boldsymbol{\Delta}^* \boldsymbol{Z}_t)(\boldsymbol{I} + \alpha \boldsymbol{Z}_t^* \boldsymbol{Z}_t)^{-1} \right\|_{\mathrm{F}} \tag{105}$$

$$\leq \left\| \boldsymbol{Z}_t(\boldsymbol{I} + \alpha \boldsymbol{Z}_t^* \boldsymbol{Z}_t)^{-1} \boldsymbol{Z}_t^* \boldsymbol{\Delta}(\boldsymbol{I} + \alpha \boldsymbol{Z}_t^* \boldsymbol{Z}_t)^{-1} \right\|_{\mathrm{F}} \tag{106}$$

$$+ \left\| (\boldsymbol{I} + \alpha \boldsymbol{Z}_t^* \boldsymbol{Z}_t)^{-1} \boldsymbol{Z}_t^* \boldsymbol{\Delta}(\boldsymbol{I} + \alpha \boldsymbol{Z}_t^* \boldsymbol{Z}_t)^{-1} \boldsymbol{Z}_t^* \right\|_{\mathrm{F}}. \tag{107}$$

For the first term, we have

$$\left\| \boldsymbol{Z}_t(\boldsymbol{I} + \alpha \boldsymbol{Z}_t^* \boldsymbol{Z}_t)^{-1} \boldsymbol{Z}_t^* \boldsymbol{\Delta}(\boldsymbol{I} + \alpha \boldsymbol{Z}_t^* \boldsymbol{Z}_t)^{-1} \right\|_{\mathrm{F}} \tag{108}$$

$$= \left\| \left( (\boldsymbol{I} + \alpha \boldsymbol{Z}_t^* \boldsymbol{Z}_t)^{-1} \otimes \boldsymbol{Z}_t(\boldsymbol{I} + \alpha \boldsymbol{Z}_t^* \boldsymbol{Z}_t)^{-1} \boldsymbol{Z}_t^* \right) \mathrm{vec}(\boldsymbol{\Delta}) \right\|_{\mathrm{F}} \tag{109}$$

$$\leq \sigma_{\max} \left( (\boldsymbol{I} + \alpha \boldsymbol{Z}_t^* \boldsymbol{Z}_t)^{-1} \right) \sigma_{\max} \left( \boldsymbol{Z}_t(\boldsymbol{I} + \alpha \boldsymbol{Z}_t^* \boldsymbol{Z}_t)^{-1} \boldsymbol{Z}_t^* \right) \| \boldsymbol{\Delta} \|_{\mathrm{F}} \tag{110}$$

$$\leq \frac{1}{\alpha} \| \boldsymbol{\Delta} \|_{\mathrm{F}}. \tag{111}$$

The last estimate follows from a computation using the SVD of $\boldsymbol{Z}_t$. Meanwhile, we have for the second term by a similar argument (using the fact that the singular values of $\boldsymbol{A}$ and $\boldsymbol{A}^*$ are identical for any matrix $\boldsymbol{A}$)

$$\left\| (\boldsymbol{I} + \alpha \boldsymbol{Z}_t^* \boldsymbol{Z}_t)^{-1} \boldsymbol{Z}_t^* \boldsymbol{\Delta}(\boldsymbol{I} + \alpha \boldsymbol{Z}_t^* \boldsymbol{Z}_t)^{-1} \boldsymbol{Z}_t^* \right\|_{\mathrm{F}} \leq \sigma_{\max} \left( (\boldsymbol{I} + \alpha \boldsymbol{Z}_t^* \boldsymbol{Z}_t)^{-1} \boldsymbol{Z}_t^* \right)^2 \| \boldsymbol{\Delta} \|_{\mathrm{F}} \tag{112}$$

$$\leq \frac{1}{4\alpha} \| \boldsymbol{\Delta} \|_{\mathrm{F}}, \tag{113}$$

where once again the estimate follows from a computation involving the SVD of $\boldsymbol{Z}_t$ (together with the fact that the function $\sigma \mapsto \sigma/(1 + \alpha \sigma^2)$ is bounded on $\sigma \geq 0$ by $1/(2\sqrt{\alpha})$). Putting it together, we have obtained

$$\left\| \left( \boldsymbol{\Delta} - \alpha \boldsymbol{Z}_t(\boldsymbol{I} + \alpha \boldsymbol{Z}_t^* \boldsymbol{Z}_t)^{-1}(\boldsymbol{Z}_t^* \boldsymbol{\Delta} + \boldsymbol{\Delta}^* \boldsymbol{Z}_t) \right)(\boldsymbol{I} + \alpha \boldsymbol{Z}_t^* \boldsymbol{Z}_t)^{-1} \right\|_{\mathrm{F}} \leq \frac{9}{4} \| \boldsymbol{\Delta} \|_{\mathrm{F}}, \tag{114}$$

which gives the claim after taking suprema.

$\square$

# B  Additional Experiments and Details

In this section, we provide details about our experiments, and report the results of additional experiments that were not covered in the main text. CRATE takes arguably the most basic design choices possible, and so we do *not* attempt to directly compete with state-of-the-art performance from heavily engineered and empirically designed transformers. The results of our experiments are meant to convey a few core messages:

- *Despite not being engineered to compete with the state-of-the-art,* CRATE *performs strongly on large-scale real-world datasets*, including classification on ImageNet-1K. CRATE also achieves strong transfer learning performance.

- *Because our model is designed through unrolled optimization of a well-understood objective, each layer is interpretable*. In particular, we can analyze the performance of CRATE, as well as design network modifications, on a *layer-wise basis*. This is powered by an arguably unparalleled level of insight into the role of each operator in our network.

- *We make the simplest possible choices during the design of* CRATE, *but these can be changed easily while keeping the same framework*. We study a few modifications later in this section (Appendix B.4) and show that they do not significantly hurt empirical performance, but emphasize here that there is significant potential for improvement with different architecture choices (and in particular a different theoretical analysis).

## B.1  Implementation details

In this subsection, we provide more details for implementing CRATE on vision tasks.

### B.1.1  Architecture of CRATE

**Architectural modifications.**  Compared to the conceptual architecture proposed in Sections 2.5 and 3, we make the following change for the sake of implementation simplicity:

- In the compression step, replace the term $\frac{p}{N\epsilon^2}[\boldsymbol{U}_1, \ldots, \boldsymbol{U}_K]$ in the MSSA operator with another trainable parameter $\boldsymbol{W} \in \mathbb{R}^{d \times pK}$. Thus the MSSA block becomes

$$\texttt{MSSA}(\boldsymbol{Z} \mid \boldsymbol{U}_{[K]}, \boldsymbol{W}) \doteq \boldsymbol{W} \begin{bmatrix} \texttt{SSA}(\boldsymbol{Z} \mid \boldsymbol{U}_1) \\ \vdots \\ \texttt{SSA}(\boldsymbol{Z} \mid \boldsymbol{U}_K) \end{bmatrix}. \tag{115}$$

PyTorch **code for CRATE.** We provide PyTorch-style code for implementing our proposed network architecture. Algorithm 1 defines the overall architecture, Algorithm 2 and Algorithm 3 contain details for the transformer block, self-attention block (MSSA-block), and MLP block (ISTA-block).

### B.1.2  Training Setup

**Pre-training on ImageNet-1K.**  We apply the Lion optimizer [73] for pre-training both CRATE and ViT models. We configure the learning rate as $2.4 \times 10^{-4}$, weight decay as 0.5, and batch size as 2,048. We incorporate a warm-up strategy with a linear increase over 5 epochs, followed by training the models for a total of 150 epochs with cosine decay. For data augmentation, we only apply the standard techniques, random cropping and random horizontal flipping, on the ImageNet-1K dataset. We apply label smoothing with smoothing parameter $0.1$. One training epoch of CRATE$-Base$ takes around 240 seconds using 16 A100 40GB GPUs.

**Fine-tuning.**  We fine-tune our pre-trained CRATE and ViT models on the following target datasets: CIFAR10/CIFAR100 [10], Oxford Flowers-102 [7], Oxford-IIIT-Pets [16]. We also evaluate our pre-trained models on the commonly used ImageNet Real [36] benchmark. For each fine-tuning task, we use the AdamW optimizer [26]. We configure the learning rate as $5 \times 10^{-5}$, weight decay as 0.01, and batch size to be 512. To allow transfer learning, we first resize our input data to 224. For data augmentations, we also adopt several standard techniques: random cropping, random horizontal flipping, and random augmentation (with number of transformations $n = 2$ and magnitude of transformations $m = 14$).[14]

---

[14]https://github.com/huggingface/pytorch-image-models/blob/main/timm/data/auto_augment.py

**Algorithm 1:** PyTorch-style pseudocode for CRATENetwork

```
# Class ViT_dictionary definition
CRATE:
    # initialization
    def init(self, image_size, patch_size, num_classes, dim, depth, heads,
     mlp_dim, pool = 'cls', channels = 3, dim_head = 64, dropout = 0.,
     emb_dropout = 0.):
        # define patch, image dimensions and number of patches
        image_height, image_width = pair(image_size)
        patch_height, patch_width = pair(patch_size)
        num_patches = (image_height // patch_height) * (image_width //
         patch_width)
        patch_dim = channels * patch_height * patch_width

        # define patch embedding, positional embedding, dropout, and transformer
        self.to_patch_embedding = Sequential(Rearrange, LayerNorm(patch_dim),
         Linear(patch_dim, dim), LayerNorm(dim))
        self.pos_embedding = Parameter(random(1, num_patches + 1, dim))
        self.cls_token = Parameter(random(1, 1, dim))
        self.dropout = Dropout(emb_dropout)
        self.transformer = Transformer(dim, depth, heads, dim_head, mlp_dim,
         dropout)

        # define pooling, latent layer, and MLP head
        self.pool = pool
        self.to_latent = Identity()
        self.mlp_head = Sequential(LayerNorm(dim), Linear(dim, num_classes))

    # forward pass
    def forward(self, img):
        x = self.to_patch_embedding(img)
        b, n, _ = shape(x)
        cls_tokens = repeat(self.cls_token, '1 1 d -> b 1 d', b = b)
        x = concatenate((cls_tokens, x), dim=1)
        x += self.pos_embedding[:, :(n + 1)]
        x = self.dropout(x)
        x = self.transformer(x)
        x = mean(x, dim = 1) if self.pool == 'mean' else x[:, 0]
        x = self.to_latent(x)
        return self.mlp_head(x)
```

**Algorithm 2:** Pytorch Style Pseudocode for Transformer Block in CRATE

```
# Class Transformer definition
class Transformer:
    # initialization
    def init(self, dim, depth, heads, dim_head, mlp_dim, dropout = 0.):
        # define layers
        self.layers = []
        self.depth = depth
        for _ in range(depth):
            self.layers.append([LayerNorm(dim, Attention(dim, heads, dim_head,
             dropout))])
            self.layers.append([LayerNorm(dim, FeedForward(dim, mlp_dim,
             dropout))])

    # forward pass
    def forward(self, x):
        for attn, ff in self.layers:
            x_ = attn(x) + x
            x = ff(x_)
        return x
```

**Algorithm 3:** Pseudocode for Attention and FeedForward

```
# Class FeedForward definition
class FeedForward:
    # initialization
    def init(self, dim, hidden_dim, dropout = 0., step_size=0.1, lambd=0.1):
        self.weight = Parameter(Tensor(dim, dim))
        init.kaiming_uniform_(self.weight)
        self.step_size = step_size
        self.lambd = lambd
    # forward pass
    def forward(self, x):
        x1 = linear(x, self.weight, bias=None)
        grad_1 = linear(x1, self.weight.t(), bias=None)
        grad_2 = linear(x, self.weight.t(), bias=None)
        grad_update = self.step_size * (grad_2 - grad_1) - self.step_size *
         self.lambd
        output = relu(x + grad_update)
        return output
# Class Attention definition
class Attention:
    # initialization
    def init(self, dim, heads = 8, dim_head = 64, dropout = 0.):
        inner_dim = dim_head * heads
        project_out = not (heads == 1 and dim_head == dim)
        self.heads = heads
        self.scale = dim_head ** -0.5
        self.attend = Softmax(dim = -1)
        self.dropout = Dropout(dropout)
        self.qkv = Linear(dim, inner_dim, bias=False)
    self.to_out = Sequential(Linear(inner_dim, dim), Dropout(dropout)) if
     project_out else nn.Identity()
    # forward pass
    def forward(self, x):
        w = rearrange(self.qkv(x), 'b n (h d) -> b h n d', h = self.heads)
        dots = matmul(w, w.transpose(-1, -2)) * self.scale
        attn = self.attend(dots)
        attn = self.dropout(attn)
        out = matmul(attn, w)
        out = rearrange(out, 'b h n d -> b n (h d)')
        return self.to_out(out)
```

## B.2 Experimental Results

In this subsection, we provide additional experimental results on CRATE, including layer-wise measurements, visualizations, as well as ablation studies.

### B.2.1 Layer-wise Evaluation and Visualization

**Layer-wise evaluation of compression and sparsity.** Similar to Figure 3, we conduct the layer-wise evaluation of compression term and sparsity for CRATE-Tiny, CRATE-Base, and CRATE-Large. We observe similar behavior as mentioned in Section 3.1: both the compression term and the sparsity term improves as the layer index increases.

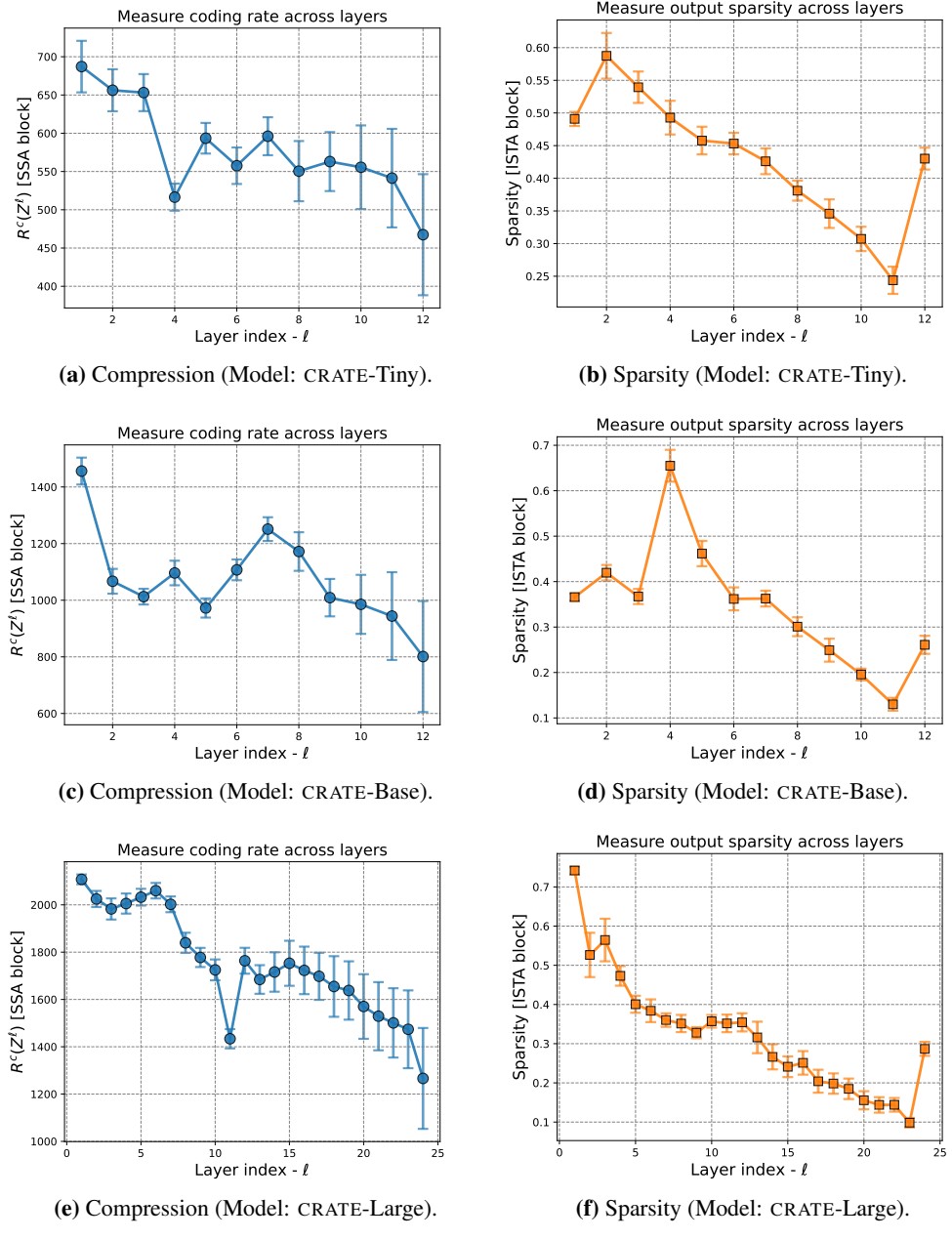

**(a)** Compression (Model: CRATE-Tiny).

**(b)** Sparsity (Model: CRATE-Tiny).

**(c)** Compression (Model: CRATE-Base).

**(d)** Sparsity (Model: CRATE-Base).

**(e)** Compression (Model: CRATE-Large).

**(f)** Sparsity (Model: CRATE-Large).

**Figure 5:** *Left*: The compression term $R^c(\mathbf{Z}^{\ell+1/2})$ of the MSSA outputs at different layers. *Right*: the sparsity of the ISTA output block, $\|\mathbf{Z}^{\ell+1}\|_0/(d \cdot N)$, at different layers.

**Visualizing layer-wise token representations.** In Figure 6, we visualize the token representations $\boldsymbol{Z}^\ell$ at different layers $\ell \in \{1, \ldots, 12\}$. We provide more results evaluated on other samples in Appendix B.2.2.

**Visualizing layer-wise subspaces in multi-head self-attention.** We provide the visualization of $\boldsymbol{U}^\ell_{[K]}$ in Figure 7.

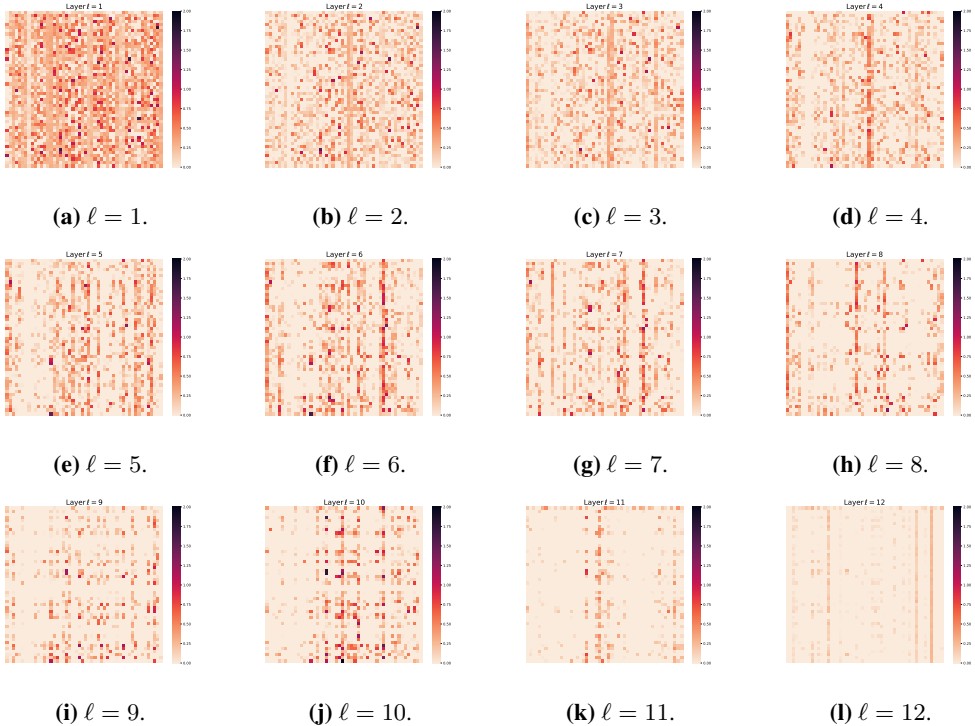

**(a)** $\ell = 1$.    **(b)** $\ell = 2$.    **(c)** $\ell = 3$.    **(d)** $\ell = 4$.

**(e)** $\ell = 5$.    **(f)** $\ell = 6$.    **(g)** $\ell = 7$.    **(h)** $\ell = 8$.

**(i)** $\ell = 9$.    **(j)** $\ell = 10$.    **(k)** $\ell = 11$.    **(l)** $\ell = 12$.

**Figure 6:** Visualizing layer-wise token $\boldsymbol{Z}^\ell$ representations at each layer $\ell$. To enhance the visual clarity, we randomly extract a $50 \times 50$ sub-matrix from $\boldsymbol{Z}^\ell$ for display purposes. (Model: CRATE-Tiny)

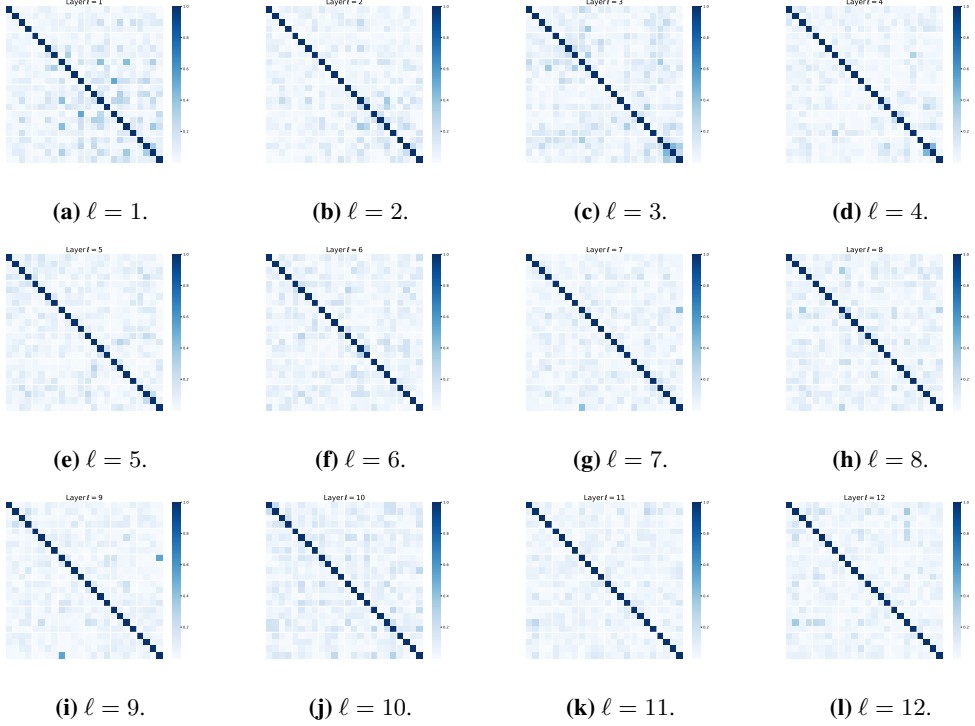

**Figure 7:** We visualize the $[\boldsymbol{U}_1^\ell, \ldots, \boldsymbol{U}_K^\ell]^* [\boldsymbol{U}_1^\ell, \ldots, \boldsymbol{U}_K^\ell] \in \mathbb{R}^{pK \times pK}$ at different layers. The $(i, j)$-th block in each sub-figure corresponds to $(\boldsymbol{U}_i^\ell)^* \boldsymbol{U}_j^\ell$ for $i, j \in [K]$ at a particular layer $\ell$. To enhance the visual clarity, for each subspace $\boldsymbol{U}_i$, we randomly pick 4 directions for display purposes. (Model: CRATE-Tiny)

### B.2.2 Additional Layer-wise Visualization

We provide more results of the layer-wise token representation visualization on other samples in Figure 8, Figure 9, Figure 10, and Figure 11 (Model: CRATE-Base).

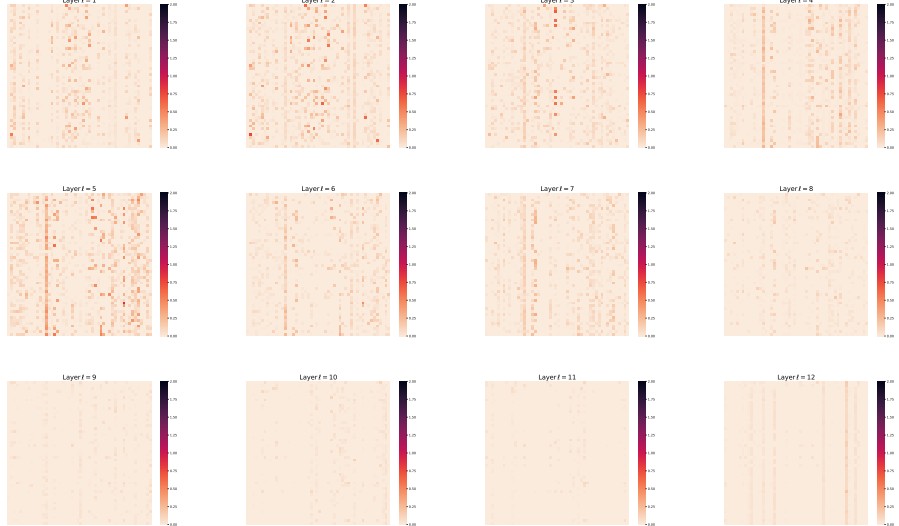

**Figure 8:** Visualizing layer-wise token $\boldsymbol{Z}^{\ell}$ representations at each layer $\ell$. To enhance the visual clarity, we randomly extract a 50×50 sub-matrix from $\boldsymbol{Z}^{\ell}$ for display purposes. (*Sample 1*)

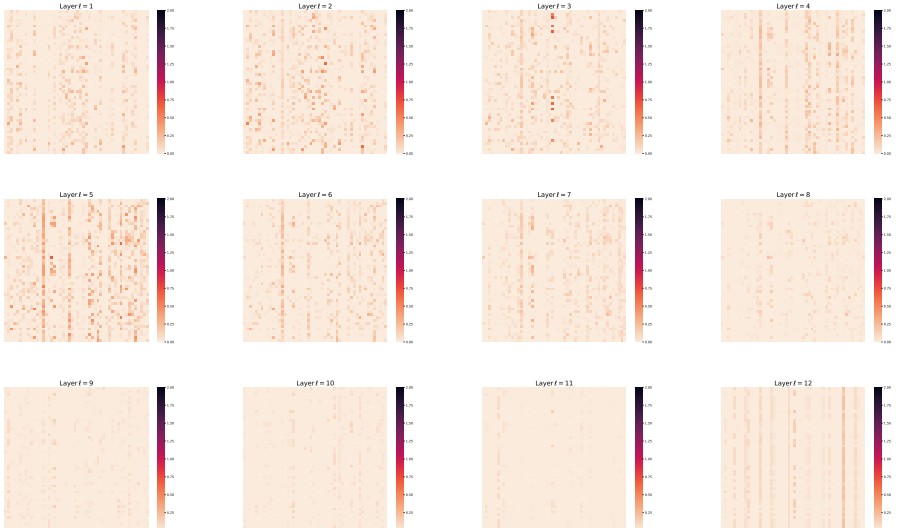

**Figure 9:** Visualizing layer-wise token $\boldsymbol{Z}^{\ell}$ representations at each layer $\ell$. To enhance the visual clarity, we randomly extract a 50×50 sub-matrix from $\boldsymbol{Z}^{\ell}$ for display purposes. (*Sample 2*)

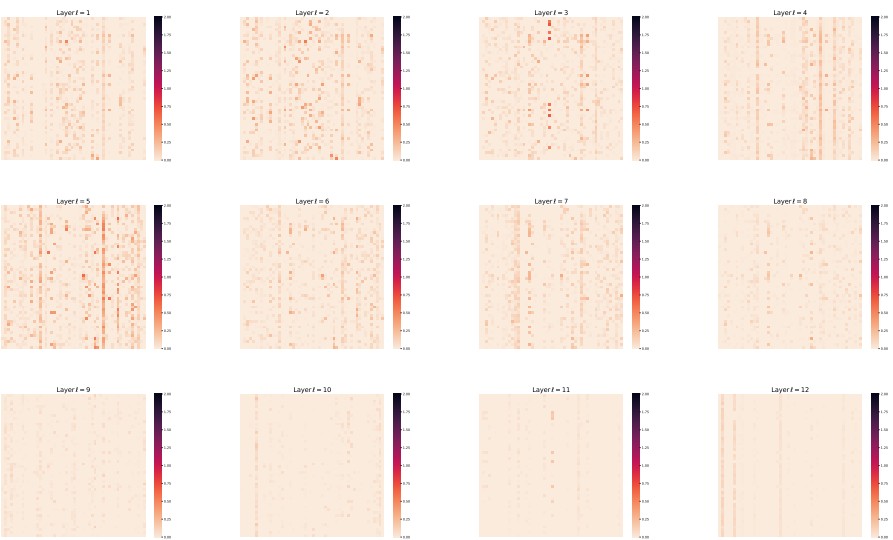

**Figure 10:** Visualizing layer-wise token $\boldsymbol{Z}^\ell$ representations at each layer $\ell$. To enhance the visual clarity, we randomly extract a $50{\times}50$ sub-matrix from $\boldsymbol{Z}^\ell$ for display purposes. (*Sample 3*)

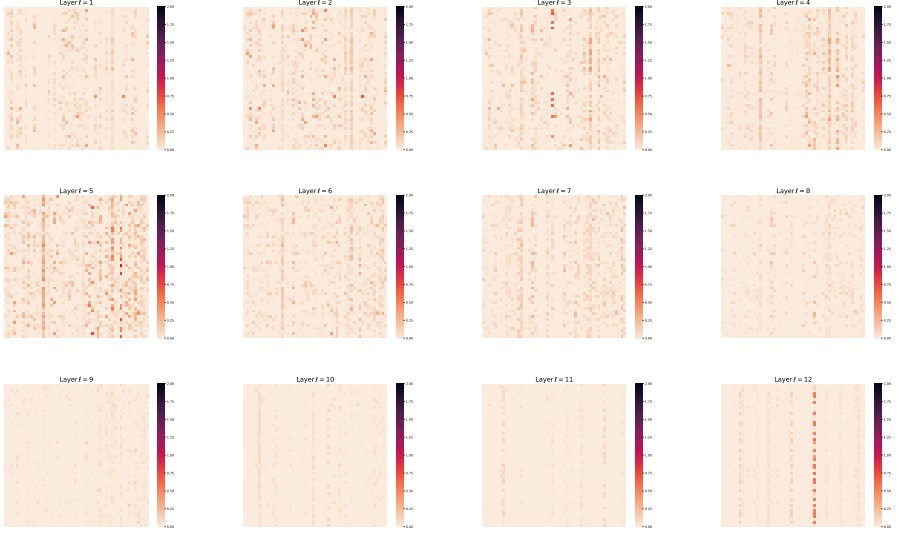

**Figure 11:** Visualizing layer-wise token $\boldsymbol{Z}^\ell$ representations at each layer $\ell$. To enhance the visual clarity, we randomly extract a $50{\times}50$ sub-matrix from $\boldsymbol{Z}^\ell$ for display purposes. (*Sample 4*)

## B.3 CRATE Ablation

**Hyperparameters of CRATE.** In Table 2, we present evaluation of CRATE trained with various parameters. More specifically, we investigate the effect of number of epochs, weight decay, learning rate, step size ($\eta$) and the regularization term ($\lambda$) in ISTA block. As shown in Table 2, CRATE demonstrates consistently satisfactory performance across a diverse range of hyperparameters.

**Table 2:** Top 1 accuracy of CRATE on various datasets with different architecture design variants when trained on ImageNet.

| Model | epoch | weight decay | lr | $\eta$ (ISTA) | $\lambda$ (ISTA) | ImageNet |
|---|---|---|---|---|---|---|
| CRATE-B | 150 (default) | 0.5 (default) | $2.4 \times 10^{-4}$ | 0.1 | 0.1 | 70.8 |
| CRATE-B | 150 | 0.5 | $2.4 \times 10^{-4}$ | *0.02* | 0.1 | 70.7 |
| CRATE-B | 150 | 0.5 | $2.4 \times 10^{-4}$ | *0.5* | 0.1 | 66.7 |
| CRATE-B | 150 | 0.5 | $2.4 \times 10^{-4}$ | 0.1 | *0.02* | 70.8 |
| CRATE-B | 150 | 0.5 | $2.4 \times 10^{-4}$ | 0.1 | *0.5* | 70.5 |
| CRATE-B | *90* | 0.5 | $2.4 \times 10^{-4}$ | 0.1 | 0.1 | 69.5 |
| CRATE-B | *300* | 0.5 | $2.4 \times 10^{-4}$ | 0.1 | 0.1 | 70.9 |
| CRATE-B | 150 | *1.0* | $2.4 \times 10^{-4}$ | 0.1 | 0.1 | 70.3 |
| CRATE-B | 150 | *0.05* | $2.4 \times 10^{-4}$ | 0.1 | 0.1 | 70.2 |
| CRATE-B | 150 | 0.5 | *$4.8 \times 10^{-4}$* | 0.1 | 0.1 | 70.2 |
| CRATE-B | 150 | 0.5 | *$1.2 \times 10^{-4}$* | 0.1 | 0.1 | 70.3 |

## B.4 Exploring Architecture Variants

In this section, we explore the two following alternative architectures. One architecture involves a modification to the attention mechanism, while the other involves a modification to the sparsification mechanism. Again, we re-emphasize that these choices, although principled, are entirely modular and the choices we make here still lead to very simple architectures. A more sophisticated analysis may lead to different, more complicated architectures that perform better in practice. The architectures we experiment with are:

- Compression-inspired attention mechanism: revert the change in (115). That is, the attention mechanism implements (11) and (12) directly.
- Majorization-minimization proximal step sparsification: instead of (17), implement (92).

We obtain the following classification results in Table 3. After conducting additional simplifications to the network architecture (i.e., imposing additional constraints to the network architecture design), we discover that CRATE maintains reasonable performance on ImageNet-1K.

**Table 3:** Top 1 accuracy of CRATE on various datasets with different architecture design variants when trained on ImageNet.

| Model | MSSA-block | ISTA-block | ImageNet |
|---|---|---|---|
| CRATE-B | default | default | 70.8 |
| CRATE-B | Eq. (11) and (12) | default | 63.3 |
| CRATE-B | default | Eq. (92) | 68.6 |

## B.5 Sparse Coding vs. Non-Negative Sparse Coding

In the main body, we used a non-negative sparse coding formulation (16) to obtain the ISTA block as an unrolled proximal gradient step in (17). Suppose that we hadn't done this, and instead directly computed an ISTA block using an unrolled proximal gradient step on (15). Such a block would give the following update rule:

$$\boldsymbol{Z}^{\ell+1} = S_{\lambda\eta}(\boldsymbol{Z}^{\ell+1/2} + \eta\boldsymbol{D}^*(\boldsymbol{Z}^{\ell+1/2} - \boldsymbol{D}\boldsymbol{Z}^{\ell+1/2})), \tag{116}$$

where $S_{\lambda\eta}$ is the soft-thresholding function

$$S_{\lambda\eta}(x) = \text{sgn}(x) \cdot (|x| - \lambda\eta)_+. \tag{117}$$

applied element-wise to its input matrix. The resulting architecture would be an alternative to CRATE; below, we discuss some empirical and theoretical similarities and differences between the two formulations and architectures.

**Empirical evaluation of soft-thresholding-based architecture.**  We summarize the results of CRATE with soft-thresholding activation in Table 4. We use $\lambda = 10$ in $S_{\lambda\eta}$ and set all other hyperparameters the same as in the original CRATE-Base evaluation on ImageNet-1K. We find that such a soft-thresholding model achieves slightly worse performance–a drop of 3.2% top-1 accuracy—compared to the default CRATE-Base (with ReLU activation).

**Table 4:** Top 1 accuracy of CRATE on ImageNet-1k with different architecture design variants. The soft-thresholding activation $S_{\lambda\eta}$ is defined in Eq. (117).

| Model | MSSA-block | ISTA-block | ImageNet | CIFAR 10* | CIFAR 100* |
|---|---|---|---|---|---|
| CRATE-B | default | ReLU activation (default) | 70.8% | 96.8% | 82.7% |
| CRATE-B | default | soft-thresholding activation | 67.6% | 96.0% | 76.8% |

**Potential theoretical justification for the performance differential.** Previous work [49] studied the phase collapse mechanism for understanding the non-linearities and the convolutional filters used in CNNs such as ResNet [23] on classification tasks. Specifically, they found that replacing the phase collapses with thresholding operators which enforce sparsity largely degrades the classification performance of CNNs. The effect of phase collapse analyzed in [49] is to better separate out the means of different classes within a classification task. This may account in part for the increase in classification accuracy reported in Table 4. On the other hand, we believe that the CRATE architecture will be applicable beyond just classification tasks. In CRATE, the purpose of the training process is to learn the local signal models at each layer (see e.g., Section 2.5). From this perspective, so long as the downstream training task requires semantically meaningful representations of the data distribution, the exact training configuration is of secondary importance. In particular, we may use self-supervised learning methods such as (masked) autoencoding to learn the signal models, whence there may not be any well-defined notion of class mean. In such cases, *a priori* we may expect both soft thresholding and nonnegative soft thresholding to perform comparably well. We leave the verification of this to future work.

**Theoretical justification for non-negative sparse coding.**  The sparse rate reduction formulation (1) does not include a non-negative constraint, and the token representations have marginal distribution equal to a mixture of zero-mean Gaussians (which are symmetric around $\mathbf{0}$). Below, we argue that the non-negative sparse rate reduction optimization and the regular sparse rate reduction optimization engender representations which are qualitatively similar in many ways, which confirms our conceptual understanding of how CRATE performs structured representation learning.

First, we formalize the non-negative sparse rate reduction optimization problem. Let $\chi$ be the characteristic function (with codomain $\{0, \infty\}$) of its input proposition. Then the non-negative analogue to (1) is

$$\max_{f \in \mathcal{F}}[\Delta R(\mathbf{Z}; \mathbf{U}_{[K]}) - \lambda \|\mathbf{Z}\|_0 - \chi(\mathbf{Z} \geq 0)] \qquad \text{where} \qquad \mathbf{Z} = f(\mathbf{X}). \qquad (118)$$

Although formal analysis of the optimal points of the sparse rate reduction maximization problem (1) or its nonnegative variant (118) is out of scope of this work, we see that the rate reduction maximization (i.e., $\max_{f \in \mathcal{F}}[\Delta R(\mathbf{Z}; \mathbf{U}_{[K]})]$ has optimal points characterized similarly to [46, Theorem A.6], namely that the representation of each distribution in the mixture is supported on a subspace with nearly isotropic covariance on this subspace, and the supporting subspaces are (nearly) orthogonal. Adding the sparsity term $\lambda \|\mathbf{Z}\|_0$ for some regularizer $\lambda$ would enforce the axis-alignment of the supporting subspaces; when adding in addition the nonnegativity term $\chi(\mathbf{Z} \geq 0)$, following through the proof of [46, Theorem A.6] suggests that the argument goes through with suitable modifications (in particular, considering the conclusions for the covariance rather than $\mathbf{Z}\mathbf{Z}^\top$). This sketch suggests that the statistical and geometric properties of the optimal representation remain the same when adding the non-negative constraint to the sparse rate reduction formulation. We leave a detailed proof to future work.

### B.6    Pre-training on ImageNet-21K

We inestigate a larger pre-training dataset for training CRATE. Specifically, we first pretrain on ImageNet-21K [9], which contains 14 million images, and then fine-tuned on ImageNet-1K. As

shown in Table 5, with the CRATE-Base model (22.80M parameters), we achieve 80.2% top-1 accuracy; this is comparable to ViT-Base ( 86M parameters, 83.9%) [40] with around 25% of the parameters. For pre-training on ImageNet-21K, we configure the learning rate to $1 \times 10^{-4}$, set the weight decay to 0.05, and use a batch size of 4,096. The total number of epochs is 90, with 10 warmup epochs. For fine-tuning on ImageNet-1K, we use the same set of parameters as described in Appendix B.1.2, with the exception of setting the learning rate to $5 \times 10^{-5}$ and having a total of 50 epochs.

**Table 5:** Top 1 accuracy of CRATE-Base and ViT-Base [40] on various datasets when both models are pre-trained on ImageNet-21k. We use the ViT-Base results from [40] as a basis for comparison.

| Model | # parameters | ImageNet | CIFAR 10 | CIFAR 100 |
|---|---|---|---|---|
| CRATE-Base | 22.80M | 80.2% | 98.3% | 88.3% |
| ViT-Base [40] | 86M | 83.9% | 99.0% | 91.7% |

## B.7 Evaluating $R^c$ and sparsity for ViT

We conduct experiments to evaluate the $R^c$ and sparsity of token representations from each layer of a pre-trained ViT-Base (downloaded from `https://github.com/huggingface/pytorch-image-models`). We summarize the results in Figure 12. We find that without our white-box design, the vanilla ViT does not optimize our proposed sparse rate reduction objective. This contrasts with the results shown in Figures 3 and 4 of the work, wherein we can observe that the compression term $R^c$ and sparsity value decrease layerwise for CRATE, in accordance with our theory.

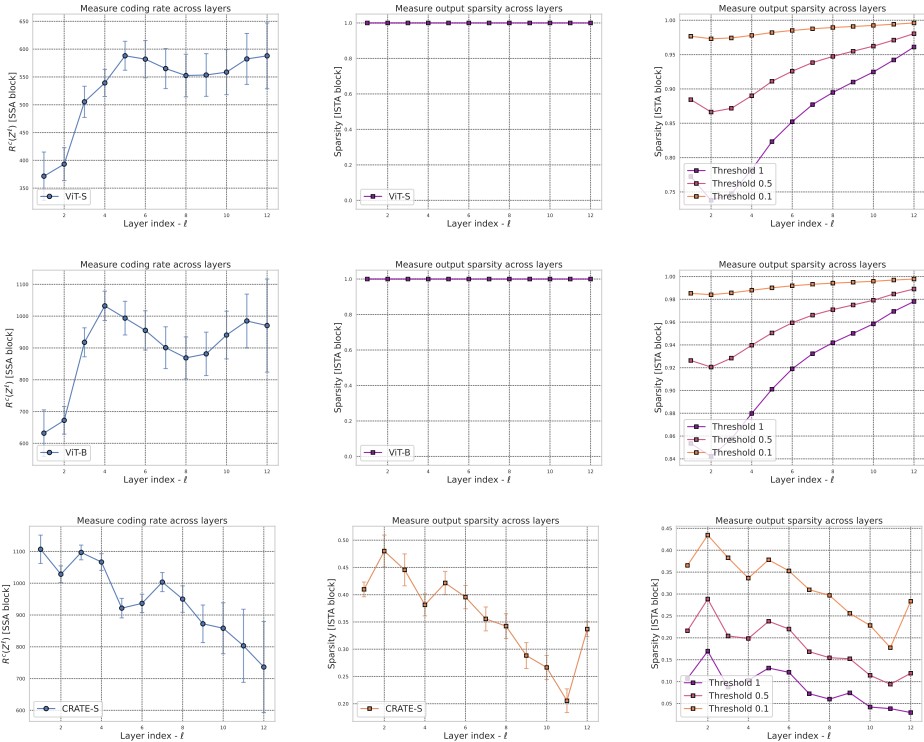

**Figure 12:** *Left*: The compression term $R^c(\boldsymbol{Z}^{\ell+1/2})$ of the multi-head self-attention outputs at different layers. *Middle*: The sparsity of outputs of the MLP block, $\|\boldsymbol{Z}^{\ell+1}\|_0/(d \cdot N)$, at different layers. *Right*: To get a more fine-grained understanding of the sparsity of MLP block outputs of ViT, we use three different thresholds $\tau \in \{1.0, 0.5, 0.1\}$ and measure $\sum_{i,j} \mathbf{1}\{|\boldsymbol{Z}_{i,j}^{\ell+1}| < \tau\}/(d \cdot N)$, where $\boldsymbol{Z}_{i,j}^{\ell+1}$ represents the $j$-th element in the $i$-th token representation. (First row model: ViT-Small; second row model: ViT-Base; third row model: our proposed CRATE-Small).

