# White-Box Transformers via Sparse Rate Reduction

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

(Z; U_{[K]}) = \sum_{k=1}^{K} R(U_k^* Z) = \frac{1}{2} \sum_{k=1}^{K} \log\det\left( I + \frac{p}{N\epsilon^2} (U_k^* Z)^* (U_k^* Z) \right). \tag{8}$$

We would like to compress (or denoise) the set of tokens against these subspaces by minimizing the coding rate. The gradient of $R^c(Z; U_{[K]})$ is

$$\nabla_Z R^c(Z; U_{[K]}) = \frac{p}{N\epsilon^2} \sum_{k=1}^{K} U_k U_k^* Z \left( I + \frac{p}{N\epsilon^2} (U_k^* Z)^* (U_k^* Z) \right)^{-1}. \tag{9}$$

The above expression approximates the residual of each projected token $U_k^* z_i$ regressed by other tokens $U_k^* z_j$ [54]. But, differently from [54], not all tokens in $Z$ are from the same subspace. Hence, to denoise each token with tokens from its own group, we can compute their similarity through an auto-correlation among the projected tokens as $(U_k^* Z)^* (U_k^* Z)$ and convert it to a distribution of membership with a softmax, namely $\mathrm{softmax}((U_k^* Z)^* (U_k^* Z))$. Then, as we show in Appendix A.2, if we only use similar tokens to regress and denoise each other, then a gradient step on the coding rate with learning rate $\kappa$ can be naturally approximated as follows:

$$Z^{\ell+1/2} = Z^\ell - \kappa \nabla_Z R^c(Z^\ell; U_{[K]}) \approx \left( 1 - \kappa \cdot \frac{p}{N\epsilon^2} \right) Z^\ell + \kappa \cdot \frac{p}{N\epsilon^2} \cdot \mathtt{MSSA}(Z^\ell \mid U_{[K]}), \tag{10}$$

where $\mathtt{MSSA}$ is defined through an SSA operator as:

$$\mathtt{SSA}(Z \mid U_k) \doteq (U_k^* Z) \, \mathrm{softmax}((U_k^* Z)^* (U_k^* Z)), \quad k \in [K], \tag{11}$$

---