# OpenReview forum: "White-Box Transformers via Sparse Rate Reduction"
_NeurIPS.cc/2023/Conference — NeurIPS 2023 poster_

### Official Review · Reviewer_tJjQ · 2023-06-22

**Soundness:** 3 good
**Presentation:** 3 good
**Contribution:** 4 excellent
**Rating:** 8
**Confidence:** 4

**Summary:**

This paper proposes a optimization target called sparse rate reduction, which is built on previous optimization target called rate reduction [49].
By unrolling the iterative optimization process of sparse rate reduction into neural layers, a transformer-like architecture can be obtained.
The derived white-box transformer-like architecture achieves similar performance with ViT.

**Strengths:**

Overall the manuscript is well written and related works are properly cited and discussed, very insightful work.
The idea is novel. This manuscript provides a significant extension to previous redunet [49]. The rate reduction is extended into sparse rate reduction, based on which transformer-like neural architectures can be derived by unrolling the iterative optimization process.
The manuscript provides new insights concerning several important aspects of modern neural networks, i.e., score function is shown to be connected to self-attention and rate reduction under idealized token distribution.
The results are promising, showing that the idea of white-box unrolling-based neural network design might be a possible alternative to current black-box design.



**Weaknesses:**

The results on imagenet are very promising, while it would be more convincing if the proposed white-box architecture can achieve SOTA performance under fair comparison.
Although overall the manuscript is well written, some sentences are too concise and a little confusing. I suggest the authors can go over the whole manuscript and improve the text for general readers.

1) L143 and footnote 4, I think the explaination here is too concise, the footnote confuses me. What is the separation and the mathematical roles? I cannot find related contents elsewhere.
2) L185 and footnote 6, can you explain more about the content in footnote 6? What is the rigorous math here?
3) I think the motivation of sparse coding (the L0) term in eq. 1 can be further clarified. I can understand the motivation of rate reduction [49], but I think it is not clear why we introduce sparse coding here. Further, is L0 norm the best implementation of sparse coding here? I hope the authors can discuss more about the designing principle of those optimization metrics or targets like eq. 1.

Following previous question, I think the optimization metric is closely related to specific task. In my opinion, rate reduction has a natural connection to classification problem. Can the authors comment on the connection between optimization metric and specific tasks? If we are considering object detection or more complex problem like image reconstruction, what is the general principle that can guide the design of optimization metrics?

**Questions:**

see above.

**Limitations:**

see above.

---

> ### Author Rebuttal · Authors · 2023-08-10
>
> Thank you for your comments, and your compliments on the insightfulness of the contribution, the novelty of the idea, the quality of the exposition, and the strength of the empirical results. Below we attempt to resolve the questions you posed.
>
> ## ImageNet results versus SOTA
>
> Thank you for the comment on the results. Please note that the main goal of this work is to not push the state-of-the-art, and as such the provided results incorporate minimal engineering compared to ViT or any more recent state-of-the-art model. With a more thorough engineering effort, and using the greater understanding of CRATE compared to ViT, one may potentially push the performance of CRATE-like models using the proposed framework beyond the state-of-the-art. Please see the Public Response for details on more experiments that push the performance of CRATE.
>
> ## Terseness of presentation
>
> Thank you for the suggestion. We will use some of the extra space afforded by the camera-ready version to improve the text, especially incorporating targeted feedback raised by you and other reviewers.
>
> ### Footnote 4
>
> There is some more elaboration on this point in Section 2.5, but we recapitulate it here. The mathematical roles that are separated in this dichotomy are the transformation of the data distribution towards the desired structured form (optimizers of the sparse rate reduction) in the forward pass (which we refer to as “optimization” in the footnote), and the learning of the parameters of these incremental transformations in the backward pass (which we refer to as “learning” in the footnote). Please let us know if this explanation has resolved the confusion; we will make appropriate revisions to the exposition in the paragraph at line 133 in the camera-ready version for improved clarity (merging the footnote into the text).
>
> ### Footnote 6
>
> The “rigorous math” referred to in this footnote refers to the mathematical theory of diffusion models, advanced primarily in [1], captured by two key concepts:
> - Given the score function for the noisy data distribution at a range of noise levels, a diffusion process that ‘follows the score function’ can be used to denoise the input noise distribution towards the data distribution.
> - In fact, it is not necessary to follow a (noisy) diffusion process to generate the data distribution, as there is a fully-deterministic “probability flow” ODE, which also involves the data distribution’s score function, that is mathematically equivalent to the score-following diffusion process. This means that even if one does not add noise when following the score function (as in CRATE), a suitable deterministic process can still transform the input distribution to the data distribution.
>
> We would be happy to provide more details if necessary in the discussion.
>
> ### $\ell^{0}$ norm explanation
>
> A brief answer is given in Section 2.1, but we reiterate it here. The rate reduction by itself is invariant to arbitrary rotations of the representations. To make the features more efficiently computable, and have human-interpretable structure (e.g., the principal components are just the standard basis vectors), we wish to align the representations with the coordinate axes, so that they become sparse. Thus, we penalize the $\ell^{0}$ norm, which counts the number of nonzero entries, hence the sparsity, of the representation $Z$. There are several "relaxations" of the $\ell^{0}$ norm which are efficiently optimizable – in Section 2.4 we picked arguably the most basic, namely the $\ell^{1}$-norm (c.f. LASSO regression, [2]). More choices are possible, and may even yield better performance, but studying this is left to future work.
>
> ## Relation between unrolled optimization objective and task; rate reduction and classification
>
> Indeed, the rate reduction framework as introduced in [3] was targeted to the specific case of classification – it uses class labels $\Pi$ to determine how to group the samples. However, in CRATE, our design of the network architecture (described in Section 2) is completely independent of any labels for the data samples: we derive the architecture from the goal of learning a representation that optimizes the sparse rate reduction objective, and this objective incorporates learnable "local signal models" where the labels $\Pi$ were used in [3]. In our experiments, we eventually learn the parameters of the CRATE model by training on a supervised classification task, but the white-box construction of the model makes the model applicable far beyond the setting of classification. Besides, we also mention that recently rate reduction/information gain type objectives have been used to produce representations which are useful in a variety of tasks, including self-supervised learning [4], generative modeling [5], image segmentation [6], etc.
>
> We hope that the points raised above resolve the doubts you have about this work, and constitute satisfactory responses to your questions. Please let us know if you have further questions or comments.
>
> [1]: Song, Yang et al., “Score-Based Generative Modeling through Stochastic Differential Equations,” in International Conference on Learning Representations, 2021.
>
> [2]: Wainwright, Martin J. High-dimensional statistics: A non-asymptotic viewpoint. Cambridge university press, 2019.
>
> [3]: Yu, Yaodong, et al., "Learning diverse and discriminative representations via the principle of maximal coding rate reduction." Advances in Neural Information Processing Systems 33 (2020).
>
> [4]: Ding, Tianjiao, et al., "Unsupervised manifold linearizing and clustering." arXiv preprint arXiv:2301.01805 (2023).
>
> [5]: Yang, Allen Y., et al., "Unsupervised segmentation of natural images via lossy data compression." Computer Vision and Image Understanding 110, no. 2 (2008): 212-225.
>
> [6]: Dai, Xili et al., "Ctrl: Closed-loop transcription to an LDR via minimaxing rate reduction." Entropy 24, no. 4 (2022): 456.

---

> > ### Comment · Reviewer_tJjQ · 2023-08-11
> >
> > I have read other reviews and author replies. My concerns are well solved, and thus I decide to increase my score

---

> > > ### Author Response · Authors · 2023-08-14
> > > **Response to Reviewer tJjQ**
> > >
> > > Thank you again for thoroughly reviewing our manuscript and response and raising your score. We are grateful for your valuable feedback on our work, which will no doubt improve it. Please let us know if you have any other questions or comments during the discussion period.

---

### Official Review · Reviewer_YxEu · 2023-07-01

**Soundness:** 3 good
**Presentation:** 3 good
**Contribution:** 3 good
**Rating:** 6
**Confidence:** 3

**Summary:**

The authors propose an interpretation of the transformer architecture wherein the component blocks may be interpreted as unrolled optimization steps:

* Multi-Head Self-Attention (MHSA) is said to be approximately the same as Multi-Head Subspace Self-Attention, which is an unrolled optimization of the following objective:
    * $ \sum_{k=1}^K R ( \mathbf{U}_k^* \mathbf{Z})$ (see eq. 8), where $R$ is an estimator for the "lossy coding rate" $R(\mathbf{Z}) = \frac{1}{2} \text{logdet} ( \mathbf{I} + \frac{d}{N \epsilon^2} \mathbf{Z} \mathbf{Z}^*  )$
* Multi-Layer Perceptrons are said to be approximately the same as Iterative Shrinkage-Thresholding Algorithms (ISTA) which are an unrolled optimization of the following objective:
    * $\lambda ||\mathbf{Z}||_1 + ||\mathbf{Z}^{l+1/2} - \mathbf{D} \mathbf{Z}||^2_F$ where $\mathbf{Z}$ are activations, $\mathbf{Z}^{l+1/2} = \mathbf{D} \mathbf{Z}^{l+1}$ and this is justified as being a relaxed LASSO objective that will sparsify the representation of $\mathbf{Z}$

The relationship between the architecture and the objective is quite involved and I cannot summarise it here.

As a consequence of constructing the network in this way, the authors are able to track both of these objectives as a way to gain insight how a network is operating; this is the "white-box" of the title.
Specifically, they track the sparse coding $R$ mentioned above and the sparsity at each layer in a network.
During training it is then possible to observe the sparse coding rate and sparsity both decrease with the layer index during training.
In addition, the network modifications required to match the theory do not appear to reduce performance on ImageNet compared to a similarly sized Vision Transformer.

**Strengths:**

Understanding the performance and popularity of self-attention in deep learning is a valuable goal and finding a theoretical motivation for this specific architecture could be very useful to anyone using or developing transformers.
If this goal is achieved by this paper then it is a significant work.

The strengths of this paper are therefore mainly found in Figures 3 and 4, which show the statistics $R^c$ and sparsity.
Figure 3 compares train versus validation while Figure 4 shows how these statistics change during training.
It is interesting to observe these values decreasing as the signal travels through the network as this approximately matches what is predicted by the theory.
These are the original empirical observations of the paper and they support the paper's claims.

The motivation and experimental results are both clearly stated.
Section 1 provides a reasoned argument for why learning representations that are interpretable would be valuable in current deep learning practice, and that it is lacking in current architectures, either diffusion or autoregressive models.
The results in Table 1 also address concerns that this method would reduce performance, it looks like performance is mostly maintained, which is promising.

**Weaknesses:**

Addressing following weaknesses would substantially improve the paper

* Legibility of the derivation: the derivations in Section 2 are dense and difficult to follow
    * The sparse rate coding function $R^c$ is key but is only introduced in Section 2.3, introducing it earlier and giving the reader an idea of how it relates to the distribution of $\mathbf{Z}$ would make it much easier to understand. For example, noting it's relationship to the entropy of a multivariate normal, how it will grow with the variance of $\mathbf{Z}$ etc
    * At the end of Section 2.2 the reader has just finished reading a derivation of "Self-Attention via Denoising Tokens Toward Multiple Subspaces" and then immediately afterwards is faced with "Self-Attention via Compressing Token Sets through Optimizing Rate Reduction". I am still confused as to which derivation I'm supposed to pay attention to for understanding the MSSA block. If both derive the same MSSA block then include only one and put the other in the Appendix, this will also help free up space for more experimental results
    * Do not rely on terminology from previous papers that are not common knowledge, for example if you agree that most readers will not understand the usage of "incoherent" without explanation, do not use it like that. Same for "linearized and compact".
    * On line 123 there's a reference to (8), which appears to be the ImageNet paper from 2008 and I could not find a formal definition of the lossy coding rate in it
    * The theory demonstrates how each layer could be performing an unrolled optimization step but it does not explain why this is beneficial to the overall problem of learning a function, such as predicting the class label using the entire network; in fact the target class labels are not present in the notation. I believe the entire network was optimized according to the cross-entropy loss, so where is that in Section 2?
* Additional experimental results would be worthwhile:
    * Table 1 does not contain results mentioned on lines 354-355 showing the scaling performance of ViT. Even if these results are from other work, including it with citation could be valuable in context
    * It is known that ViT architectures underperform on ImageNet versus larger datasets. Unfortunately, it would be valuable to see results training on Imagenet-21k, as in the original ViT paper, but this may be beyond the authors resource capacity
    * Computing $R^c$ and sparsity with depth in public pretrained models would also be useful, either to demonstrate that these models fail to minimize these implied objectives or to demonstrate that this is why transformers work
    * Comparing these statistics to activation norms would be a good comparison

**Questions:**

What experiment could you plan that would disprove the theory in Section 2? What architectural modification would not allow a low sparse coding rate or sparsity but would still allow similar performance when trained?

What do these results imply for transformer design, can you predict anything that is obviously incorrect or valuable just from these results?

Can you design a new layer that performs an additional form of unrolled optimization that is also useful?

Is there a regime of transformer operation that completely fails where this theory would provide a useful insight?

I think this direction of research is valuable and if I could understand precisely why the white-box observations in Figure 3 and 4 are extremely valuable I would increase my rating.

**Limitations:**

The limitation of trying to explain why a given deep learning architecture works is:

1. A method becomes popular because it is surprisingly effecitive
2. A theory is constructed to explain why it works
3. The theory demands some small change to the network which only slightly reduces performance
4. The theory does not immediately permit an improvement to the method that is valuable

I think this is why Figures 3 and 4 are key. If the statistics shown in Figures 3 and 4 were extremely valuable to understanding transformer training dynamics then it would be obvious that this work is significant.
It could be that they are, but I do not see this argument made clearly enough in the paper.

Alternatively, maybe there is some architectural or training improvement that this theory implies that would be extremely valuable.

---

> ### Author Rebuttal · Authors · 2023-08-10
>
> Thank you for your thoughtful review. Below, in response to the issues you have raised, we briefly reiterate how we view our work’s core motivations and contributions, then follow with precise responses to specific points raised in your review. Unfortunately due to space constraints we cannot answer all points in full detail, but we are glad to answer any follow-ups in the discussion phase.
>
> ## General Comments
>
> Your review suggests that our work’s goals and contributions are wrapped up in providing an interpretation of the popular transformer architecture, and as a byproduct obtaining implications for the design of transformer/self-attention architectures. We agree that this is an excellent path to achieving high-impact research in this area. However, we do not believe it is the only way. We would like to reiterate that the goal of our work *is not to directly explain the existing transformer, but to introduce a new white-box alternative whose operational characteristics are both different and transparent in a useful way.* As we describe in Section 2 of the submission, the model we propose is designed around the principle of learning a representation for nonlinear, multimodal data by incrementally transforming it to a standardized form. Conceptually, we derive our architecture to achieve this goal following an unrolled optimization perspective, leading to a derivation where we understand the role played by each parameter (i.e., a “white-box architecture”). Experimentally, we demonstrate that for a supervised classification task, the characteristics of the learned model agree with its white-box design, and performance is not sacrificed much either, in terms of both accuracy and scaling. Crucially, experiments on pretrained ViTs do not show the same characteristics as our white-box model, suggesting our white-box architecture has novel and highly interpretable characteristics, some of which are visualized in the Appendix (see Figures 6 to 11), that will be useful for network design and analysis beyond classification.
>
> ## Responses to Individual Points
>
> ### Results training on Imagenet-21k
>
> Please see the discussion in the general response.
>
> ### Computing $R^c$ and sparsity in public pretrained models vs CRATE (the value of Figures 3 and 4)
>
> Thank you for your insightful suggestion. We have conducted new experiments to evaluate the $R^c$ and sparsity of token representations from each layer of a pre-trained ViT-Base (downloaded from the `timm` GitHub repo). We summarize the results in Figure 1 of the uploaded rebuttal .pdf file. We find that without our white-box design, the vanilla ViT does not optimize our proposed sparse rate reduction objective. This contrasts with the results shown in Figures 3 and 4 of the work, wherein we see that the $R^c$ and sparsity value decrease layerwise for CRATE, in accordance with our theory. Since CRATE is an architectural modification of the vanilla ViT, we think this presents a compelling answer to your questions about an architectural modification that achieves similar performance without the same white-box characteristics, as well as a compelling experiment that could falsify our theory in Section 2 (and fails to do so). We will add these results and this comparison to Section 3 of our main body in our camera-ready version.
>
> Overall, this shows that the CRATE model, while looking and performing quite similar to a transformer in experiments, is fully interpretable through the perspective of sparse rate reduction, in a manner that is distinct from the black-box ViT. This is a significant advance in the development of layer-wise interpretable networks -- this is the first, to our knowledge, to achieve performance comparable with standard models such as ViT on ImageNet.
>
> ### Implications of our results for transformer design
>
> Both our theoretical and empirical results suggest that it is possible to design transformer-like architectures from the principle of unrolled optimization. One valuable design choice from our work is that QKV heads in attention may not be necessary, and this could help reduce the number of model parameters (indeed, our design, instantiated in CRATE-Base, has about 25% of the parameters of ViT-Base while demonstrating comparable performance) and make the whole network more efficient. Furthermore, both the MSSA block and the ISTA block are simpler and more interpretable than the existing multi-head attention block and MLP block, and we hope such a minimal and functional network architecture could further improve our understanding of transformer-like architectures.
>
> ### Unrolled optimization, class labels in the derivation
>
> Our CRATE architecture is derived from the goal of transforming the data distribution to a structured form, rather than a specific task. This is why the conceptual derivation in Section 2 does not include any discussion of labels: although labeled samples may be used to learn the parameters of the CRATE model, so long as the downstream training task requires semantically meaningful representations of the data distribution, the exact training configuration is of secondary importance. Intuitively, once we could identify the low-dimensional representations of the high-dimensional data (ie, via optimizing the sparse rate reduction), such representations are effective for classification problems.
>
> We believe that the unrolled optimization perspective is a major potential advantage of our white-box framework: in settings where, for example, prior information is available about the data distribution (for example, in medical imaging applications or other problems with scientific data), our white-box design allows these structures to be incorporated in a transparent way.
>
> ### Comments on the presentation
>
> Thank you for your helpful comments. We will incorporate your suggestions in the camera-ready version, using the extra space.
>
> We hope that the points raised above help clarify the significance of our contributions.

---

> > ### Comment · Reviewer_YxEu · 2023-08-11
> > **Falsification Results**
> >
> > In reply to this comment:
> >
> > > We find that without our white-box design, the vanilla ViT does not optimize our proposed sparse rate reduction objective. This contrasts with the results shown in Figures 3 and 4 of the work, wherein we see that the $R^c$ and sparsity value decrease layerwise for CRATE, in accordance with our theory. Since CRATE is an architectural modification of the vanilla ViT, we think this presents a compelling answer to your questions about an architectural modification that achieves similar performance without the same white-box characteristics, as well as a compelling experiment that could falsify our theory in Section 2 (and fails to do so).
> >
> > I don't understand the results presented in Figure 1 of your pdf rebuttal:
> >
> > 1. The sparsity of a vanilla ViT should not be 1.0, in that case no activations would ever be sufficiently negative entering the GeLU activation functions, which would mean the model is failing to learn nonlinear functions. Specifically, [other papers][lazy] have demonstrated activation sparsity of 6.3% in vision transformers recently.
> > 2. While the coding rate decreases for CRATE as the layer index increases, I don't see this as being a very conclusive result, it only decreases 27% through the entire network, while ViT-B decreases 15% from layers 4 to 8.
> >
> > I apologise for the confusion because I did not state what I thought your hypothesis was so that makes it difficult to talk about falsification. The hypothesis I had in mind was, "sparse rate reduction is sufficient for learning useful functions in deep networks".
> >
> > In that case, I don't see this as falsifying anything, if sparse rate reduction is critical for your network to learn then what you need to is ablate those capabilities from the model and show that it is no longer able to learn. In fact, if I accept the statement above that the ViT has no capability for sparse rate reduction and is still able to outperform CRATE then I can only conclude the sparse rate reduction in CRATE is irrelevant. I would attribute the performance to the architectural similarity to a transformer and the effectiveness of contemporary minibatch SGD on a cross-entropy objective.
> >
> > The derivation is interesting and the results matching contemporary networks are promising but I don't see what the "white box" buys you. Specifically:
> >
> > 1. It's not significantly correlated with performance, your results comparing to ViT demonstrate this
> > 2. It doesn't provide a significant benefit in architectural design beyond prescribing a block that is similar to a transformer
> > 3. It doesn't change how models are trained in any significant way (unless the paper fails to mention that this model converges in a significantly different way to a transformer)
> > 4. It encourages sparsity but the computation benefit isn't explored in the paper, nor is it demonstrated that this sparsity is significantly lower than it is now known to be in transformers. For example, ["The Lazy Neuron Phenomenon"][lazy] demonstrates 6.3% nonzero entries in ViT-B16
> > 5. Interpretability is a fuzzy concept, but I don't see any experiments in the paper aimed at interpreting what the network is doing based on the sparse rate reduction metrics, the experiments simply observe the metrics decrease while the network learns
> >
> > [lazy]: https://arxiv.org/abs/2210.06313

---

> > > ### Author Response · Authors · 2023-08-12
> > > **Discussion with Reviewer YxEu (Part 1)**
> > >
> > > We are grateful for you engaging with our rebuttal further, and for your critical perspective on the work, which will no doubt improve it. Thank you also for pointing out that **‘[t]he derivation is interesting and the results matching contemporary networks are promising.’**
> > >
> > > ### Interpreting the requested experiments on Figure 3/4 on public ViTs
> > >
> > > Our results are an accurate reflection of the experiment that you suggested (e.g., computing $R^c$ and sparsity as a function of depth). The last two columns of the first two rows of Figure 1 in the .pdf rebuttal evaluate the sparsity of the tokens after the second block of each transformer layer, $z\_{\\ell} = \\mathrm{MLP}(\\mathrm{LN}(z^{\\prime}\_{\ell})) + z^{\\prime}\_{\\ell}$, as defined in Eq. (3) of the ViT paper [1]), which make this a consistent comparison with how we evaluate the sparsity of the second block (i.e., ISTA block) of each CRATE layer. We applied the original weights and architecture of the public pre-trained ViT model from the `timm` package. Compared with the other paper [2] you mentioned, [2] evaluated the sparsity of the hidden layer output of the MLP, which is different from ours. Meanwhile, [2] replaced the GeLU activation with ReLU activation in the MLP layer. We did not find public checkpoints for the models in [2].
> > >
> > > Similarly for the $R^c$ results, although we can agree that a subjective interpretation is possible, we think it is unambiguous that in a network derived from unrolled optimization, it “optimizes” the objective if the objective trends in the appropriate direction on average. The result shows that for the compression part of the objective, this is true of CRATE, and not true of ViT; the CRATE-S model’s $R^c$ term is reduced by about 30% over the course of forward propagation, whereas the analogous terms for ViT-S and ViT-B increase by about 50%. If you would like to see additional comparisons here, we are happy to run them and report the results.
> > >
> > > **We want to state clearly that the results we are reporting are accurate**; we are only providing our interpretations of the results of the experiments you suggested. But let us emphasize that evaluating the $R^c$ and sparsity metrics in the parts of the network we have is most reasonable for CRATE precisely because _we have designed the network to learn a representation that has these characteristics_. Your suggested experiments demonstrate that the token embeddings of the ViT – analogous to what we evaluated in Figures 3 and 4 – do not have these same properties. However, this does _not_ imply that the ViT does not learn low-dimensional or parsimonious (e.g., compressed and sparse) representations of the data. Rather, it implies that the ViT’s learned representations are less accessible, and thus harder to evaluate, due to its parameter-redundant black-box design. This is a key benefit of our derivation, and the simplified white-box architecture of CRATE: the places where the representations are transformed to standard forms (axis-aligned, hence sparse, orthogonal subspaces) are completely exposed to the network architect, removing any ambiguity in measuring these quantities. We believe these insights present an excellent opportunity for follow-up work to better understand the ViT, as well, but this is firmly out of scope of the present submission.
> > >
> > > [1]: A. Dosovitskiy et al., “An Image is Worth 16x16 Words: Transformers for Image Recognition at Scale,” in International Conference on Learning Representations, 2021.
> > >
> > > [2]: Z. Li et al., “The Lazy Neuron Phenomenon: On Emergence of Activation Sparsity in Transformers,” in The Eleventh International Conference on Learning Representations, 2023.

---

> > > ### Author Response · Authors · 2023-08-12
> > > **Discussion with Reviewer YxEu (Part 2)**
> > >
> > > ### Our work’s hypotheses
> > >
> > > Thank you for clearly stating your thinking on this point. We think you may be misunderstanding our work’s principal experimental hypothesis: "sparse rate reduction is sufficient for learning useful functions in deep networks" seems to us to be a misinterpretation. Let us also mention in this connection that your assertion
> > >
> > > > if I accept the statement above that the ViT has no capability for sparse rate reduction and is still able to outperform CRATE then I can only conclude the sparse rate reduction in CRATE is irrelevant.
> > >
> > > does not seem to be logically sound: **we have not claimed anywhere that using the sparse rate reduction is necessary to construct high performance deep models**. In fact, its role in our derivation is quite the opposite: its use in the design of the architecture promotes the learning of mathematically-interpretable representations of the data in the network. Please see the discussion of “what white-box buys us” for more on this point.
> > >
> > > To clearly state our central hypothesis, let us reiterate our primary motivations, which were written in Section 2.1. Our goal is to design a network architecture that transforms the data to a mixture of nearly-orthogonal axis-aligned subspaces, the optimizers of the sparse rate reduction [3]. We thus obtain our architecture from unrolled optimization on the sparse rate reduction, then learn its parameters with backpropagation, since the structure of the data distribution is unknown. In particular, since we are learning these parameters, there is no guarantee that the resulting network will optimize a sparse rate reduction objective for the data distribution. This leads precisely to **our main hypothesis, that *it is possible to train a transformer-like architecture (i.e., CRATE) to simultaneously achieve high accuracy at scale and optimize the sparse rate reduction***. The results in the left panel of Figure 4 demonstrate that at random initialization, the CRATE-Small model does not optimize the sparse rate reduction for the data distribution – only through learning does the network optimize the sparse rate reduction for the data distribution.
> > >
> > > Note that in this line of reasoning, the *goal* is to obtain a useful representation of the data. We have argued in the introduction of the submission why this goal is valuable and a central ‘grail’ for learning. **The goal is not only to obtain a network with high performance; the goal is to obtain a white-box network which learns useful representations.**
> > >
> > > [3]: Y. Yu et al., "Learning diverse and discriminative representations via the principle of maximal coding rate reduction." Advances in Neural Information Processing Systems 33 (2020).

---

> > > ### Author Response · Authors · 2023-08-12
> > > **Discussion with Reviewer YxEu (Part 3)**
> > >
> > > ### What white-box buys us
> > >
> > > We would like to push back on your characterization of our white-box model. In our work, a “white-box model” can be thought of as a model whose architecture and parameters are derived mathematically from first principles, in a manner where the data distribution plays a central role. In this view, your first assertion is self-evident: the most natural white-box model for representation learning would be sparse coding of the data, possibly in a learnable signal dictionary, which gives a mathematically-interpretable and practically-robust model with performance that unfortunately cannot match that of modern deep learning architectures. Our contribution is to present a white-box derivation of a transformer-like architecture that is simultaneously highly performant. We truly believe there is significant novelty in this contribution: despite notable efforts from the theoretical community to suggest possible interpretations for the self-attention operation in transformers (e.g., summarized in [4]), a holistic and practically-verified interpretation for an entire transformer-like block (i.e., both the self-attention operation and MLP) has not been proposed before our work. In response to your second point, we would like to reiterate that as we wrote in the rebuttal, there are in fact **concrete practical implications** of our work for standard transformers: specifically, that **the QKV matrices in self-attention layers of ViT are redundant, and can be combined to save almost a factor of 4 in the overall parameter count**, with only a minor performance hit that can surely be reduced further with additional engineering work. Regarding your remaining points, we already mentioned the interpretability experiments we conducted in the submission in our rebuttal “**General Comments**” (e.g., visualizing the learned dictionaries and subspaces of CRATE in Figures 6 through 11 in the appendix); we think studying the computational benefit of sparsity based on our results is an interesting direction for future work, but firmly out-of-scope for the present work.
> > >
> > > We appreciate from your response here and to `AAKz` that you harbor some skepticism of research on the “model-centric” understanding of deep networks – your valuation of our work seems to be primarily a function of the extent to which such work directly implies improvements to specific metrics in practice. Consider that, if some of the significant methodological innovations in deep learning from the last five years were subjected to the same standard, they would have been dismissed – for example, diffusion models were not demonstrated to have sample quality anywhere close to the state-of-the-art GANs of the time [5]. The important aspect of these works was their conceptual insight that pointed the community towards more principled approaches and led to tremendous performance gains in the long run. We believe that CRATE has similar potential for future developments – it becomes possible to realize other novel improvements not just through empirical design, but also by using the guidance of principles from optimization and compression through the white-box approach.
> > >
> > > [4]: R. Vidal, “Attention: Self-Expression Is All You Need,” Sep. 29, 2021. Accessed: Apr. 05, 2022.
> > >
> > > [5]: J. Sohl-Dickstein et al. "Deep unsupervised learning using nonequilibrium thermodynamics." International conference on machine learning. PMLR, 2015.

---

> > > > ### Comment · Reviewer_YxEu · 2023-08-13
> > > >
> > > > Thank you for this direct and comprehensive response, I apologise for the bluntness of my previous comments, I was aiming to be brief.
> > > >
> > > > > We appreciate from your response here and to AAKz that you harbor some skepticism of research on the “model-centric” understanding of deep networks – your valuation of our work seems to be primarily a function of the extent to which such work directly implies improvements to specific metrics in practice. Consider that, if some of the significant methodological innovations in deep learning from the last five years were subjected to the same standard, they would have been dismissed – for example, diffusion models were not demonstrated to have sample quality anywhere close to the state-of-the-art GANs of the time [5].
> > > >
> > > > I agree that this is not a standard we should hold research to, and I understand that asking for state of the art results is not reasonable in this case. My goal was trying to make the correspondence between the theoretical results and experiments clear, I didn't have a good frame of reference of whether a $R^c$ value of 1600 means anything compared to 900, or whether sparsity of 0.2 is a good result for your model specifically. I can see that those numbers go down during training, which is what we should hope to see from the derivation of the unrolled optimization but it did not seem conclusive to me.
> > > >
> > > > # Figure 3/4 Sparsity Result
> > > >
> > > > To be precise, my misunderstanding is the sparsity you're measuring is at the output of a residual MLP and in [2] the authors only measure sparsity at the output of activation functions (ie no residual connection is present). Looking at the ISTA block in Figure 2, the output is an activation function. In my opinion a more useful metric in Figure 1 of the pdf rebuttal would be the output of the activation function of the MLP, ie:
> > > >
> > > > $$
> > > > z_l = MLP(LN(z_l')) + z_l'
> > > > $$
> > > >
> > > > $MLP(LN(z_l'))$ is functionally more similar to the output of the ISTA block and much more likely to be sparse, because it is the output of an activation function. It is no surprise for $z_l$ to exhibit no sparsity because $z_l'$ is the output of the self-attention block. Whether this is a relevant comparison to CRATE is another question, but I think that figure would have made much more sense if it had represented the sparsity present in the ViT.
> > > >
> > > > # Necessary vs Sufficient
> > > >
> > > > My statement was "sparse rate reduction is sufficient for learning useful functions in deep networks" and you reject "using the sparse rate reduction is necessary to construct high performance deep models". These statements are not saying the same thing.
> > > >
> > > > In fact I think we agree on what the hypothesis of your work is, ie that CRATE is sufficient to learn useful deep representations.
> > > >
> > > > # What white-box buys us
> > > >
> > > > I agree that the following contributions are valuable:
> > > >
> > > > 1. The concrete example given: "the QKV matrices in self-attention layers of ViT are redundant, and can be combined to save almost a factor of 4 in the overall parameter count"
> > > > 2. Visualizations in Figures 6 to 11 in the Appendices could be informative
> > > > 3. Building a network from "axis-aligned, hence sparse, orthogonal subspaces" is likely to open up other valuable lines of inquiry
> > > >
> > > > Your response here, and further reviewing the Appendices, have made that clear. I will update my review based on this.
> > > >
> > > > [2]: Z. Li et al., “The Lazy Neuron Phenomenon: On Emergence of Activation Sparsity in Transformers,” in The Eleventh International Conference on Learning Representations, 2023.
> > > > [5]: J. Sohl-Dickstein et al. "Deep unsupervised learning using nonequilibrium thermodynamics." International conference on machine learning. PMLR, 2015.

---

> > > > > ### Author Response · Authors · 2023-08-14
> > > > > **Discussion with Reviewer YxEu**
> > > > >
> > > > > Thank you for engaging with our rebuttal further and raising your score after reviewing our response. We are grateful for your valuable feedback on our work.
> > > > >
> > > > > > To be precise, my misunderstanding is the sparsity you're measuring is at the output of a residual MLP and in [2] the authors only measure sparsity at the output of activation functions (i.e., no residual connection is present).
> > > > >
> > > > > - For ViT, we measured the sparsity at the output of the residual MLP block, i.e., $z\_{\\ell} = \\mathrm{MLP}(\\mathrm{LN}(z^{\\prime}\_{\\ell})) + z^{\\prime}\_{\\ell}$.
> > > > > - For our proposed CRATE model, we measured the sparsity at the output of the ISTA block.
> > > > > We would like to clarify that, in CRATE, the residual connection appears in the ISTA block before the activation: see Eq.(16) in the main body for the precise structure.
> > > > >
> > > > > > In my opinion a more useful metric in Figure 1 of the pdf rebuttal would be the output of the activation function of the MLP.
> > > > >
> > > > > We agree that measuring the sparsity of $\\mathrm{MLP}(\\mathrm{LN}(z^{\\prime}\_{\\ell}))$ is another useful metric in addition to our existing measurements. Note that in the ViT, we have $\\mathrm{MLP}(\\mathrm{LN}(z^{\\prime}\_{\\ell})) = W\_2 \\sigma(W\_1 \\mathrm{LN}(z\_\\ell^\\prime))$, where $\sigma$ is the activation function in the MLP block (in the `timm` ViT this is GeLU while in [2] this is ReLU), so the MLP output is not actually the output of an activation function. We have visualized the sparsity of $\\mathrm{MLP}(\\mathrm{LN}(z^{\\prime}\_{\\ell}))$ for ViT and did not observe clear sparsity patterns, e.g., the ratio of non-sparse elements is larger than 90% for ViT-Base. We also measured the sparsity of $\\sigma(W\_1 \\mathrm{LN}(z^{\\prime}\_{\\ell}))$, the hidden layer activations; they are indeed sparse in a manner consistent with [2].
> > > > >
> > > > > We hope this clarification makes clear our point from the last paragraph of Part 1 of our previous response about the fact that token-level representations in CRATE are sparse, whereas for ViT, what is sparse is a less-exposed hidden layer output in the MLP. We thank you for your suggestion here; we will add these results and this discussion to our camera-ready version.
> > > > >
> > > > > We thank you again for your thoughtful review and valuable feedback! Please let us know if you have any other questions or comments during the discussion period.
> > > > >
> > > > > [2]: Z. Li et al., “The Lazy Neuron Phenomenon: On Emergence of Activation Sparsity in Transformers,” in The Eleventh International Conference on Learning Representations, 2023.

---

### Official Review · Reviewer_vvYz · 2023-07-05

**Soundness:** 3 good
**Presentation:** 3 good
**Contribution:** 3 good
**Rating:** 7
**Confidence:** 3

**Summary:**

This paper proposes to structure a classification pipeline based on Transformer networks using precisely defined mathematical operators that are designed to perform a gradient step to minimize a well defined objective, e.g. Lasso objective  sparse representation, or maximizing auto-correlation between "noisy" and "denoised" tokens.
As such, it not only propose a new architecture with some shared weights, but it also proposes an interpretation for the role of each block to achieve the goal of representation learning.
The pipeline is tested on popular Image Classification benchmark with end-to-end learning (ImageNet) as well as transfer learning (pre-trained on ImageNet, finetuned on CIFAR-10/100, Oxford flowers and pets.

**Strengths:**

The paper really proposes a new architecture based on an intuition " the objective of representation learning is to compress and transform the distribution of the data towards a mixture of low-dimensional Gaussian distributions supported on incoherent subspaces ".
They propose an architecture which resembles the visual transformers with a strong effort of modeling, i.e. trying to affect an objective to the usual transformers blocks (i.e. multi-head attention).
The proposed architecture is conceptually significantly simpler than ViT for example, and the performance drop is arguably very slight (-1% top1 on ImageNet). A posteriori analysis before and after training  (e.g. Figure 4) seem to validate the intuition of the authors for the coding rate aspect (it seems not so clear for sparsity).
This modelisation work in a tough work, and this paper is a significant contribution.


**Weaknesses:**

To me there is a caveat about the sparse coding hypothesis.
Line 252:
" In our implementation, motivated by Sun et al. [29] and Zarka et al. [31], we also add a non-negative constraint to Z^{l+1} "

This non-negative constraint is not just a detail, compared to a ReLU, the soft-thresholding reduced and possibly zeros the coding coefficient but it preserves the sign, i.e. the phase. The ReLU collapses the sign, i.e. the phase. This "detail" is also under-estimated in Sun et al. [29] and Zarka et al. [31] .

I suggest the authors to read the follow up work by Guth et al https://arxiv.org/pdf/2110.05283.pdf especially section 4 Phase Collapse Versus Amplitude Reduction. This might give further intuitions in this foggy world of modeling neural networks.  I'm not claiming that phase collapse is the good interpretation for this block, but for sure SoftShrink vs ReLU, i.e. sparse code vs non-negative sparse code is probably not a detail.

**Questions:**

How much does the performance degrade when replacing ReLU by softshrink ?

Do you have ideas / remarks / intuitions on the possible importance of this non-negativity constraint ? How does it articulate with the mixture of Gaussian / low-dimensioal subspace hypothesis ? Gaussians do no care about the sign, isn't it ?

Note that with non-negativity constraints , a subspace becomes a cone. Why would it be important to have a cone rather that a subspace ?

**Limitations:**

Apart the aspect mentioned below, this is a really strong work.

---

> ### Author Rebuttal · Authors · 2023-08-09
>
> Thank you for your comments, and your compliments on the quality of the contribution, the strength of the ideas, and the empirical insights. You bring up a very interesting point that the difference between the sparse coding and non-negative sparse coding formulations may seem important. In the sequel, we will attempt to explain why in fact the choice does not make too much difference conceptually, algorithmically, or empirically.
>
> > *How much does the performance degrade when replacing ReLU by softshrink ?*
>
> Regarding the empirical performance drop when using the regular sparse coding formulation, we report that the CRATE-Base model trains more or less the same (~67.6% top-1 accuracy on ImageNet-1K, which is a drop of 3.2% compared to the ReLU case, using $\lambda = 10$ and all other hyperparameters the same as in the original CRATE-Base evaluation on ImageNet; c.f., Table 1 in the paper and the rebuttal .pdf file). The results are summarized in Table 2 of the .pdf file. We will add this result to our camera-ready version. The message is that the two networks train comparably well, and one can push performance on either one by more dedicated hyperparameter tuning.
>
> >  *I'm not claiming that phase collapse is the good interpretation for this block, but for sure SoftShrink vs ReLU, i.e. sparse code vs non-negative sparse code is probably not a detail.*
>
> Thank you for bringing the interesting work [1] to our attention. Regarding the specific issue of phase collapse discussed in [1], it is our understanding that the effect of phase collapse which is analyzed in [1] is to better separate out the means of different classes within a classification task. While this may be a cause of the increase in classification accuracy reported above, we believe that our method will be applicable beyond just classification tasks. Indeed, in our framework, we contend that the purpose of the training process is to learn the local signal models at each layer (see e.g., Section 2.5). From this perspective, so long as the downstream training task requires semantically meaningful representations of the data distribution, the exact training configuration is of secondary importance. In particular, we may use self-supervised learning methods to learn the signal models, whence there may not be any well-defined notion of class mean, but such exploration is left to future work. In the camera-ready version, we will expand the discussion to include the work [1] and its discussion of phase collapse along with these clarifications.
>
> > *Do you have ideas / remarks / intuitions on the possible importance of this non-negativity constraint ? How does it articulate with the mixture of Gaussian / low-dimensional subspace hypotheses? Gaussians do not care about the sign, isn't it ?*
>
> > *Note that with non-negativity constraints, a subspace becomes a cone. Why would it be important to have a cone rather that a subspace?*
>
> This is a very good point; if we push a set of input representations through a non-negative sparse coding layer, they will always end up as non-negative, and thus cannot have marginal distributions equal to (an approximation of) a mixture of zero-mean Gaussians, but rather some other distribution. If we propagate this non-negative constraint to the sparse rate reduction problem, we obtain a “nonnegative sparse rate reduction” problem:
>
> $$\max_{f \in \mathcal{F}}\mathbb{E}[\Delta R(Z \mid U_{[K]}) - \lambda \\|Z\\|_{0} - \chi(Z \geq 0)]$$
>
> where $\chi$ denotes the characteristic function of a set and the algebraic definition of $\Delta R(Z \mid U_{[K]})$ (as a linear combination of logdet functions) is given in the paper. Our claim that the CRATE model transforms the data to a mixture of incoherent subspaces stems from the analysis of [2] of the minimizers of the rate reduction objective; in this view, understanding the questions you raise about representations in the presence of nonnegative soft thresholding amounts to whether optimal configurations in our nonnegative sparse rate reduction objective can be understood analogously. We sketch an argument below to this effect.
>
> Although formal analysis of the optimal points of the sparse rate reduction maximization problem is out of scope of this work, we see that the rate reduction maximization (i.e., $\max_{f \in \mathcal{F}} \mathbb{E}[\Delta R(Z \mid U_{[K]})]$) has optimal points characterized similarly to [2, Theorem A.6], namely that the representation of each distribution in the mixture is supported on a subspace with nearly isotropic covariance on this subspace, and the supporting subspaces are (nearly) orthogonal. Adding the sparsity term for some regularizer $\lambda$ would enforce the axis-alignment of the supporting subspaces; when adding in addition the nonnegativity constraint, following through the proof of [2, Theorem A.6] suggests that the argument goes through with suitable modifications (in particular, considering the conclusions for the covariance rather than $Z Z^T$). This sketch suggests that the statistical and geometric properties of the optimal representation remain the same when adding the non-negative constraint to the sparse rate reduction formulation. Since our CRATE model is derived by unrolling this objective, we believe this justifies the conceptual picture we describe in the submission around the ISTA block, although we use ReLU instead of soft thresholding. In the camera-ready version, we will expand the discussion to clarify these conceptual points.
>
> We again thank you for your detailed and interesting point, and hope we have provided satisfactory responses to your questions. Please let us know if you have further questions or comments.
>
> [1]: Guth, Florentin et al., "Phase collapse in neural networks." arXiv preprint arXiv:2110.05283 (2021).
>
> [2]: Yu, Yaodong et al., "Learning diverse and discriminative representations via the principle of maximal coding rate reduction." Advances in Neural Information Processing Systems 33 (2020).

---

> > ### Comment · Reviewer_vvYz · 2023-08-13
> > **Response to rebuttal**
> >
> > Thank you for this interesting discussion, happy to read that you have already lines of explanation for the non-negativity constraint, look forward reading the camera ready version.

---

> > > ### Author Response · Authors · 2023-08-14
> > > **Response to Reviewer vvYz**
> > >
> > > Thank you again for thoroughly reviewing our manuscript and response. We are grateful for your valuable feedback on our work, which will no doubt improve it. Please let us know if you have any other questions or comments during the discussion period.

---

### Official Review · Reviewer_13ev · 2023-07-10

**Soundness:** 3 good
**Presentation:** 3 good
**Contribution:** 2 fair
**Rating:** 5
**Confidence:** 2

**Summary:**

The authors propose a novel theoretical framework which shows that the popular transformer can be motivated by Maximizing Rate Reduction. The key idea of this work follows previous information-gain framework, mostly ReduNet, but with more special careful treatment on connections to transformer architecture design. By a few approximations, the authors show that maximizing rate reduction with sparsity constraint indeed derives a transformer-like deep structure. The derived white-box transformer-like architectures are verified on multiple datasets.

**Strengths:**

1) The paper is overall well written and easy to follow. Related Work section includes comprehensive surveys. Formulas are clearly explained. Experiments come with detailed settings.

2) The idea is novel and interesting. Especially, deriving multi-head attention from Maximizing Code Reduction is novel.

3) The proposed white-box architecture is verified on real-world datasets such as ImageNet-1k and compared to ViT models.

**Weaknesses:**

1) Although the paper claims that the proposed white-box model is competitive to ViT models, the numerical results seem not strong. On ImageNet-1k, the proposed model is clearly underperforming even with larger number of parameters.

2) The real power of ViTs is on high accuracy regime, where the model size is large. The authors only consider small model regime with low to medium accuracy, which lacks of convincing.

**Questions:**

1) Please consider larger model with ImageNet-1k top-1 accuracy above 80.0%

---

> ### Author Rebuttal · Authors · 2023-08-09
>
> Thank you for your comments and compliments on the idea being “novel and interesting”  as well as the exposition being “well written and easy to follow”.
>
> As we mentioned in our Public Response, the primary goal of our work is not meant to simply push the state-of-the-art in a particular metric, but rather to demonstrate the promise of the CRATE approach (i.e., white-box deep networks constructed via unrolled optimization). Nevertheless, in addition to achieving this goal, CRATE additionally obtains strong performance and has promising scaling behavior on increasingly larger-scale real world datasets. In particular, per your suggestion (as well as that of Reviewer `YxEu`) to try larger models, we investigated the performance of CRATE on ImageNet-1K when pretrained on ImageNet-21K and fine-tuned on ImageNet-1K. We found that in this setting, CRATE-Base could achieve 80.2% top-1 accuracy, which is comparable to the performance of ViT-Base with around 25% of the parameters -- see our Public Response for more precise details. We will add these experiments into our main tables and explicitly mention that our models perform slightly worse than ViTs in the paper text. Unfortunately, due to time limits, we could not pretrain CRATE-Large on ImageNet-21K; we will add the corresponding empirical results and comparison to ViT-Large in our camera-ready version.
>
> We hope that the points raised above resolve the doubts you have about this work. Please let us know if you have further questions or comments.

---

### Official Review · Reviewer_AAKz · 2023-07-12

**Soundness:** 4 excellent
**Presentation:** 4 excellent
**Contribution:** 4 excellent
**Rating:** 10
**Confidence:** 4

**Summary:**

This paper provides an interesting claim "the standard transformer block can be derived from alternating optimization on complementary parts of this objective: the multi-head self-attention operator can be viewed as a gradient descent step to compress the token sets by minimizing their lossy coding rate, and the subsequent multi-layer perceptron can be viewed as attempting to sparsify the representation of the tokens". This results in white-box transformer-like deep network architectures which are mathematically fully interpretable.

**Strengths:**

- Very interesting claim about transformer.
- Extensive experiments to verify the effectiveness of the proposed method.

**Weaknesses:**

I tried and failed to find any weaknesses. I really like such work on interpretable neural networks.

**Questions:**

No.

**Limitations:**

No.

---

> ### Author Rebuttal · Authors · 2023-08-09
>
> We are particularly grateful for and truly encouraged by your high assessment of our work. Thank you for dedicating your time and expertise to review our paper, and please let us know if you have any additional questions or comments during the discussion period.

---

> ### Comment · Reviewer_YxEu · 2023-08-11
> **No argument justifying award quality score of 10**
>
> I'm sorry to point this out but in order to justify a score of 10, it is necessary to write more than four sentences. The assertion that there are no weaknesses to the work does not seem reasonable, there are certainly weaknesses as pointed out by other reviewers.
>
> I would be very interested to read a convincing argument as to why this paper is award quality and I hope you will be able to revise this review to provide it. If not I hope an unjustified appraisal of award quality does not affect the final decision on this paper's acceptance.

---

### Author Rebuttal · Authors · 2023-08-09

First, we thank all reviewers for their insightful comments. We are particularly encouraged that reviewers have appreciated:
- The novelty and impact of our central ideas (`13ev`: “...deriving multi-head attention from Maximizing Code Reduction is novel”; `tJjQ`: “provid[ing] a significant extension to [prior work]”; `vvYz`: “...this paper is a significant contribution [to modelisation work]”);
- The benefits of the conceptual framework we have proposed (`vvYz`: “The proposed architecture is conceptually significantly simpler than ViT… and the performance drop is arguably very slight”; `tJjQ`: “The results are promising… the idea of white-box unrolling-based neural network design might be a possible alternative to current black-box design”);
- The quality of the exposition (`YxEu`: “The motivation and experimental results are both clearly stated”; `tJjQ`: “Overall the manuscript is well written and related works are properly cited and discussed, very insightful work”; `13ev`: “The paper is overall well written and easy to follow. Related Work section includes comprehensive surveys. Formulas are clearly explained. Experiments come with detailed settings.”);
- The insight presented by our empirical evaluations (`YxEu`: “The results in Table 1… address concerns that this method would reduce performance”; `AAKz`: “Extensive experiments… verify the effectiveness of the proposed method”).

In the remainder of this message, we wish to reiterate our key contributions and address certain concerns raised by the reviewers. In particular, we discuss new empirical results undertaken in response to issues raised by reviewers around CRATE’s scaling behavior, where we demonstrate ImageNet-1K accuracy above 80% after pretraining our CRATE-Base model on ImageNet-21K; these results are presented in the attached .pdf and discussed in full detail below.

## Key Contributions

Our central contribution is that we introduce a new transformer-like architecture (named CRATE), where each network layer/operator is constructed _ab initio_ from the principles of data compression and representation learning. This provides a clear and principled mathematical interpretability to transformer-like networks, by revealing the functions of each network layer while removing unnecessary redundancy from previous empirically designed transformers. In addition, we have shown through experiments that this cleaner and simpler architecture is competitive in performance with the base transformer models (such as ViT) in large-scale real-world vision tasks (e.g. classification on ImageNet). Empirical evaluations further confirm that the overall learned deep networks and their layers clearly perform the mathematical functions they were designed for, i.e., reducing the coding rate and sparsifying the learned representation. We believe this work has shown the promise of eventually bridging the gap between theory and practice of (transformer-like) deep networks.

## Comparison with SOTA; New Experimental Results
Of course, further improving the performance and demonstrating the potential of such new principled models is important -- we agree with several of the reviewers on this point. In this work, however, our goal is not to push the state-of-the-art per se, but rather to develop a more clear and systematic understanding of the extremely ubiquitous transformer-like deep network architectures, by developing a white-box model in this family of architectures. Several reviewers have, nevertheless, suggested some additional experimental improvements so that we may fairly compare to the ViT within more realistic regimes of data and compute; we wish to broadcast the results here. Most notably, as suggested by Reviewers `13ev` and `YxEu`, we have further scaled up the CRATE models. In particular, we pretrained on ImageNet-21K and fine-tuned on ImageNet-1K. As shown in Table 1 of the uploaded pdf file, with the CRATE-Base model (22.80M parameters), we achieve 80.2% top-1 accuracy; this is comparable to ViT-Base (~86M parameters, 83.9%) [1] with around 25% of the parameters. Here, we provide details about the experiments mentioned above. For pre-training on ImageNet-21K, we configure the learning rate to 1e-4, set the weight decay to 0.05, and use a batch size of 4,096. The total number of epochs is 90, with 10 warmup epochs. For fine-tuning on ImageNet-1K, we use the same set of parameters as described in Appendix B.1.2, with the exception of setting the learning rate to 5e-5 and having a total of 50 epochs. Unfortunately, due to time limits, we could not pretrain CRATE-Large on ImageNet-21K; we will add the corresponding empirical results and comparison to ViT-Large in our camera-ready version.

We again thank the reviewers for their insight and hope for a continually enlightening discussion period.

[1]: An Image is Worth 16x16 Words: Transformers for Image Recognition at Scale. Alexey Dosovitskiy, Lucas Beyer, Alexander Kolesnikov, Dirk Weissenborn, Xiaohua Zhai, Thomas Unterthiner, Mostafa Dehghani, Matthias Minderer, Georg Heigold, Sylvain Gelly, Jakob Uszkoreit, Neil Houlsby.  ICLR 2021.

---

### Decision · Program_Chairs · 2023-09-21

**Decision:**

Accept (poster)

**Comment:**

Reviewers appreciated the paper's novelty in connecting transformers with Maximizing Rate Reduction, noting its well-structured content and experiments. Strengths included its connection of information-gain frameworks to transformer architecture and verification on real-world datasets. Concerns revolved around the model's performance on ImageNet-1k compared to ViT models and the clarity of dense derivations. The sparse coding hypothesis and its relation to SoftShrink and ReLU were questioned. The authors properly addressed the questions in rebuttal and appendices. Overall, the paper offers promising insights and acs believe it is good for acceptance. It's highly recommended that the clarifications can be added in the camera ready version.